# A KERNEL DISTRIBUTION CLOSENESS TESTING

## ABSTRACT

The *distribution closeness testing* (DCT) assesses whether the distance between a distribution pair is at least $\epsilon$-far. Existing DCT methods mainly measure discrepancies between a distribution pair defined on discrete spaces (e.g., using total variation), which limits their applications to complex data (e.g., images). To extend DCT to more types of data, a natural idea is to introduce *maximum mean discrepancy* (MMD), a powerful measurement of the distributional discrepancy between two complex distributions, into DCT scenarios. However, we find that MMD's value can be the same for many pairs of distributions that have different norms in the same *reproducing kernel Hilbert space* (RKHS), which potentially have different closeness levels, making MMD less informative when assessing the closeness of multiple distribution pairs. To mitigate the issue, we design a new measurement of distributional discrepancy, *norm-adaptive MMD* (NAMMD), which scales MMD's value using the RKHS norms of distributions. Based on the asymptotic distribution of NAMMD, we finally propose the NAMMD-based DCT to assess the closeness level of a distribution pair. Theoretically, we prove that NAMMD-based DCT has higher test power compared to MMD-based DCT, with bounded type-I error, which is also validated by extensive experiments on many types of data (e.g., synthetic noise, real images).

## 1 INTRODUCTION

Distribution shift between training and test sets often exists in many real-world scenarios where machine learning methods are used [1, 2]. According to the classical machine learning theory [3], it is well-known that such a shift will influence the performance on the test set. In a worst case: having a very large distributional discrepancy between training and test data, we might have poor performance on test data for a model trained on the training data [4, 5]. The obtained poor performance can be explained by many theoretical results [4, 6]. However, we can also observe the other interesting phenomenon: it is also empirically proved that models trained on a large dataset (e.g., ImageNet [7]) can have good performance on relevant/similar downstream test data (e.g., Pascal VOC [8]) that is different from training dataset [9]. This means that, even if training and test data are from different distributions, we can still expect relatively good performance as they might be close to each other.

Therefore, *seeing to what statistically significant extent* two distributions are close to each other is important and might help us decide if we really need to adapt a model when we observe upcoming data that follow a different distribution from training data. Two-sample testing (TST) can naturally see if training and test data are from the same distribution [10], but it is less useful in the phenomenon above as we might also have good performance when the training and test data are close to each other. Fortunately, in theoretical computer science, researchers have proposed *distribution closeness testing* (DCT) to see if the distance between a distribution pair is at least $\epsilon$-far, including TST as a specific case with $\epsilon = 0$ [11–14]. The DCT exactly fits the aim of *seeing to what statistically significant extent* two distributions are close to each other, and has been used to evaluate Markov chain mixing time [15], test language membership [16] and analyze feature combinations [17].

However, existing DCT methods mainly measure closeness using total variation [18–21], and primarily focus on the theoretical analysis of the sample complexity of sub-linear algorithms applied to *discrete distributions* defined on a support set only containing finite elements (e.g., distribution defined on a positive-integer domain $\{1, 2, ..., n\}$). This limits their applications to complex data, which is often used in machine learning tasks (e.g., image classification). Although it is possible to

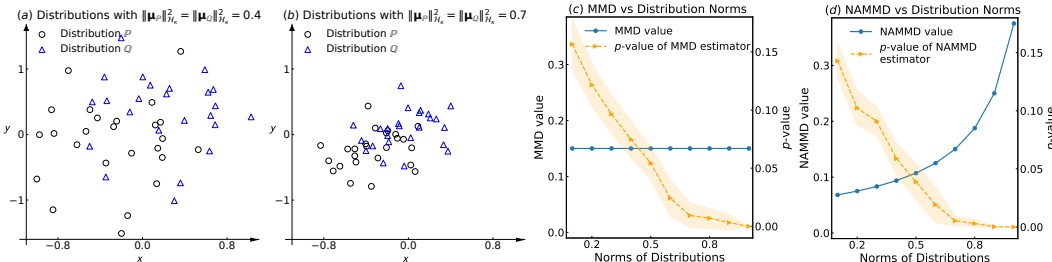

**Figure 1:** MMD is less informative when two distributions are different. All visualizations are presented with a constant MMD value $\|\boldsymbol{\mu}_{\mathbb{P}} - \boldsymbol{\mu}_{\mathbb{Q}}\|^2_{\mathcal{H}_\kappa} = 0.15$ on the Gaussian kernel with bandwidth 1, extendable to other kernels of the form: $\kappa(\boldsymbol{x}, \boldsymbol{x}') = \Psi(\boldsymbol{x} - \boldsymbol{x}') \leq K$ with $K > 0$ for a positive-definite $\Psi(\cdot)$ and $\Psi(\boldsymbol{0}) = K$ (see Appendix F for kernel limitations and Appendix D.1.1 for experimental details.). Subfigures (a) and (b) depict distributions $\mathbb{P}$ and $\mathbb{Q}$ with varying norms ($\|\boldsymbol{\mu}_{\mathbb{P}}\|^2_{\mathcal{H}_\kappa}$ and $\|\boldsymbol{\mu}_{\mathbb{Q}}\|^2_{\mathcal{H}_\kappa}$), yet they yield the same MMD value in two subfigures, indicating that MMD is less informative. Subfigure (c) presents the MMD value and the $p$-values of its estimator in TST. Subfigure (d) presents the NAMMD value and the $p$-values of its estimator in TST. It is evident that NAMMD exhibits a stronger correlation with the $p$-value compared to MMD. Namely, *larger NAMMD corresponds to smaller $p$-value*, while MMD *keeps the same value* when the $p$-value changes.

discretize complex data to a simple support set (then using existing DCT methods [22]), it is not easy to maintain intrinsic structures and patterns of complex data after the discretization [23, 24].

To extend DCT to more types of data, a natural idea is to introduce *maximum mean discrepancy* (MMD), a powerful kernel-based measurement of the distributional discrepancy between two complex distributions [25, 26], into DCT scenarios. MMD provides a versatile approach across both discrete and continuous domains, and many approaches have extended it to various scenarios, including mean embeddings with test locations [27, 28], local difference exploration [29], stochastic process [30], multiple kernel [31, 32], adversarial learning [33], and domain adaptation [34]. Yet, no one has explored how to extend DCT to complex data with MMD.

**Limitations of Using MMD Directly in DCT.** In this paper, however, we find it is not ideal to directly use MMD in DCT, because the MMD is less informative when comparing the closeness levels of different distribution pairs for a fixed kernel $\kappa$. Specifically, the MMD value can be the same for many pairs of distributions that have different norms in the RKHS $\mathcal{H}_\kappa$, which potentially have different closeness levels. We present an example to analyze the above issue in Figure 1. The empirical results show that the MMD estimator of a distribution pair $(\mathbb{P}, \mathbb{Q})$ with the same MMD value but larger RKHS norms tend to exhibit a smaller $p$-value (See Figure 1c) in assessing equivalence between two distributions, i.e., TST, which indicates that $\mathbb{P}$ and $\mathbb{Q}$ are less likely satisfy the null hypothesis (i.e., $\mathbb{P} = \mathbb{Q}$). This reflects that $\mathbb{P}$ and $\mathbb{Q}$ are more significantly different, i.e., less close to each other[1].

**A Norm-Adaptive MMD Approach for DCT.** To mitigate the above issue, we design a new measurement of distributional discrepancy, *norm-adaptive MMD* (NAMMD), which scales MMD's value using the RKHS norms of distributions. The motivation is that distribution pairs with the same MMD values but larger RKHS norms tend to be farther apart in distributional closeness. Accordingly, NAMMD increases its value as the RKHS norms rise, while the underlying MMD value remains fixed. Combining both of Figure 1c and 1d, we can find that our NAMMD exhibits a stronger correlation with the $p$-value compared to MMD. Namely, *larger NAMMD corresponds to smaller $p$-value* (Figure 1d), while MMD *keeps the same value* when the $p$-value changes (Figure 1c).

Eventually, we propose a new NAMMD-based DCT, *which derives its testing threshold from the analytical and asymptotic null distribution of NAMMD*, with a guaranteed bound on type-I error. In Theorem 9, we prove that, under a specific scenario, if MMD-based DCT rejects a null hypothesis correctly, NAMMD-based DCT also rejects it with high probability; moreover, NAMMD-based DCT can reject a null hypothesis in cases where MMD-based DCT fails, leading to higher test power. We also provide an analysis regarding the sample complexity of NAMMD-based DCT (Theorem 7), i.e., how many samples we need to correctly reject a null hypothesis with high probability.

In experiments, we validate NAMMD-based DCT on benchmark datasets in comparison with state-of-the-art methods. Furthermore, considering practical scenarios in testing distribution closeness, we

---

[1]See Appendix D.1.2, which further explains why the $p$-values in Figure 1c decrease via changes in the standard deviation of the MMD estimator.

might use a reference (known) pair of distributions $\mathbb{P}_1$ and $\mathbb{Q}_1$, with their distance serving as the $\epsilon$; and we then test whether the distance between an unknown distribution pair $\mathbb{P}_2$ and $\mathbb{Q}_2$ exceeds that between $\mathbb{P}_1$ and $\mathbb{Q}_1$. Given this, we conduct experiments in three practical case studies to demonstrate the effectiveness of our NAMMD test in evaluating whether a classifier performs relatively similarly across training and test datasets, compared to a prespecified level, without labels (Section 5.2).

## 2 PRELIMINARIES

**Distribution Closeness Testing (DCT).** Denote by $\mathbb{P}$ and $\mathbb{Q}$ two unknown Borel probability measures over an instance space $\mathcal{X} \subseteq \mathbb{R}^d$. The DCT assesses whether $\mathbb{P}$ and $\mathbb{Q}$ are $\epsilon$-far from each other under a closeness measurement $d$, where $d$ can be any distance or metric that quantifies the closeness or difference between probability distributions. For convenience, we assume that $d$ is scaled to $[0, 1]$. Formally, given $d$, the goal of DCT is to test between the null and alternative hypotheses as follows

$$\boldsymbol{H}_0 : d(\mathbb{P}, \mathbb{Q}) \leq \epsilon \quad \text{and} \quad \boldsymbol{H}_1 : d(\mathbb{P}, \mathbb{Q}) > \epsilon \,,$$

where $\epsilon \in [0, 1)$ is the predetermined closeness parameter.

**Maximum Mean Discrepancy (MMD).** The MMD [25] is a typical kernel-based distance between two distributions. Let $\kappa : \mathcal{X} \times \mathcal{X} \to \mathbb{R}$ be the kernel of a reproducing kernel Hilbert space $\mathcal{H}_\kappa$, with feature map $\kappa(\cdot, \boldsymbol{x}) \in \mathcal{H}_\kappa$ and $0 \leq \kappa(\boldsymbol{x}, \boldsymbol{y}) \leq K$. The kernel mean embeddings [35, 36] of distributions $\mathbb{P}$ and $\mathbb{Q}$ are given as

$$\boldsymbol{\mu}_\mathbb{P} = E_{\boldsymbol{x} \sim \mathbb{P}}[\kappa(\cdot, \boldsymbol{x})] \quad \text{and} \quad \boldsymbol{\mu}_\mathbb{Q} = E_{\boldsymbol{y} \sim \mathbb{Q}}[\kappa(\cdot, \boldsymbol{y})] \,.$$

We now define the MMD of $\mathbb{P}$ and $\mathbb{Q}$ as

$$\text{MMD}^2(\mathbb{P}, \mathbb{Q}, \kappa) = \|\boldsymbol{\mu}_\mathbb{P} - \boldsymbol{\mu}_\mathbb{Q}\|_{\mathcal{H}_\kappa}^2 \quad = \quad \|\boldsymbol{\mu}_\mathbb{P}\|_{\mathcal{H}_\kappa}^2 + \|\boldsymbol{\mu}_\mathbb{Q}\|_{\mathcal{H}_\kappa}^2 - 2\langle \boldsymbol{\mu}_\mathbb{P}, \boldsymbol{\mu}_\mathbb{Q} \rangle_{\mathcal{H}_\kappa} \,,$$

where $\boldsymbol{x}, \boldsymbol{x}' \sim \mathbb{P}$, $\boldsymbol{y}, \boldsymbol{y}' \sim \mathbb{Q}$, and $\| \cdot \|_{\mathcal{H}_\kappa}^2 = \langle \cdot, \cdot \rangle_{\mathcal{H}_\kappa}$ is the inner product in RKHS $\mathcal{H}_\kappa$.

**Two-sample Testing (TST).** TST aims to assess a fundamentally different hypothesis testing problem compared to DCT, with the null and the alternative hypotheses as follows

$$\boldsymbol{H}_0^\text{t} : \mathbb{P} = \mathbb{Q} \quad \text{and} \quad \boldsymbol{H}_1^\text{t} : \mathbb{P} \neq \mathbb{Q} \,,$$

For characteristic kernels, $\text{MMD}(\mathbb{P}, \mathbb{Q}; \kappa) = 0$ if and only if $\mathbb{P} = \mathbb{Q}$. Hence, MMD can be readily applied to the two-sample testing. A wide range of discrepancy measurements has also been developed for comparing probability distributions in the TST setting, including integral probability metrics (IPMs) such as total variation and Wasserstein distance, and $f$-divergences such as KL and $\chi^2$.

However, TST tests the equality $\mathbb{P} = \mathbb{Q}$, which guarantees exchangeability between the two i.i.d. samples from the distribution, and enables permutation calibration without requiring a closed-form asymptotic null distribution. In contrast, DCT tests a margin-based composite null $d(\mathbb{P}, \mathbb{Q}) \leq \epsilon$, under which samples from $\mathbb{P}$ and $\mathbb{Q}$ are not interchangeable. Permutation calibration is therefore invalid, and a closed-form asymptotic null distribution becomes essential. Many discrepancies used in TST, including total variation and KL divergence, do not admit a tractable asymptotic null distribution. Furthermore, although general nonparametric frameworks exist for estimating such discrepancies in continuous settings [37, 38], certain metrics such as total variation require additional structural assumptions to achieve consistency [39, 40], and without such assumptions they can not guarantee a correct ordering of distribution pairs by their closeness, which is essential for DCT. Considering both consistency and tractable asymptotic null distribution, MMD provides a practically useful and theoretically well-behaved discrepancy measure for DCT.

## 3 NAMMD-BASED DISTRIBUTION CLOSENESS TESTING

As discussed in introduction and Figure 1, while MMD can detect whether two distributions are identical, it is less informative in measuring the closeness between distributions. Specifically, different distribution pairs with the same MMD values but larger RKHS norms tend to be farther apart in distributional closeness, as reflected by the decrease in their TST $p$-values when the norms increase.

**NAMMD and Its Asymptotic Property.** To mitigate this issue, we define our NAMMD distance by rescaling the MMD value using the RKHS norms of the distributions. This scaling ensures that NAMMD increases as the RKHS norms rise, while the underlying MMD value remains fixed.

**Definition 1.** For a kernel $\kappa$ with $\mathcal{H}_\kappa$ and $0 \leq \kappa(\boldsymbol{x}, \boldsymbol{y}) \leq K$, we define the *norm-adaptive maximum mean discrepancy* (NAMMD) w.r.t. distributions $\mathbb{P}$ and $\mathbb{Q}$ as follows:

$$\text{NAMMD}(\mathbb{P}, \mathbb{Q}; \kappa) = \frac{\|\boldsymbol{\mu}_\mathbb{P} - \boldsymbol{\mu}_\mathbb{Q}\|^2_{\mathcal{H}_\kappa}}{4K - \|\boldsymbol{\mu}_\mathbb{P}\|^2_{\mathcal{H}_\kappa} - \|\boldsymbol{\mu}_\mathbb{Q}\|^2_{\mathcal{H}_\kappa}} \ . \tag{1}$$

Here, the numerator of NAMMD is $\text{MMD}^2(\mathbb{P}, \mathbb{Q}; \kappa)$, which lies in $[0, 2K]$ for any bounded, shift-invariant kernel $\kappa(\boldsymbol{x}, \boldsymbol{x}') = \Psi(\boldsymbol{x} - \boldsymbol{x}')$ with $\Psi(\boldsymbol{0}) = K > 0$ and $\Psi(\cdot) \leq K$ (positive-definite). The denominator is the scaling term $4K - \|\boldsymbol{\mu}_\mathbb{P}\|^2_{\mathcal{H}_\kappa} - \|\boldsymbol{\mu}_\mathbb{Q}\|^2_{\mathcal{H}_\kappa}$, which lies in $[2K, 4K]$. Consequently, $0 \leq \text{NAMMD}(\mathbb{P}, \mathbb{Q}; \kappa) \leq 1$, and NAMMD approaches 1 when two distributions $\mathbb{P}$ and $\mathbb{Q}$ are *well-separated* from each other and both are *highly concentrated*[2].

In NAMMD, we essentially capture differences between two distributions using their characteristic kernel mean embeddings (i.e. $\boldsymbol{\mu}_\mathbb{P}$ and $\boldsymbol{\mu}_\mathbb{Q}$), which uniquely represent distributions and capture distinct characteristics for effective comparison [41]. Equivalently, NAMMD can be viewed as MMD scaled by the RKHS variances of distributions $\mathbb{P}$ and $\mathbb{Q}$. Specifically, for a bounded, shift-invariant kernel $\kappa(\boldsymbol{x}, \boldsymbol{x}') = \Psi(\boldsymbol{x} - \boldsymbol{x}') \leq K$ with $K > 0$, where $\Psi(\cdot)$ is positive definite and $\Psi(\boldsymbol{0}) = K$, the variances take the form $\text{Var}(\mathbb{P}; \kappa) = E_{\boldsymbol{x} \sim \mathbb{P}}[\kappa(\boldsymbol{x}, \boldsymbol{x})] - \|\boldsymbol{\mu}_\mathbb{P}\|^2_{\mathcal{H}_\kappa} = K - \|\boldsymbol{\mu}_\mathbb{P}\|^2_{\mathcal{H}_\kappa}$ and $\text{Var}(\mathbb{Q}; \kappa) = K - \|\boldsymbol{\mu}_\mathbb{Q}\|^2_{\mathcal{H}_\kappa}$. Hence, we have $\text{NAMMD}(\mathbb{P}, \mathbb{Q}; \kappa) = \text{MMD}(\mathbb{P}, \mathbb{Q}; \kappa) / (2K + \text{Var}(\mathbb{P}; \kappa) + \text{Var}(\mathbb{Q}; \kappa))$.

Several prior normalization strategies for kernel tests have been developed in the TST setting, but their objectives differ from ours. For example, Kernel FDA [42] focuses on testing equality of the mean and covariance of two distributions; Spectral-MMD [43] introduces normalization to achieve minimax separation rates under $\Delta$-separated alternatives; and analytic or feature-based approaches [27, 28] are constructed to improve sensitivity to local discrepancies. These methods are highly effective for detecting differences under $P = Q$, the classical TST null. However, DCT places more emphasis on distinguishing different closeness levels for different distribution pairs, and NAMMD is introduced to mitigate a specific limitation of MMD in this regard.

In practice, $\mathbb{P}$ and $\mathbb{Q}$ are generally unknown, and we can only observe two i.i.d. samples[3]

$$X = \{\boldsymbol{x}_i\}_{i=1}^m \sim \mathbb{P}^m \ \text{ and } \ Y = \{\boldsymbol{y}_j\}_{j=1}^m \sim \mathbb{Q}^m \ .$$

Based on two samples $X$ and $Y$, we introduce the empirical estimator of our NAMMD as follows

$$\widehat{\text{NAMMD}}(X, Y; \kappa) = \frac{\sum_{i \neq j} H_{i,j}}{\sum_{i \neq j} [4K - \kappa(\boldsymbol{x}_i, \boldsymbol{x}_j) - \kappa(\boldsymbol{y}_i, \boldsymbol{y}_j)]},$$

where $H_{i,j} = \kappa(\boldsymbol{x}_i, \boldsymbol{x}_j) + \kappa(\boldsymbol{y}_i, \boldsymbol{y}_j) - \kappa(\boldsymbol{x}_i, \boldsymbol{y}_j) - \kappa(\boldsymbol{y}_i, \boldsymbol{x}_j)$. Then, we prove an asymptotic distribution of NAMMD when two distributions are different in the following theorem.

**Lemma 2.** *If* $\text{NAMMD}(\mathbb{P}, \mathbb{Q}; \kappa) = \epsilon > 0$*, we have*

$$\sqrt{m}(\widehat{\text{NAMMD}}(X, Y; \kappa) - \epsilon) \xrightarrow{d} \mathcal{N}(0, \sigma^2_{\mathbb{P},\mathbb{Q}}) \ ,$$

*where* $\sigma_{\mathbb{P},\mathbb{Q}} = \sqrt{4E[H_{1,2}H_{1,3}] - 4(E[H_{1,2}])^2} / (4K - \|\boldsymbol{\mu}_\mathbb{P}\|^2_{\mathcal{H}_\kappa} - \|\boldsymbol{\mu}_\mathbb{Q}\|^2_{\mathcal{H}_\kappa})$*, and the expectation are taken over* $\boldsymbol{x}_1, \boldsymbol{x}_2, \boldsymbol{x}_3 \sim \mathbb{P}^3$ *and* $\boldsymbol{y}_1, \boldsymbol{y}_2, \boldsymbol{y}_3 \sim \mathbb{Q}^3$*.*

We now present the DCT by taking our NAMMD as the closeness measure, along with an appropriately estimated testing threshold from the above analytical and asymptotic distribution.

**NAMMD-DCT Testing Procedure.** In the following, we instantiate the distribution closeness testing in Section 2 using NAMMD as the closeness measurement.

**Definition 3.** Given the closeness parameter $\epsilon \in (0, 1)$, the goal is to test between hypotheses

$$\boldsymbol{H}_0 : \text{NAMMD}(\mathbb{P}, \mathbb{Q}; \kappa) \leq \epsilon \qquad \text{and} \qquad \boldsymbol{H}_1 : \text{NAMMD}(\mathbb{P}, \mathbb{Q}; \kappa) > \epsilon \ ,$$

with the significance level $\alpha \in (0, 1)$.

---

[2]See Appendix B.1, which provides further details on the conditions under which NAMMD approaches 1.

[3]Following Liu et al. [26], we assume equal size for two samples to simplify the notation, yet our results can be easily extended to unequal sample sizes by using multi-sample $U$-statistic [44]. See Appendix B.2 for details.

To conduct a hypothesis testing procedure for distribution closeness, we first estimate the testing threshold $\hat{\tau}_\alpha$ under the null hypothesis $\boldsymbol{H}_0$ : $\mathrm{NAMMD}(\mathbb{P}, \mathbb{Q}; \kappa) \leq \epsilon$ at significance level $\alpha$. The null hypothesis is composite, consisting of the case $\mathrm{NAMMD}(\mathbb{P}, \mathbb{Q}; \kappa) = \epsilon$ and the case $\mathrm{NAMMD}(\mathbb{P}, \mathbb{Q}; \kappa) < \epsilon$. Since the value $\mathrm{NAMMD}(\mathbb{P}, \mathbb{Q}; \kappa)$ is unknown, we set the testing threshold $\hat{\tau}_\alpha$ as the estimated $(1-\alpha)$-quantile of the the asymptotic Gaussian distribution of $\widehat{\mathrm{NAMMD}}(X, Y; \kappa)$ under the case where $\mathrm{NAMMD}(\mathbb{P}, \mathbb{Q}; \kappa) = \epsilon$ (i.e., the least-favorable boundary of the composite null hypothesis) as shown in Lemma 2. For the asymptotic distribution, the term $\sigma_{\mathbb{P},\mathbb{Q}}^2$ is unknown and we use its estimator

$$\sigma_{X,Y} = \frac{\sqrt{((4m-8)\zeta_1 + 2\zeta_2)/(m-1)}}{(m^2 - m)^{-1} \sum_{i \neq j} 4K - \kappa(\boldsymbol{x}_i, \boldsymbol{x}_j) - \kappa(\boldsymbol{y}_i, \boldsymbol{y}_j)}, \tag{2}$$

where $\zeta_1$ and $\zeta_2$ are standard variance components of the MMD [45, 46] (See Appendix B.3). Lemma 4 shows that the estimator $\sigma_{X,Y}^2$ converges to $\sigma_{\mathbb{P},\mathbb{Q}}^2$ at a rate of $O(1/\sqrt{m})$.

We now have the testing threshold for null hypothesis $\boldsymbol{H}_0 : \mathrm{NAMMD}(\mathbb{P}, \mathbb{Q}; \kappa) \leq \epsilon$ with $\epsilon \in (0, 1)$ as

$$\hat{\tau}_\alpha = \epsilon + \sigma_{X,Y} \mathcal{N}_{1-\alpha}/\sqrt{m}, \tag{3}$$

where $\mathcal{N}_{1-\alpha}$ is the $(1 - \alpha)$-quantile of the standard normal distribution $\mathcal{N}(0, 1)$.

Finally, we have the following testings procedure with testing threshold $\hat{\tau}_\alpha$

$$h(X, Y; \kappa) = \mathbb{I}[\widehat{\mathrm{NAMMD}}(X, Y; \kappa) > \hat{\tau}_\alpha]. \tag{4}$$

**Performing DCT in Practice.** We have demonstrated the NAMMD-based DCT above, yet it is still not clear how the $\epsilon$ of Definition 3 should be set in practice. Normally, when we want to test the closeness, we often have a reference pair of distributions $\mathbb{P}_1$ and $\mathbb{Q}_1$ where the closeness between $\mathbb{P}_1$ and $\mathbb{Q}_1$ is acceptable/satisfactory. For example, although ImageNet and Pascal VOC are from different distributions, the model trained on ImageNet can still have good performance on Pascal VOC. Thus, we can use the NAMMD's empirical value between ImageNet and Pascal VOC as the prespecified $\epsilon$ in this case. Then, given two samples $X$ and $Y$ drawn from an unknown pair of distributions $\mathbb{P}_2$ and $\mathbb{Q}_2$ respectively, we seek to determine whether the distance between $\mathbb{P}_2$ and $\mathbb{Q}_2$ is as close or closer to that between $\mathbb{P}_1$ and $\mathbb{Q}_1$, by applying distribution closeness testing. Here, given the specified $\epsilon$, and this DCT problem can be formalized by Definition 3 with hypotheses as

$$\boldsymbol{H}_0 : \mathrm{NAMMD}(\mathbb{P}_2, \mathbb{Q}_2; \kappa) \leq \epsilon \qquad \text{and} \qquad \boldsymbol{H}_1 : \mathrm{NAMMD}(\mathbb{P}_2, \mathbb{Q}_2; \kappa) > \epsilon.$$

Finally, we can perform NAMMD testing procedure with samples $X$ and $Y$.

**Kernel Selection and related Works.** For kernel selection in DCT, we select a fixed global kernel for all distribution pairs, which is essential for effectively comparing their closeness levels under a unified measurement. However, existing kernel selections are primarily designed for TST [26, 47], selecting a kernel to maximize the $t$-statistic in test power estimation to distinguish a fixed distribution pair. In DCT, deriving a test power estimator with several different distribution pairs remains an open question and we follow the TST approaches to select a kernel to distinguish between $\mathbb{P}_1$ and $\mathbb{Q}_1$ in practice (see Appendix B.5). See Appendix B.4 for more related works, including those on testing-threshold estimation, kernel selection approaches, etc.

**Applying NAMMD to Two-Sample Testing.** Although the NAMMD is specially designed for DCT, it is still a statistic to measure the distributional discrepancy between two distributions. Thus, it is interesting to apply it to two-sample testing (TST) scenarios. In TST, we aim to assess the equivalence between distributions $\mathbb{P}$ and $\mathbb{Q}$, where the null hypothesis assumes $\mathbb{P} = \mathbb{Q}$ and is tested against the alternative hypothesis $\mathbb{P} \neq \mathbb{Q}$. Following MMD-based approaches to TST [47], we use a standard permutation test to estimate the test threshold $\hat{\tau}_\alpha$, which estimate the null distribution by repeatedly re-computing estimator with the samples randomly re-assigned to $X$ or $Y$ (see Appendix B.6).

## 4 THEORETICAL ANALYSIS OF NAMMD-BASED DCT

In this section, we make theoretical investigations regarding NAMMD-based DCT and compare NAMMD and the MMD in addressing the DCT problem. All the proofs are presented in Appendix C. We first provide theoretical guarantees for the variance estimation and concentration properties of the NAMMD estimator. Specifically, for the variance estimator $\sigma_{X,Y}$ in Eqn. (2), we have

**Lemma 4.** *Given samples $X$ and $Y$ with size $m$, we have that $\left|E[\sigma_{X,Y}^2] - \sigma_{\mathbb{P},\mathbb{Q}}^2\right| = O(1/\sqrt{m})$.*

We now present the large deviation bound for our NAMMD estimator.

**Lemma 5.** *The following holds over sample $X$ and $Y$ of size $m$,*

$$\Pr\left(|\widehat{\text{NAMMD}}(X,Y;\kappa) - \text{NAMMD}(\mathbb{P},\mathbb{Q};\kappa)| \geq t\right) \leq 4\exp(-mt^2/9) \ \ \text{for } t > 0.$$

Lemma 4 establishes the convergence rate of the variance estimator $\sigma_{X,Y}$, showing that the estimation error in expectation decays at the rate $O(1/\sqrt{m})$ with sample size $m$. Lemma 5 presents a large deviation bound for the NAMMD estimator, indicating that the probability of deviation from its population value decays exponentially with rate $\exp(-mt^2/9)$.

Next, we study type-I error control for NAMMD-based DCT.

**Theorem 6.** *Under the null hypothesis $\boldsymbol{H}_0 : \text{NAMMD}(\mathbb{P},\mathbb{Q};\kappa) \leq \epsilon$ with $\epsilon \in (0,1)$, the type-I error of NAMMD-based DCT is bounded by $\alpha$, i.e., $\Pr_{\boldsymbol{H}_0}(h(X,Y;\kappa) = 1) \leq \alpha$.*

Theorem 6 shows the validity of the NAMMD-based DCT as type-I error of the proposed test can be *bounded* by $\alpha$. We then analyze the sample complexity regarding NAMMD-based DCT to correctly reject the null hypothesis with high probability as follows.

**Theorem 7.** *For our NAMMD test, as formalized in Eqn. 4, we correctly reject null hypothesis $\boldsymbol{H}_0 : \text{NAMMD}(\mathbb{P},\mathbb{Q};\kappa) \leq \epsilon \in (0,1)$ with probability at least $1 - \upsilon$ given the sample size*

$$m \geq \left(2 * \mathcal{N}_{1-\alpha} + \sqrt{9\log 2/\upsilon}\right)^2 / (\text{NAMMD}(\mathbb{P},\mathbb{Q};\kappa) - \epsilon)^2 .$$

This theorem shows that the ratio $1/(\text{NAMMD}(\mathbb{P},\mathbb{Q};\kappa) - \epsilon)^2$ is the main quantity dictating the sample complexity of our NAMMD test under alternative hypothesis $\boldsymbol{H}_1 : \text{NAMMD}(\mathbb{P},\mathbb{Q};\kappa) > \epsilon$.

**Comparison between NAMMD-based DCT and MMD-based DCT.** As demonstrated in Section 3, in practice, we might often need a reference pair to confirm the value of $\epsilon$, thus, we first reformalize the DCT testing procedure with the reference pair, which is shown in the following definition.

**Definition 8.** Given the reference distributions $\mathbb{P}_1$ and $\mathbb{Q}_1$, and samples $X$ and $Y$ drawn from unknown distributions $\mathbb{P}_2$ and $\mathbb{Q}_2$, the goal of DCT is to correctly determine whether the distance between $\mathbb{P}_2$ and $\mathbb{Q}_2$ is larger than that between $\mathbb{P}_1$ and $\mathbb{Q}_1$. To compare the test power, we perform NAMMD-based DCT and MMD-based DCT separately, under scenarios where the following two null hypotheses for NAMMD-based DCT and MMD-based DCT are simultaneously false:

$$\boldsymbol{H}_0^N : \text{NAMMD}(\mathbb{P}_2,\mathbb{Q}_2,\kappa) \leq \epsilon^N \qquad \text{and} \qquad \boldsymbol{H}_0^M : \text{MMD}(\mathbb{P}_2,\mathbb{Q}_2,\kappa) \leq \epsilon^M \ ,$$

and following alternative hypotheses simultaneously hold true:

$$\boldsymbol{H}_1^N : \text{NAMMD}(\mathbb{P}_2,\mathbb{Q}_2,\kappa) > \epsilon^N \qquad \text{and} \qquad \boldsymbol{H}_1^M : \text{MMD}(\mathbb{P}_2,\mathbb{Q}_2,\kappa) > \epsilon^M \ ,$$

where $\epsilon^N = \text{NAMMD}(\mathbb{P}_1,\mathbb{Q}_1,\kappa)$ and $\epsilon^M = \text{MMD}(\mathbb{P}_1,\mathbb{Q}_1,\kappa)$.

Based on the specific setting in the above definition, we show in the following theoretical analysis that NAMMD-based DCT can still provide advantages even when the limitation of MMD (see Section 1) does not arise, provided that a certain norm condition holds.

**Theorem 9.** *Under $\boldsymbol{H}_1^N : \text{NAMMD}(\mathbb{Q}_2,\mathbb{P}_2,\kappa) > \epsilon^N$ and $\boldsymbol{H}_1^M : \text{MMD}(\mathbb{Q}_2,\mathbb{P}_2,\kappa) > \epsilon^M$, and assuming $\|\boldsymbol{\mu}_{\mathbb{P}_1}\|_{\mathcal{H}_\kappa}^2 + \|\boldsymbol{\mu}_{\mathbb{Q}_1}\|_{\mathcal{H}_\kappa}^2 < \|\boldsymbol{\mu}_{\mathbb{P}_2}\|_{\mathcal{H}_\kappa}^2 + \|\boldsymbol{\mu}_{\mathbb{Q}_2}\|_{\mathcal{H}_\kappa}^2$, then the following relation holds with probability at least $1 - \exp\left(-m\Delta^2(4K - \|\boldsymbol{\mu}_{\mathbb{P}_2}\|_{\mathcal{H}_\kappa}^2 - \|\boldsymbol{\mu}_{\mathbb{Q}_2}\|_{\mathcal{H}_\kappa}^2)^2/(4K^2(1-\Delta)^2)\right)$,*

$$\sqrt{m}\widehat{\text{MMD}}(X,Y,\kappa) > \tau_\alpha^M \ \ \Rightarrow \ \ \sqrt{m}\widehat{\text{NAMMD}}(X,Y,\kappa) > \tau_\alpha^N \ ,$$

*where $\tau_\alpha^M$ and $\tau_\alpha^N$ are asymptotic $(1-\alpha)$-thresholds of the null distributions of $\sqrt{m}\widehat{\text{MMD}}$ and $\sqrt{m}\widehat{\text{NAMMD}}$, respectively. Given $\sigma_M$ defined in Eqn. (6) (Appendix C.6.1), it follows that*

$$\Delta = \sqrt{m}\text{NAMMD}(\mathbb{P}_1,\mathbb{Q}_1,\kappa)\frac{\|\boldsymbol{\mu}_{\mathbb{P}_2}\|_{\mathcal{H}_\kappa}^2 + \|\boldsymbol{\mu}_{\mathbb{Q}_2}\|_{\mathcal{H}_\kappa}^2 - \|\boldsymbol{\mu}_{\mathbb{P}_1}\|_{\mathcal{H}_\kappa}^2 - \|\boldsymbol{\mu}_{\mathbb{Q}_1}\|_{\mathcal{H}_\kappa}^2}{\sqrt{m}\text{MMD}(\mathbb{P}_1,\mathbb{Q}_1,\kappa) + \sigma_M\mathcal{N}_{1-\alpha}} \in (0,1/2) \ .$$

**Table 1:** Comparisons of test power (mean±std) on DCT with respect to different total variation values $\epsilon'$, and the bold denotes the highest mean between our NAMMD test and Canonne's test. The experiments are conducted on discrete distributions defined over the same support set.

| Dataset | $\epsilon' = 0.1$ | | $\epsilon' = 0.3$ | | $\epsilon' = 0.5$ | | $\epsilon' = 0.7$ | |
|---------|-----------|-------|-----------|-------|-----------|-------|-----------|-------|
| | Canonne's | NAMMD | Canonne's | NAMMD | Canonne's | NAMMD | Canonne's | NAMMD |
| blob | .856±.023 | **.968±.022** | .809±.014 | **.912±.053** | .944±.013 | **.960±.020** | **.998±.002** | .961±.029 |
| higgs | .883±.015 | **.908±.050** | .825±.010 | **.947±.027** | .960±.005 | **.962±.023** | .994±.003 | **.995±.005** |
| hdgm | .861±.011 | **.942±.023** | .888±.016 | **.946±.017** | .937±.014 | **.965±.014** | .987±.004 | **.989±.004** |
| mnist | .715±.021 | **.931±.024** | .786±.026 | **.965±.007** | .896±.013 | **.997±.001** | .971±.008 | **1.00±.000** |
| cifar10 | .686±.030 | **.919±.017** | .751±.021 | **.923±.021** | .917±.006 | **.997±.002** | .981±.004 | **.999±.001** |
| Average | .800±.020 | **.934±.027** | .812±.017 | **.939±.025** | .931±.010 | **.976±.012** | .986±.004 | **.989±.008** |

*Furthermore, the following relation holds with probability $\varsigma \geq 1/65$ over samples $X$ and $Y$,*

$$\sqrt{m}\widehat{\mathrm{MMD}}(X, Y, \kappa) \leq \tau_\alpha^M \quad yet \quad \sqrt{m}\widehat{\mathrm{NAMMD}}(X, Y, \kappa) > \tau_\alpha^N ,$$

*if $C_1 \leq m \leq C_2$, where $C_1$ and $C_2$ are dependent on distributions $\mathbb{P}$ and $\mathbb{Q}$, and probability $\varsigma$.*

This theorem shows that, under the same kernel, if MMD test rejects null hypothesis correctly, our NAMMD test also rejects null hypothesis with high probability. Furthermore, we present that our NAMMD test can correctly reject null hypothesis even in cases where the original MMD test fails to do so. While the theoretical analysis is asymptotic, we complement it with empirical results in Section 5, which provide supporting evidence for the practical benefits of NAMMD. In Appendix C.6.2, we further provide detailed explanations regarding the condition $\|\boldsymbol{\mu}_{\mathbb{P}_1}\|_{\mathcal{H}_\kappa}^2 + \|\boldsymbol{\mu}_{\mathbb{Q}_1}\|_{\mathcal{H}_\kappa}^2 < \|\boldsymbol{\mu}_{\mathbb{P}_2}\|_{\mathcal{H}_\kappa}^2 + \|\boldsymbol{\mu}_{\mathbb{Q}_2}\|_{\mathcal{H}_\kappa}^2$ and the constants $C_1$ and $C_2$ in Theorem 9.

Although Theorem 9 is based on the same kernel for both NAMMD-based DCT and MMD-based DCT, it can be also useful to analyze the test power of NAMMD-based DCT and MMD-based DCT when they choose their corresponding optimal kernels. The key insight is that, for the (unknown) optimal kernel of MMD-based DCT $\kappa_*^M$, the NAMMD-based DCT with $\kappa_*^M$ performs better than MMD-based DCT with $\kappa_*^M$. Thus, the NAMMD-based DCT with its (unknown) optimal kernel $\kappa_*^N$ also performs better than MMD-based DCT with $\kappa_*^M$.

## 5 EXPERIMENTS

We perform DCT and TST on five benchmark datasets used by previous hypothesis testing studies [26, 29]. Specifically, "blob" and "hdgm" are synthetic Gaussian mixtures with dimensions 2 and 10. The "higgs" are tabular dataset consisting of the 4 dimension $\phi$-momenta distributions of Higgs-producing and background processes. "mnist" and "cifar" are image datasets consisting of original and generative images. We also conduct experiments on practical tasks related to domain adaptation using ImageNet and its variants, and evaluating adversarial perturbations on CIFAR10. More experiments, including **type-I error** for both DCT and TST, can be found in Appendix E.

### 5.1 EXPERIMENTS ON BENCHMARK DATASETS

**First,** we compare the test power of DCTs using our NAMMD and the statistic based on total variation introduced by Canonne et al. [48], and the experiments are conducted on *discrete distributions with the same support set containing only finite elements*. For each dataset, we randomly draw 50 elements $Z = \{z_1, z_2, ..., z_{50}\}$, and denote by $\mathbb{P}_{50}$ the uniform distribution over domain $Z$. We further construct distributions $\mathbb{Q}_{50}$ and $\mathbb{Q}_{50}^A$ for null and alternative hypotheses respectively, which satisfies $\mathrm{TV}(\mathbb{P}_{50}, \mathbb{Q}_{50}) = \epsilon'$ and $\mathrm{TV}(\mathbb{P}_{50}, \mathbb{Q}_{50}^A) = \epsilon' + 0.2$ (Details are provided in Appendix D.2). In experiments, we draw two i.i.d samples from $\mathbb{P}_{50}$ and $\mathbb{Q}_{50}^A$ to evaluate if the distance between $\mathbb{P}_{50}$ and $\mathbb{Q}_{50}^A$ is larger than that between $\mathbb{P}_{50}$ and $\mathbb{Q}_{50}$, i.e, $\epsilon'$. Table 1 summarizes the average test powers and standard deviations of NAMMD-based DCT and Canonne's DCT (Appendix D.2) based on total variaton. For comparison, we set $\epsilon' \in \{0.1, 0.3, 0.5, 0.7\}$[4]. From Table 1, NAMMD-based DCT generally performs better than Canonne's DCT, except on 2-dimensional blob dataset with $\epsilon' = 0.7$,

---

[4]Notably, although $\epsilon'$ increases, the difference between the two total variation values, namely the ground-truth total variation between $\mathbb{P}_{50}$ and $\mathbb{Q}_{50}^A$ minus that between $\mathbb{P}_{50}$ and $\mathbb{Q}_{50}$, remains fixed at 0.2.

**Figure 2:** The comparisons of test power vs sample size for our NAMMDFuse and SOTA two-sample tests.

**Table 2:** Comparisons of test power (mean±std) on distribution closeness testing with respect to different NAMMD values, and the bold denotes the highest mean between tests with our NAMMD and original MMD. Notably, the same selected kernel is applied for both NAMMD and MMD in this table. The experiments are not limited to discrete distributions defined over the same support set, which is different from those in Table 1.

| Dataset | $\epsilon = 0.1$ | | $\epsilon = 0.3$ | | $\epsilon = 0.5$ | | $\epsilon = 0.7$ | |
|---|---|---|---|---|---|---|---|---|
| | MMD | NAMMD | MMD | NAMMD | MMD | NAMMD | MMD | NAMMD |
| blob | .974±.009 | **.978**±.**008** | .890±.030 | **.923**±.**025** | .902±.032 | **.924**±.**021** | .909±.024 | **.933**±.**011** |
| higgs | .998±.002 | **.999**±.**001** | .938±.020 | **.965**±.**013** | .975±.012 | **.993**±.**003** | .978±.010 | **.996**±.**002** |
| hdgm | .980±.007 | **.984**±.**007** | .883±.027 | **.921**±.**021** | .901±.025 | **.941**±.**013** | **1.00**±.**000** | **1.00**±.**000** |
| mnist | **.982**±.**004** | **.982**±.**004** | .961±.006 | **.974**±.**004** | .946±.014 | **.983**±.**005** | .962±.010 | **.991**±.**003** |
| cifar10 | .932±.007 | **.938**±.**007** | .968±.019 | **.994**±.**003** | .898±.054 | **.912**±.**041** | **1.00**±.**000** | **1.00**±.**000** |
| Average | .973±.006 | **.976**±.**005** | .928±.020 | **.955**±.**013** | .924±.027 | **.951**±.**017** | .970±.009 | **.984**±.**003** |

where Canonne's DCT has lower variance and captures fine-grained distributional difference. See Appendix E.1 for NAMMD versus Wasserstein experimental results under the same setting.

**Second,** we compare NAMMD with more baselines (Appendix D.4) on TST, include: 1) MMDFuse [32]; 2) MMD-D [26]; 3) MMDAgg [31]; 4) AutoTST [49]; 5) $ME_{MaBiD}$ [29]; 6) ACTT [50]. Although we discuss NAMMD with a fixed kernel in this paper, it is compatible with various kernel selection frameworks as MMD. To illustrate this, we adapt NAMMD with multiple kernels using the fusion method [32] and refer to it as NAMMDFuse (Appendix D.5). From Figure 2, it is observed that NAMMDFuse achieves test power that is either higher or comparable to other methods. Besides the multiple kernel scheme, we also empirically demonstrate that NAMMD can be applied with various kernels (Gaussian, Laplace, Mahalanobis, and deep kernels) and achieves better performance than MMD under the same kernel, as shown in Table 13 (Appendix E).

**Third,** to compare our NAMMD and original MMD in DCT, we first *select the kernel $\kappa$* based on the original distribution pair $(\mathbb{P}, \mathbb{Q})$ of the dataset, following the TST approach [26]. Based on the selected kernel $\kappa$ and following the setup in Definition 8, we construct two pairs of distributions: $\mathbb{P}_1$ and $\mathbb{Q}_1$, and $\mathbb{P}_2$ and $\mathbb{Q}_2$, where $NAMMD(\mathbb{P}_1, \mathbb{Q}_1; \kappa) = \epsilon$ and $NAMMD(\mathbb{Q}_2, \mathbb{P}_2; \kappa) = \epsilon + 0.01$, and $MMD(\mathbb{P}_1, \mathbb{Q}_1; \kappa) < MMD(\mathbb{Q}_2, \mathbb{P}_2; \kappa)$. The details of construction are provided in Appendix D.3.

For comparison, we set $\epsilon \in \{0.1, 0.3, 0.5, 0.7\}$ . We randomly draw two samples from $\mathbb{Q}_2$ and $\mathbb{P}_2$ evaluate if distance between $\mathbb{P}_2$ and $\mathbb{Q}_2$ is larger than that between $\mathbb{P}_1$ and $\mathbb{Q}_1$. Table 2 summarizes the average test powers and standard deviations of our NAMMD distance and original MMD distance in DCT for *distributions over different domains*. It is evident that our NAMMD test achieves better performances than the original MMD test with respect to different datasets, and this improvement is achieved through scaling with the norms of mean embeddings of distributions according to Theorem 9.

## 5.2 Performing DCT in Practical Tasks

We present three practical case studies demonstrating the effectiveness of NAMMD-based DCT test. First, given the pre-trained ResNet50 that performs well on ImageNet, we wish to evaluate its performance on variants of ImageNet. A natural metric is accuracy margin (Eqn. 13 in Appendix D.7), defined as the difference in model accuracy between ImageNet and its variant, where a smaller margin indicates more comparable performance. For variants {ImageNetsk, ImageNetr, ImageNetv2, ImageNeta}, we compute their accuracy margins as {0.529,0.564,0.751,0.827} with true labels.

However, obtaining ground truth labels for ImageNet variants is often challenging or expensive. In such cases, we demonstrate that model performance can be assessed using NAMMD-based DCT

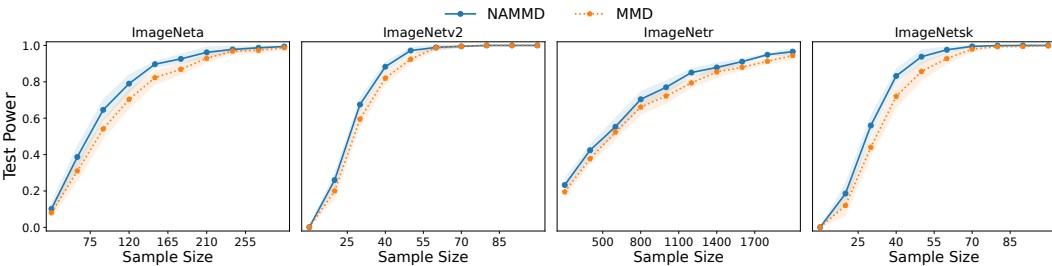

**Figure 3:** Comparisons in distinguishing the closeness levels between the original and variants of ImageNet.

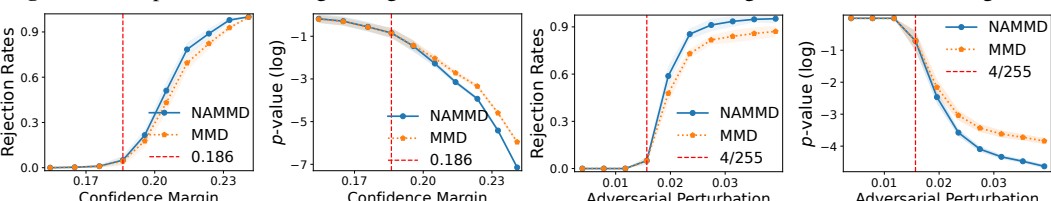

**Figure 4:** Comparison of NAMMD-based DCT and MMD-based DCT in detecting the confidence margin between ImageNet and ImageNetv2 datasets.

**Figure 5:** Comparison of the performance of NAMMD-based DCT and MMD-based DCT in detecting adversarial perturbations on the cifar10 dataset.

without labels. Following Definition 8, we set ImageNet as $\mathbb{P}_1$ and $\mathbb{P}_2$, and sequentially set each of its variants (ImageNeta, ImageNetv2, ImageNetr, and ImageNetsk) as $\mathbb{Q}_2$. Meanwhile, we sequentially set each of the variants (ImageNetv2, ImageNetr, ImageNetsk, slightly perturbed ImageNet) as $\mathbb{Q}_1$, and performs DCT. Figure 3 shows that NAMMD-based DCT achieves higher test power than MMD-based DCT, and effectively reflects the closeness relationships indicated by accuracy margin with limited sample size (much smaller than that of ImageNet and its variants). In the testing procedure, the ground truth about whether $\mathbb{Q}_2$ is actually closer or farther than $\mathbb{Q}_1$ is not known in advance. While Figure 3 reports results for cases where $\mathbb{Q}_2$ is indeed farther, the complementary case, where $\mathbb{Q}_2$ is not farther, is shown in the type I error results in Table 9 to Table 12 (Appendix E.4).

For datasets with limited samples, accuracy margin may be dispersed and fail to reliably capture differences in model performance. Instead, a natural metric is the confidence margin (Eqn. 12 in Appendix D.7), which measures the absolute difference in the model's expected prediction confidence between two distributions and a smaller margin indicate similar model performance. We also validate that our NAMMD reflects the same closeness relationships as confidence margin. We compute confidence margin for each class individually between ImageNet and ImageNetv2. Following Definition 8, we define the classes with average margin $0.186$ in ImageNet and ImageNetv2 as $\mathbb{P}_1$ and $\mathbb{Q}_1$. We further set $\mathbb{P}_2$ and $\mathbb{Q}_2$ as the classes in ImageNet and ImageNetv2 with margins in $\{0.154, 0.165, 0.176, 0.186, 0.196, 0.205, 0.214, 0.224, 0.233, 0.241\}$. We test with sample size 150 and present the rejection rates and $p$-values in Figure 4. For margins up to $0.186$ (left side of red line), rejection rates (type-I errors) are bounded given $\alpha = 0.05$. Conversely, for margins exceed $0.186$ (right side of red line), our NAMMD achieves higher rejection rates (test powers) and lower $p$-values.

Similarly, we validate that our NAMMD can be used to assess the level of adversarial perturbation over the cifar10 dataset. Using ResNet18 as the base model, we apply the PGD attack [51] with perturbations $\{i/255\}_{i=1}^{[10]}$. As expected, a larger perturbation generally result in poor model performance on the perturbed cifar10 dataset, indicating that the perturbed cifar10 is farther from the original cifar10. Following Definition 8, we define the original cifar10 as $\mathbb{P}_1 = \mathbb{P}_2$ and the cifar10 dataset with $4/255$ perturbation as $\mathbb{Q}_1$. We further set $\mathbb{Q}_2$ as the cifar10 after applying perturbations $\{i/255\}_{i=1}^{[10]}$, and perform testing with sample size 1500. It is evident that our NAMMD performs better than MMD and effectively assesses the levels of adversarial perturbations, as shown in Figure 5.

## 6 CONCLUSION

This work introduces new kernel-based distribution closeness and two-sample testing by proposing the *norm-adaptive MMD* (NAMMD) distance, which mitigate the issue that MMD value can be the same for multiple distribution pairs with different RKHS norms. An intriguing future research direction is to selecting an optimal global kernel for distribution closeness testing.

ETHICS STATEMENT

We confirm that this study adheres to the ICLR Code of Ethics. This research does not involve human subjects, and all datasets used are publicly available, ensuring compliance with privacy and security regulations. We have taken necessary precautions to avoid any potentially harmful insights or applications that may arise from our methodologies. Additionally, there are no potential conflicts of interest or sponsorships that could bias the outcomes of this work. This research complies with all relevant legal, ethical, and research integrity guidelines.

REPRODUCIBILITY STATEMENT

All assumptions and full proofs of our theoretical results are provided in the appendix (see Appendix C), with key lemmas and theorem statements in the main text. For experiments, we document datasets, preprocessing, and evaluation protocols (Appendix D). We release anonymized code and configuration files as supplementary material. Our reported numbers are the mean ± std over multiple runs, and we specify any deviations from default settings where applicable.

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

# Appendix

## A   NOTATIONS

In this section, we summarize important notations in Tables 3 and 4.

**Table 3:** Notation (Part 1)

| Symbol | Description |
|---|---|
| **• Basic Notations in Setting** | |
| $\mathcal{X} \subseteq \mathbb{R}^d$ | Instance space / domain of data |
| $\mathbb{P}, \mathbb{Q}, \mathbb{P}_1, \mathbb{Q}_1, \mathbb{P}_2, \mathbb{Q}_2$ | Borel probability measures on $\mathcal{X}$ |
| $\mathbb{P}_n, \mathbb{Q}_n$ | Discrete distributions over domain $Z = \{\boldsymbol{z}_1, \boldsymbol{z}_2, ..., \boldsymbol{z}_n\} \subseteq \mathbb{R}^d$ |
| $\kappa : \mathcal{X} \times \mathcal{X} \to \mathbb{R}$ | Positive-definite kernel, with $0 \leq \kappa(\boldsymbol{x}, \boldsymbol{x}') \leq K$ for any $\boldsymbol{x}, \boldsymbol{x}' \in \mathcal{X}$ |
| $\mathcal{H}_\kappa$ | Reproducing kernel Hilbert space (RKHS) associated to $\kappa$ |
| $\|\cdot\|_{\mathcal{H}_\kappa}$ | Norm in the RKHS $\mathcal{H}_\kappa$ |
| $\boldsymbol{\mu}_\mathbb{P}, \boldsymbol{\mu}_\mathbb{Q}$ | Kernel mean embeddings of $\mathbb{P}$ and $\mathbb{Q}$ |
| $\sigma^2_{\mathbb{P},\mathbb{Q}}$ | Asymptotic variance of $\sqrt{m}\,\widehat{\mathrm{NAMMD}}(X, Y; \kappa)$ under $\mathbb{P}, \mathbb{Q}$ |
| $\sigma^2_M$ | Asymptotic variance of $\sqrt{m}\,\widehat{\mathrm{MMD}}(X, Y; \kappa)$ under $\mathbb{P}, \mathbb{Q}$ or $\mathbb{P}_1, \mathbb{Q}_1$ |
| $\epsilon$ | Closeness parameter in NAMMD-based DCT with $\boldsymbol{H}_0$ and $\boldsymbol{H}_1$ |
| $\mathcal{N}$ | The standard normal distribution $\mathcal{N}(0, 1)$ |
| $\mathcal{N}_{1-\alpha}$ | The $(1-\alpha)$-quantile of $\mathcal{N}$ |
| $B$ | The iteration number of permutation test in TST |
| **• Distances** | |
| $\mathrm{TV}(\mathbb{P}_n, \mathbb{Q}_n)$ | Total variation distance between $\mathbb{P}_n$ and $\mathbb{Q}_n$ |
| $\mathrm{MMD}(\mathbb{P}, \mathbb{Q}; \kappa)$ | MMD distance between $\mathbb{P}$ and $\mathbb{Q}$ |
| $\mathrm{NAMMD}(\mathbb{P}, \mathbb{Q}; \kappa)$ | NAMMD distance between $\mathbb{P}$ and $\mathbb{Q}$ |
| **• Hypotheses** | |
| $\boldsymbol{H}_0, \boldsymbol{H}_1$ | Null and alternative hypotheses of NAMMD-based DCT with a given $\epsilon$ |
| $\boldsymbol{H}_0^N, \boldsymbol{H}_1^N$ | Hypotheses of MMD-based DCT with $\epsilon^N = \mathrm{NAMMD}(\mathbb{P}_1, \mathbb{Q}_1; \kappa)$ |
| $\boldsymbol{H}_0^M, \boldsymbol{H}_1^M$ | Hypotheses of MMD-based DCT with $\epsilon^M = \mathrm{MMD}(\mathbb{P}_1, \mathbb{Q}_1; \kappa)$ |
| $\boldsymbol{H}_0', \boldsymbol{H}_1'$ | Null and alternative hypotheses of TST |
| **• Estimations** | |
| $m$ | Sample size |
| $X, Y$ | Two independent samples of size $m$ from $\mathbb{P}, \mathbb{Q}$ or $\mathbb{P}_2, \mathbb{Q}_2$ |
| $X_{\boldsymbol{\pi}}, Y_{\boldsymbol{\pi}}$ | Permuted two samples |
| $H_{ij}$ | Pairwise function used in the NAMMD estimator |
| $\hat{\sigma}_{X,Y}$ | Plug-in estimator of $\sigma_{\mathbb{P},\mathbb{Q}}$ via $U$-statistics |
| $\widehat{\tau}_\alpha$ | Threshold of NAMMD-based DCT from asymptotic normal estimation |
| $\widehat{\tau}'_\alpha$ | Testing threshold of NAMMD-based TST from permutation test |
| $h(X, Y; \kappa)$ | Decision rule of the NAMMD-based DCT |
| $h'(X, Y; \kappa)$ | Decision rule of the NAMMD-based TST |
| $\widehat{\mathrm{NAMMD}}(X, Y, \kappa)$ | Estimator of $\mathrm{NAMMD}(\mathbb{P}, \mathbb{Q}; \kappa)$ or $\mathrm{NAMMD}(\mathbb{P}_2, \mathbb{Q}_2; \kappa)$ |
| $\widehat{\mathrm{NAMMD}}(X_{\boldsymbol{\pi}}, Y_{\boldsymbol{\pi}}, \kappa)$ | Estimator of NAMMD distance based on permuted samples $X_{\boldsymbol{\pi}}$ and $Y_{\boldsymbol{\pi}}$ |
| $\widehat{\mathrm{MMD}}(X, Y, \kappa)$ | Estimator of $\mathrm{MMD}(\mathbb{P}, \mathbb{Q}; \kappa)$ or $\mathrm{MMD}(\mathbb{P}_2, \mathbb{Q}_2; \kappa)$ |

**Table 4:** Notation (Part 2)

| Symbol | Description |
| --- | --- |
| **• Key Elements in Theoretical Results** | |
| $\tau_\alpha^M$ | Asymptotic $(1 - \alpha)$-quantile of the distribution of the MMD estimator $\sqrt{m}\widehat{\text{MMD}}(X, Y; \kappa)$ under $\mathbb{P}_1$ and $\mathbb{Q}_1$, used in Theorem 9 |
| $\tau_\alpha^N$ | Asymptotic $(1 - \alpha)$-quantile of the distribution of the NAMMD estimator $\sqrt{m}\widehat{\text{NAMMD}}(X, Y; \kappa)$ under $\mathbb{P}_1$ and $\mathbb{Q}_1$, used in Theorem 9 |
| $\epsilon^M$ | Closeness parameter for MMD-based DCT with hypotheses $\boldsymbol{H}_0^M$ and $\boldsymbol{H}_1^M$, defined as $\epsilon^M = \text{MMD}(\mathbb{P}_1, \mathbb{Q}_1; \kappa)$ and used in Theorem 9 |
| $\epsilon^N$ | Closeness parameter for NAMMD-based DCT with hypotheses $\boldsymbol{H}_0^N$ and $\boldsymbol{H}_1^N$, defined as $\epsilon^N = \text{NAMMD}(\mathbb{P}_1, \mathbb{Q}_1; \kappa)$ and used in Theorem 9 |
| $C_1, C_2$ | Constants that bound the sample complexity in Theorem 9 |

# B  FURTHER DETAILS ON NAMMD AND THE NAMMD-BASED TEST

## B.1  CONDITIONS UNDER WHICH NAMMD APPROACHES TO 1

Recall that the NAMMD is defined as:

$$
\begin{aligned}
\mathrm{NAMMD}(\mathbb{P}, \mathbb{Q}; \kappa) &= \frac{\|\boldsymbol{\mu}_{\mathbb{P}} - \boldsymbol{\mu}_{\mathbb{Q}}\|_{\mathcal{H}_\kappa}^2}{4K - \|\boldsymbol{\mu}_{\mathbb{P}}\|_{\mathcal{H}_\kappa}^2 - \|\boldsymbol{\mu}_{\mathbb{Q}}\|_{\mathcal{H}_\kappa}^2} \\
&= \frac{\|\boldsymbol{\mu}_{\mathbb{P}}\|_{\mathcal{H}_\kappa}^2 + \|\boldsymbol{\mu}_{\mathbb{Q}}\|_{\mathcal{H}_\kappa}^2 - 2\langle \boldsymbol{\mu}_{\mathbb{P}}, \boldsymbol{\mu}_{\mathbb{Q}} \rangle_{\mathcal{H}_\kappa}}{4K - \|\boldsymbol{\mu}_{\mathbb{P}}\|_{\mathcal{H}_\kappa}^2 - \|\boldsymbol{\mu}_{\mathbb{Q}}\|_{\mathcal{H}_\kappa}^2} \\
&= \frac{E_{\boldsymbol{x}, \boldsymbol{x}' \sim \mathbb{P}^2}[\kappa(\boldsymbol{x}, \boldsymbol{x}')] + E_{\boldsymbol{y}, \boldsymbol{y}' \sim \mathbb{Q}^2}[\kappa(\boldsymbol{y}, \boldsymbol{y}')] - 2E_{\boldsymbol{x} \sim \mathbb{P}, \boldsymbol{y} \sim \mathbb{Q}}[\kappa(\boldsymbol{x}, \boldsymbol{y}')]}{4K - E_{\boldsymbol{x}, \boldsymbol{x}' \sim \mathbb{P}^2}[\kappa(\boldsymbol{x}, \boldsymbol{x}')] - E_{\boldsymbol{y}, \boldsymbol{y}' \sim \mathbb{Q}^2}[\kappa(\boldsymbol{y}, \boldsymbol{y}')]} \,,
\end{aligned}
$$

where the kernel $\kappa(\boldsymbol{x}, \boldsymbol{x}') = \Psi(\boldsymbol{x} - \boldsymbol{x}')$ is positive-definite with $\Psi(\boldsymbol{0}) = K$ and $\Psi(\boldsymbol{x} - \boldsymbol{x}') \le K$ for all $\boldsymbol{x}, \boldsymbol{x}'$, and $K > 0$.

The value $\mathrm{NAMMD}(\mathbb{P}, \mathbb{Q}; \kappa) \to 1$ (i.e., maximum) is attained when:

- $\|\boldsymbol{\mu}_{\mathbb{P}}\|_{\mathcal{H}_\kappa}^2 = \|\boldsymbol{\mu}_{\mathbb{Q}}\|_{\mathcal{H}_\kappa}^2 = K$,
- $\langle \boldsymbol{\mu}_{\mathbb{P}}, \boldsymbol{\mu}_{\mathbb{Q}} \rangle_{\mathcal{H}_\kappa} \to 0$ (which essentially indicates that the two distributions have disjoint support).

Here, as an example, we consider two Dirac distributions $P$ and $Q$ over distinct supports $\boldsymbol{z}$ and $\boldsymbol{w}$, respectively, and use a Gaussian kernel with parameter $\eta$. In this case:

$$
\|\boldsymbol{\mu}_{\mathbb{P}}\|_{\mathcal{H}_\kappa}^2 = \|\boldsymbol{\mu}_{\mathbb{Q}}\|_{\mathcal{H}_\kappa}^2 = \Psi(\boldsymbol{0}) = K, \quad \text{and} \quad \langle \boldsymbol{\mu}_{\mathbb{P}}, \boldsymbol{\mu}_{\mathbb{Q}} \rangle_{\mathcal{H}_\kappa} = \Psi(\boldsymbol{x} - \boldsymbol{y}) = \exp(-\|\boldsymbol{x} - \boldsymbol{y}\|_2^2 / \eta^2) \,.
$$

As $\eta \to 0$, $\Psi(\boldsymbol{x} - \boldsymbol{y}) \to 0$, causing $\mathrm{NAMMD}(\mathbb{P}, \mathbb{Q}; \kappa) \to 1$.

We also present an empirical example for illustration. Specifically, we consider two Gaussian distributions $\mathbb{P} = \mathcal{N}(-1000, \sigma^2)$ and $\mathbb{Q} = \mathcal{N}(1000, \sigma^2)$, and compute NAMMD using a Gaussian kernel with bandwidth 1. When $\sigma$ is small, the distributions are both sharply concentrated around their respective means and have negligible overlap, effectively resulting in near-disjoint support. This setting closely approximates the idealized condition for maximizing $\mathrm{NAMMD}(\mathbb{P}, \mathbb{Q}; \kappa)$. In the following experiment, we compare the value of $\mathrm{NAMMD}(\mathbb{P}, \mathbb{Q}; \kappa)$ under varying $\sigma$ to empirically verify this behavior.

**Table 5:** Comparison of NAMMD and MMD across $\sigma$.

| $\sigma$ | $10^0$ | $10^{-1}$ | $10^{-2}$ | $10^{-3}$ | |
|---|---|---|---|---|---|
| NAMMD | 0.2679 | $1 - 4.5 \times 10^{-2}$ | $1 - 9.9 \times 10^{-5}$ | $1 - 2.1 \times 10^{-7}$ | |
| $\sigma$ | $10^{-4}$ | $10^{-5}$ | $10^{-6}$ | $10^{-7}$ | $10^{-8}$ |
| NAMMD | $1 - 2.1 \times 10^{-8}$ | $1 - 6.4 \times 10^{-10}$ | $1 - 7.0 \times 10^{-12}$ | $1 - 1.1 \times 10^{-16}$ | 1 |

When $\sigma = 10^{-8}$, the kernel value $\kappa(\boldsymbol{x}, \boldsymbol{x}')$ is close to 1 when $\boldsymbol{x}$ and $\boldsymbol{x}'$ are drawn from the same distribution, and close to 0 when they are drawn from different distributions. Consequently, $\mathrm{NAMMD}(\mathbb{P}, \mathbb{Q}; \kappa)$ approach its maximum value 1.

## B.2  EXTENSION TO UNEQUAL SAMPLE SIZES

Recall that

$$
\mathrm{NAMMD}(\mathbb{P}, \mathbb{Q}; \kappa) = \frac{\mathrm{MMD}(\mathbb{P}, \mathbb{Q}; \kappa)}{4K - \|\boldsymbol{\mu}_{\mathbb{P}}\|_{\mathcal{H}_\kappa}^2 - \|\boldsymbol{\mu}_{\mathbb{Q}}\|_{\mathcal{H}_\kappa}^2} \,.
$$

To estimate $\mathrm{NAMMD}(\mathbb{P}, \mathbb{Q}; \kappa)$ from two samples of unequal sizes,

$$
X = \{\boldsymbol{x}_i\}_{i=1}^m \sim \mathbb{P}^m \quad \text{and} \quad Y = \{\boldsymbol{y}_j\}_{j=1}^n \sim \mathbb{Q}^n \,,
$$

We analyze the behavior of NAMMD estimator by examining its numerator, corresponding to the MMD statistic, and its denominator, which depends on the RKHS norms of $\mathbb{P}$ and $\mathbb{Q}$, separately.

The numerator, $\mathrm{MMD}(\mathbb{P}, \mathbb{Q}; \kappa)$, can be estimated using a $U$-statistic. When moving from equal to unequal sample sizes, the estimator changes from a one-sample $U$-statistic to a two-sample statistic as follows

$$U_{m,n} = \frac{1}{\binom{m}{2}\binom{n}{2}} \sum_{1 \leq i < i' \leq m} \sum_{1 \leq j < j' \leq n} h(\boldsymbol{x}_i, \boldsymbol{x}_{i'}; \ \boldsymbol{y}_j, \boldsymbol{y}_{j'}) \,,$$

where

$$h(\boldsymbol{x}_1, \boldsymbol{x}_2; \ \boldsymbol{y}_1, \boldsymbol{y}_2) = \kappa(\boldsymbol{x}_1, \boldsymbol{x}_2) + \kappa(\boldsymbol{y}_1, \boldsymbol{y}_2) - \kappa(\boldsymbol{x}_1, \boldsymbol{y}_2) - \kappa(\boldsymbol{x}_2, \boldsymbol{y}_1).$$

Despite this modification, both the equal-sample and unequal-sample versions exhibit similar asymptotic properties [52]. In particular, when $\mathrm{MMD}(\mathbb{P}, \mathbb{Q}; \kappa) = 0$, the statistic converges in distribution to an (often infinite) weighted sum of $\chi^2$ random variables, where the weights are given by the eigenvalues of the covariance operator on $\mathcal{H}_\kappa \to \mathcal{H}_\kappa$.

On the other hand, the estimator of the denominator term

$$4K - \frac{1}{m(m-1)} \sum_{i \neq i'}^{m} \kappa(\boldsymbol{x}_i, \boldsymbol{x}_{i'}) - \frac{1}{n(n-1)} \sum_{j \neq j'}^{n} \kappa(\boldsymbol{y}_j, \boldsymbol{y}_{j'}) \,,$$

remains unchanged regardless of whether the sample sizes are equal or unequal, since the RKHS norms $\|\boldsymbol{\mu}_\mathbb{P}\|^2_{\mathcal{H}_\kappa}$ and $\|\boldsymbol{\mu}_\mathbb{Q}\|^2_{\mathcal{H}_\kappa}$ can be estimated independently from each sample.

### B.3 DETAILS OF VARIANCE ESTIMATOR

We adhere to the results of empirical variance estimators provided by Sutherland [46]. For simplicity, we first introduce the uncentred covariance operator as follows:

$$C_X = E_{\boldsymbol{x} \sim \mathbb{P}}[\varphi(\boldsymbol{x}) \otimes \varphi(\boldsymbol{x})] \,,$$

where $\varphi(\cdot)$ is the feature map of the corresponding RKHS $\mathcal{H}_\kappa$.

For simplicity, we define the $m \times m$ matrix $\mathbf{K_{XY}}$ with $(\mathbf{K_{XY}})_{ij} = \kappa(\boldsymbol{x}_i, \boldsymbol{y}_j)$. Let $\tilde{\mathbf{K}}_{\mathbf{XY}}$ be $\mathbf{K_{XY}}$ with diagonals set to zero. In a similar manner, we have $\mathbf{K_{XX}}$ and $\mathbf{K_{YY}}$, and $\tilde{\mathbf{K}}_{\mathbf{XX}}$ and $\tilde{\mathbf{K}}_{\mathbf{YY}}$. Let $\mathbf{1}$ be the $m$-vector of all ones. Denote by $(m)_k := m(m-1)\cdots(m-k+1)$.

We have that

$$
\begin{aligned}
\zeta_1 ={}& \langle \boldsymbol{\mu}_X, C_X \boldsymbol{\mu}_X \rangle - \langle \boldsymbol{\mu}_X, \boldsymbol{\mu}_X \rangle^2 + \langle \boldsymbol{\mu}_Y, C_Y \boldsymbol{\mu}_Y \rangle - \langle \boldsymbol{\mu}_Y, \boldsymbol{\mu}_Y \rangle^2 \\
&+ \langle \boldsymbol{\mu}_Y, C_X \boldsymbol{\mu}_Y \rangle + \langle \boldsymbol{\mu}_X, C_Y \boldsymbol{\mu}_X \rangle - \langle \boldsymbol{\mu}_X, \boldsymbol{\mu}_Y \rangle^2 - \langle \boldsymbol{\mu}_Y, \boldsymbol{\mu}_X \rangle^2 \\
&- 2\langle \boldsymbol{\mu}_X, C_X \boldsymbol{\mu}_Y \rangle + 2\langle \boldsymbol{\mu}_X, \boldsymbol{\mu}_X \rangle \langle \boldsymbol{\mu}_X, \boldsymbol{\mu}_Y \rangle - 2\langle \boldsymbol{\mu}_Y, C_Y \boldsymbol{\mu}_X \rangle + 2\langle \boldsymbol{\mu}_Y, \boldsymbol{\mu}_Y \rangle \langle \boldsymbol{\mu}_X, \boldsymbol{\mu}_Y \rangle \\
\approx{}& \frac{1}{(m)_3}\left[\left\|\tilde{\mathbf{K}}_{\mathbf{XX}}\mathbf{1}\right\|^2 - \left\|\tilde{\mathbf{K}}_{\mathbf{XX}}\right\|_F^2\right] - \frac{1}{(m)_4}\left[\left(\mathbf{1}^\top \tilde{\mathbf{K}}_{\mathbf{XX}}\mathbf{1}\right)^2 - 4\left\|\tilde{\mathbf{K}}_{\mathbf{XX}}\mathbf{1}\right\|^2 + 2\left\|\tilde{\mathbf{K}}_{\mathbf{XX}}\right\|_F^2\right] \\
&+ \frac{1}{(m)_3}\left[\left\|\tilde{\mathbf{K}}_{\mathbf{YY}}\mathbf{1}\right\|^2 - \left\|\tilde{\mathbf{K}}_{\mathbf{YY}}\right\|_F^2\right] - \frac{1}{(m)_4}\left[\left(\mathbf{1}^\top \tilde{\mathbf{K}}_{\mathbf{YY}}\mathbf{1}\right)^2 - 4\left\|\tilde{\mathbf{K}}_{\mathbf{YY}}\mathbf{1}\right\|^2 + 2\left\|\tilde{\mathbf{K}}_{\mathbf{YY}}\right\|_F^2\right] \\
&+ \frac{1}{m^2(m-1)}\left[\left\|\mathbf{K_{XY}}\mathbf{1}\right\|^2 - \left\|\mathbf{K_{XY}}\right\|_F^2\right] + \frac{1}{m^2(m-1)}\left[\left\|\mathbf{K}_{\mathbf{XY}}^\top\mathbf{1}\right\|^2 - \left\|\mathbf{K_{XY}}\right\|_F^2\right] \\
&- \frac{2}{m^2(m-1)^2}\left[\left(\mathbf{1}^\top \mathbf{K_{XY}}\mathbf{1}\right)^2 - \left\|\mathbf{K}_{\mathbf{XY}}^\top\mathbf{1}\right\|^2 - \left\|\mathbf{K_{XY}}\mathbf{1}\right\|^2 + \left\|\mathbf{K_{XY}}\right\|_F^2\right] \\
&- \frac{2}{m^2(m-1)}\mathbf{1}^\top \tilde{\mathbf{K}}_{\mathbf{XX}}\mathbf{K_{XY}}\mathbf{1} + \frac{2}{m(m)_3}\left[\mathbf{1}^\top \tilde{\mathbf{K}}_{\mathbf{XX}}\mathbf{1}\mathbf{1}^\top \mathbf{K_{XY}}\mathbf{1} - 2\mathbf{1}^\top \tilde{\mathbf{K}}_{\mathbf{XX}}\mathbf{K_{XY}}\mathbf{1}\right] \\
&- \frac{2}{m^2(m-1)}\mathbf{1}^\top \tilde{\mathbf{K}}_{\mathbf{YY}}\mathbf{K}_{\mathbf{XY}}^\top\mathbf{1} + \frac{2}{m(m)_3}\left[\mathbf{1}^\top \tilde{\mathbf{K}}_{\mathbf{YY}}\mathbf{1}\mathbf{1}^\top \mathbf{K}_{\mathbf{XY}}^\top\mathbf{1} - 2\mathbf{1}^\top \tilde{\mathbf{K}}_{\mathbf{YY}}\mathbf{K}_{\mathbf{XY}}^\top\mathbf{1}\right]
\end{aligned}
$$

and

$$
\begin{aligned}
\zeta_2 &= \mathbb{E}\left[\kappa\left(\boldsymbol{x}_1, \boldsymbol{x}_2\right)^2\right] - \langle \boldsymbol{\mu}_X, \boldsymbol{\mu}_X \rangle^2 + \mathbb{E}\left[\kappa\left(\boldsymbol{y}_1, \boldsymbol{y}_2\right)^2\right] \\
&\quad - \langle \boldsymbol{\mu}_Y, \boldsymbol{\mu}_Y \rangle^2 + 2\mathbb{E}\left[\kappa(\boldsymbol{x}, \boldsymbol{y})^2\right] - 2\langle \boldsymbol{\mu}_X, \boldsymbol{\mu}_Y \rangle^2 \\
&\quad - 4\langle \boldsymbol{\mu}_X, C_X \boldsymbol{\mu}_Y \rangle + 4\langle \boldsymbol{\mu}_X, \boldsymbol{\mu}_X \rangle \langle \boldsymbol{\mu}_X, \boldsymbol{\mu}_Y \rangle - 4\langle \boldsymbol{\mu}_Y, C_Y \boldsymbol{\mu}_X \rangle + 4\langle \boldsymbol{\mu}_Y, \boldsymbol{\mu}_Y \rangle \langle \boldsymbol{\mu}_X, \boldsymbol{\mu}_Y \rangle \\
&\approx \frac{1}{m(m-1)}\left\|\tilde{\mathbf{K}}_{\mathbf{XX}}\right\|_F^2 - \frac{1}{(m)_4}\left[\left(\mathbf{1}^\top \tilde{\mathbf{K}}_{\mathbf{XX}} \mathbf{1}\right)^2 - 4\left\|\tilde{\mathbf{K}}_{\mathbf{XX}} \mathbf{1}\right\|^2 + 2\left\|\tilde{\mathbf{K}}_{\mathbf{XX}}\right\|_F^2\right] \\
&\quad + \frac{1}{m(m-1)}\left\|\tilde{\mathbf{K}}_{\mathbf{YY}}\right\|_F^2 - \frac{1}{(m)_4}\left[\left(\mathbf{1}^\top \tilde{\mathbf{K}}_{\mathbf{YY}} \mathbf{1}\right)^2 - 4\left\|\tilde{\mathbf{K}}_{\mathbf{YY}} \mathbf{1}\right\|^2 + 2\left\|\tilde{\mathbf{K}}_{\mathbf{YY}}\right\|_F^2\right] \\
&\quad + \frac{2}{m^2}\left\|\mathbf{K}_{\mathbf{XY}}\right\|_F^2 - \frac{2}{m^2(m-1)^2}\left[\left(\mathbf{1}^\top \mathbf{K}_{\mathbf{XY}} \mathbf{1}\right)^2 - \left\|\mathbf{K}_{\mathbf{XY}}^\top \mathbf{1}\right\|^2 - \left\|\mathbf{K}_{\mathbf{XY}} \mathbf{1}\right\|^2 + \left\|\mathbf{K}_{\mathbf{XY}}\right\|_F^2\right] \\
&\quad - \frac{4}{m^2(m-1)}\mathbf{1}^\top \tilde{\mathbf{K}}_{\mathbf{XX}} \mathbf{K}_{\mathbf{XY}} \mathbf{1} + \frac{4}{m(m)_3}\left[\mathbf{1}^\top \tilde{\mathbf{K}}_{\mathbf{XX}} \mathbf{1}\mathbf{1}^\top \mathbf{K}_{\mathbf{XY}} \mathbf{1} - 2\mathbf{1}^\top \tilde{\mathbf{K}}_{\mathbf{XX}} \mathbf{K}_{\mathbf{XY}} \mathbf{1}\right] \\
&\quad - \frac{4}{m^2(m-1)}\mathbf{1}^\top \tilde{\mathbf{K}}_{\mathbf{YY}} \mathbf{K}_{\mathbf{XY}}^\top \mathbf{1} + \frac{4}{m(m)_3}\left[\mathbf{1}^\top \tilde{\mathbf{K}}_{\mathbf{YY}} \mathbf{1}\mathbf{1}^\top \mathbf{K}_{\mathbf{XY}}^\top \mathbf{1} - 2\mathbf{1}^\top \tilde{\mathbf{K}}_{\mathbf{YY}} \mathbf{K}_{\mathbf{XY}}^\top \mathbf{1}\right] .
\end{aligned}
$$

where $\langle \cdot, \cdot \rangle$ denotes the inner product in RKHS $\mathcal{H}_\kappa$. Here, we denote by

$$
\boldsymbol{\mu}_X = \boldsymbol{\mu}_{\mathbb{P}} = E_{\boldsymbol{x} \sim \mathbb{P}}[\kappa(\cdot, \boldsymbol{x})] \quad \text{and} \quad \boldsymbol{\mu}_Y = \boldsymbol{\mu}_{\mathbb{Q}} = E_{\boldsymbol{y} \sim \mathbb{Q}}[\kappa(\cdot, \boldsymbol{y})] .
$$

**Convergence of the estimators.** Having established that the estimators are unbiased [46], we now prove their convergence by analyzing each constituent term separately with bounded kernel $\kappa(\cdot, \cdot) \leq K$, as follows.

- The term $\langle \boldsymbol{\mu}_X, C_X \boldsymbol{\mu}_X \rangle$ is estimated by

$$
A = \frac{1}{(n)_3} \sum_i \sum_{j \neq \ell} \sum_{\ell \notin \{i,j\}} \kappa(\boldsymbol{x}_i, \boldsymbol{x}_j)\kappa(\boldsymbol{x}_i, \boldsymbol{x}_\ell) .
$$

It is evident that

$$
|A - \langle \boldsymbol{\mu}_X, C_X \boldsymbol{\mu}_X \rangle| \leq |A - B| + |B - \langle \boldsymbol{\mu}_X, C_X \boldsymbol{\mu}_X \rangle| ,
$$

with

$$
\begin{aligned}
B &= \frac{1}{n}\sum_i E_{\boldsymbol{x}}[\kappa(\boldsymbol{x}_i, \boldsymbol{x})]E_{\boldsymbol{x}}[\kappa(\boldsymbol{x}_i, \boldsymbol{x})] \\
&= \frac{1}{n}\sum_i \langle \kappa(\boldsymbol{x}_i, \cdot), \boldsymbol{\mu}_X \rangle^2 .
\end{aligned}
$$

As we can see, $B$ is a $U$-statistic. By the large deviation bound (Theorem 11) for $U$-statistic, we have that

$$
B \xrightarrow{p} \langle \boldsymbol{\mu}_X, C_X \boldsymbol{\mu}_X \rangle .
$$

For the term $|A - B|$, we have that

$$
\begin{aligned}
&|A - B| \\
&= \frac{1}{n}\sum_i \left[\frac{1}{(n-1)(n-2)}\sum_{j \neq \ell}\sum_{\ell \notin \{i,j\}}\kappa(\boldsymbol{x}_i, \boldsymbol{x}_j)\kappa(\boldsymbol{x}_i, \boldsymbol{x}_\ell) - E_{\boldsymbol{x}}[\kappa(\boldsymbol{x}_i, \boldsymbol{x})]E_{\boldsymbol{x}}[\kappa(\boldsymbol{x}_i, \boldsymbol{x})]\right] ,
\end{aligned}
$$

where the term $\frac{1}{(n-1)(n-2)}\sum_{j \neq \ell}\sum_{\ell \notin \{i,j\}}\kappa(\boldsymbol{x}_i, \boldsymbol{x}_j)\kappa(\boldsymbol{x}_i, \boldsymbol{x}_\ell)$ can also be viewed as a $U$-statistic, and it follows that

$$
\frac{1}{(n-1)(n-2)}\sum_{j \neq \ell}\sum_{\ell \notin \{i,j\}}\kappa(\boldsymbol{x}_i, \boldsymbol{x}_j)\kappa(\boldsymbol{x}_i, \boldsymbol{x}_\ell) \xrightarrow{p} E_{\boldsymbol{x}}[\kappa(\boldsymbol{x}_i, \boldsymbol{x})]E_{\boldsymbol{x}}[\kappa(\boldsymbol{x}_i, \boldsymbol{x})] ,
$$

by the large deviation bound (Theorem 11) for $U$-statistic.

Combine these results, we have that

$$
\frac{1}{(n)_3}\sum_i \sum_{j \neq \ell}\sum_{\ell \notin \{i,j\}}\kappa(\boldsymbol{x}_i, \boldsymbol{x}_j)\kappa(\boldsymbol{x}_i, \boldsymbol{x}_\ell) \xrightarrow{p} \langle \boldsymbol{\mu}_X, C_X \boldsymbol{\mu}_X \rangle .
$$

- The term $\langle \boldsymbol{\mu}_X, \boldsymbol{\mu}_X \rangle^2$ is estimated by

$$A = \frac{1}{(n)_4} \sum_i \sum_{j \neq i} \kappa(\boldsymbol{x}_i, \boldsymbol{x}_j) \sum_{a \notin \{i,j\}} \sum_{b \notin \{i,j,a\}} \kappa(\boldsymbol{x}_a, \boldsymbol{x}_b) \, .$$

It is evident that

$$\left| A - \langle \boldsymbol{\mu}_X, \boldsymbol{\mu}_X \rangle^2 \right| \leq |A - B| + \left| B - \langle \boldsymbol{\mu}_X, \boldsymbol{\mu}_X \rangle^2 \right| \, ,$$

with

$$B = \frac{1}{n(n-1)} \sum_i \sum_{j \neq i} \kappa(\boldsymbol{x}_i, \boldsymbol{x}_j) E_{\boldsymbol{x}, \boldsymbol{x}'}[\kappa(\boldsymbol{x}, \boldsymbol{x}')] \, .$$

Building on this, we can prove that

$$\frac{1}{(n)_4} \sum_i \sum_{j \neq i} \kappa(\boldsymbol{x}_i, \boldsymbol{x}_j) \sum_{a \notin \{i,j\}} \sum_{b \notin \{i,j,a\}} \kappa(\boldsymbol{x}_a, \boldsymbol{x}_b) \xrightarrow{p} \langle \boldsymbol{\mu}_X, \boldsymbol{\mu}_X \rangle^2 \, ,$$

using a similar argument as in the convergence proof for the estimator of $\langle \boldsymbol{\mu}_X, C_X \boldsymbol{\mu}_X \rangle$.

- The term $\langle \boldsymbol{\mu}_Y, C_X \boldsymbol{\mu}_Y \rangle$ is estimated by

$$A = \frac{1}{n^2(n-1)} \sum_i \sum_j \sum_{\ell \neq j} \kappa(\boldsymbol{x}_i, \boldsymbol{y}_j) \kappa(\boldsymbol{x}_i, \boldsymbol{y}_\ell) \, .$$

It is evident that

$$|A - \langle \boldsymbol{\mu}_Y, C_X \boldsymbol{\mu}_Y \rangle| \leq |A - B| + |B - \langle \boldsymbol{\mu}_Y, C_X \boldsymbol{\mu}_Y \rangle| \, ,$$

with

$$B = \frac{1}{n} \sum_i E_{\boldsymbol{y}}[\kappa(\boldsymbol{x}_i, \boldsymbol{y})] E_{\boldsymbol{y}}[\kappa(\boldsymbol{x}_i, \boldsymbol{y})] \, .$$

Building on this, we can prove that

$$\frac{1}{n^2(n-1)} \sum_i \sum_j \sum_{\ell \neq j} \kappa(\boldsymbol{x}_i, \boldsymbol{y}_j) \kappa(\boldsymbol{x}_i, \boldsymbol{y}_\ell) \xrightarrow{p} \langle \boldsymbol{\mu}_Y, C_X \boldsymbol{\mu}_Y \rangle \, ,$$

using a similar argument as in the convergence proof for the estimator of $\langle \boldsymbol{\mu}_X, C_X \boldsymbol{\mu}_X \rangle$.

- The term $\langle \boldsymbol{\mu}_X, C_X \boldsymbol{\mu}_Y \rangle$ is estimated by

$$A = \frac{1}{n^2(n-1)} \sum_i \sum_{j \neq i} \sum_\ell \kappa(\boldsymbol{x}_i, \boldsymbol{x}_j) \kappa(\boldsymbol{x}_i, \boldsymbol{y}_\ell) \, .$$

It is evident that

$$|A - \langle \boldsymbol{\mu}_X, C_X \boldsymbol{\mu}_Y \rangle| \leq |A - B| + |B - \langle \boldsymbol{\mu}_X, C_X \boldsymbol{\mu}_Y \rangle| \, ,$$

with

$$B = \frac{1}{n} \sum_i E_{\boldsymbol{x}}[\kappa(\boldsymbol{x}_i, \boldsymbol{x})] E_{\boldsymbol{y}}[\kappa(\boldsymbol{x}_i, \boldsymbol{y})] \, .$$

Building on this, we can prove that

$$\frac{1}{n^2(n-1)} \sum_i \sum_{j \neq i} \sum_\ell \kappa(\boldsymbol{x}_i, \boldsymbol{x}_j) \kappa(\boldsymbol{x}_i, \boldsymbol{y}_\ell) \xrightarrow{p} \langle \boldsymbol{\mu}_X, C_X \boldsymbol{\mu}_Y \rangle \, ,$$

using a similar argument as in the convergence proof for the estimator of $\langle \boldsymbol{\mu}_X, C_X \boldsymbol{\mu}_X \rangle$.

- The term $\langle \boldsymbol{\mu}_X, \boldsymbol{\mu}_X \rangle \langle \boldsymbol{\mu}_X, \boldsymbol{\mu}_Y \rangle$ is estimated by

$$A = \frac{1}{n(n)_3} \sum_i \sum_{j \neq i} \kappa(\boldsymbol{x}_i, \boldsymbol{x}_j) \sum_{\ell \notin \{i,j\}} \sum_a \kappa(\boldsymbol{x}_\ell, \boldsymbol{y}_a) \, .$$

It is evident that

$$|A - \langle \boldsymbol{\mu}_X, \boldsymbol{\mu}_X \rangle \langle \boldsymbol{\mu}_X, \boldsymbol{\mu}_Y \rangle| \leq |A - B| + |B - \langle \boldsymbol{\mu}_X, \boldsymbol{\mu}_X \rangle \langle \boldsymbol{\mu}_X, \boldsymbol{\mu}_Y \rangle| \, ,$$

with

$$B = \frac{1}{n(n-1)} \sum_i \sum_{j \neq i} \kappa(\boldsymbol{x}_i, \boldsymbol{x}_j) E_{\boldsymbol{x}, \boldsymbol{y}}[\kappa(\boldsymbol{x}, \boldsymbol{y})] \,.$$

Building on this, we can prove that

$$\frac{1}{n(n)_3} \sum_i \sum_{j \neq i} \kappa(\boldsymbol{x}_i, \boldsymbol{x}_j) \sum_{\ell \notin \{i,j\}} \sum_a \kappa(\boldsymbol{x}_\ell, \boldsymbol{y}_a) \xrightarrow{p} \langle \boldsymbol{\mu}_X, \boldsymbol{\mu}_X \rangle \langle \boldsymbol{\mu}_X, \boldsymbol{\mu}_Y \rangle \,,$$

using a similar argument as in the convergence proof for the estimator of $\langle \boldsymbol{\mu}_X, C_X \boldsymbol{\mu}_X \rangle$.

- The term $\langle \boldsymbol{\mu}_X, \boldsymbol{\mu}_Y \rangle^2$ is estimated by

$$A = \frac{1}{n^2} \sum_{i,j} \kappa(\boldsymbol{x}_i, \boldsymbol{y}_j) \frac{1}{(n-1)^2} \sum_{i' \neq i} \sum_{j' \neq j} \kappa(\boldsymbol{x}_{i'}, \boldsymbol{y}_{j'}) \,,$$

It is evident that

$$\left| A - \langle \boldsymbol{\mu}_X, \boldsymbol{\mu}_Y \rangle^2 \right| \leq |A - B| + \left| B - \langle \boldsymbol{\mu}_X, \boldsymbol{\mu}_Y \rangle^2 \right| \,,$$

with

$$B = \frac{1}{n^2} \sum_i \sum_{i,j} \kappa(\boldsymbol{x}_i, \boldsymbol{y}_j) E_{\boldsymbol{x}, \boldsymbol{y}}[\kappa(\boldsymbol{x}, \boldsymbol{y})] \,.$$

Building on this, we can prove that

$$\frac{1}{n^2} \sum_{i,j} \kappa(\boldsymbol{x}_i, \boldsymbol{y}_j) \frac{1}{(n-1)^2} \sum_{i' \neq i} \sum_{j' \neq j} \kappa(\boldsymbol{x}_{i'}, \boldsymbol{y}_{j'}) \xrightarrow{p} \langle \boldsymbol{\mu}_X, \boldsymbol{\mu}_Y \rangle^2 \,,$$

using a similar argument as in the convergence proof for the estimator of $\langle \boldsymbol{\mu}_X, C_X \boldsymbol{\mu}_X \rangle$.

- The term $\mathbb{E}\left[\kappa(\boldsymbol{x}_1, \boldsymbol{x}_2)^2\right]$ is estimated by

$$\frac{1}{n(n-1)} \sum_{i \neq j} \kappa(\boldsymbol{x}_i, \boldsymbol{x}_j)^2 \,,$$

which can also be viewed as a $U$-statistic, and it follows that

$$\frac{1}{n(n-1)} \sum_{i \neq j} \kappa(\boldsymbol{x}_i, \boldsymbol{x}_j)^2 \xrightarrow{p} \mathbb{E}\left[\kappa(\boldsymbol{x}_1, \boldsymbol{x}_2)^2\right] \,,$$

by the large deviation bound (Theorem 11) for $U$-statistic.

Based on the convergence of each constituent term, it follows that the estimators of $\zeta_1$ and $\zeta_2$ converge in probability to their respective population quantities $\zeta_1$ and $\zeta_2$, by an application of the *continuous mapping theorem*.

### B.4 RELEVANT WORKS

A well-known class of two-sample testing constructs kernel embeddings for each distribution and then test the differences between these embeddings [53–56]. Another relevant approach assesses the differences between distributions with classification performance [57–63, 49, 64]. Kernel-based MMD has been one of the most important statistic for two-sample testing, which includes popular classifier-based two-sample testing approaches as a special case [26].

Previous distribution closeness testing approaches primarily focus on theoretical analysis of the sample complexity of sub-linear algorithms, and these approaches often rely on total variation over discrete one-dimensional distributions [15, 18–21]. Other measures of closeness also include $\ell_2$ distance [65–67], entropy [68], probability difference [11, 69], etc. In comparison, we turn to kernel methods that have shown effectiveness in non-parametric testing.

Permutation tests are widely used in statistics for testing equality of distributions, providing a finite-sample guarantee on the type-I error under the null hypothesis that assumes $\mathbb{P} = \mathbb{Q}$ [70–73]. For DCT with null hypothesis $\boldsymbol{H}_0 : \text{NAMMD}(\mathbb{P}, \mathbb{Q}; \kappa) \leq \epsilon$ and $\epsilon \in (0, 1)$, the empirical estimator of our

NAMMD distance, i.e., $\text{NAMMD}(\mathbb{P}, \mathbb{Q}; \kappa) = \epsilon$, has an asymptotic Gaussian distribution as shown in Lemma 2. Consequently, the testing threshold can be easily estimated as the $(1 - \alpha)$-quantile of this asymptotic Gaussian distribution, following [29, 55, 56].

Some approaches select kernels in a supervised manner using held-out data [28, 47], while others rely on unsupervised methods, such as the median heuristic [25], or adaptively combine multiple kernels [31, 32]. Our NAMMD is compatible with these methods; for instance, the kernel can be selected by maximizing the $t$-statistic for test power estimation derived from Lemma 2 (details are provided in Appendix B.5). However, these approaches are primarily designed for distinguishing between a fixed distribution pair in two-sample testing. It remains an open question and an important future work to select an optimal global kernel for distribution closeness testing with multiple distribution pairs.

## B.5 DETAILS OF OPTIMIZATION FOR KERNEL SELECTING

---
**Algorithm 1** Kernel Selection

---
**Input**: Two samples $X$ and $Y$, a kernel $\kappa$, step size $\eta$, iteration number $N$
**Output**: Two samples $X$ and $Y$
  1: **for** $\ell = 1, 2, \cdots, N$ **do**
  2:     Calculate the estimator $\widehat{\text{NAMMD}}(X, Y; \kappa)/\sigma_{X,Y}$ according to Eqn. 5
  3:     Calculate gradient $\nabla \cdot \left( \widehat{\text{NAMMD}}(X, Y; \kappa)/\sigma_{X,Y} \right)$
  4:     Gradient ascend with step size $\eta$ by the Adam method
  5: **end for**

---

Recall Lemma 2, if $\text{NAMMD}(\mathbb{P}, \mathbb{Q}; \kappa) = \epsilon$ with $\epsilon \in (0, 1)$, we have

$$\sqrt{m}(\widehat{\text{NAMMD}}(X, Y; \kappa) - \epsilon) \xrightarrow{d} \mathcal{N}(0, \sigma_{\mathbb{P},\mathbb{Q}}^2) \ ,$$

where $\sigma_{\mathbb{P},\mathbb{Q}} = \sqrt{4E[H_{1,2}H_{1,3}] - 4(E[H_{1,2}])^2}/(4K - \|\boldsymbol{\mu}_{\mathbb{P}}\|_{\mathcal{H}_\kappa}^2 - \|\boldsymbol{\mu}_{\mathbb{Q}}\|_{\mathcal{H}_\kappa}^2)$, and the expectation are taken over $\boldsymbol{x}_1, \boldsymbol{x}_2, \boldsymbol{x}_3 \sim \mathbb{P}^3$ and $\boldsymbol{y}_1, \boldsymbol{y}_2, \boldsymbol{y}_3 \sim \mathbb{Q}^3$.

We can find the approximate test power by using the asymptotic testing threshold $\tau_\alpha^N$ as follows:

$$\text{Pr}\left( m\widehat{\text{NAMMD}}(X, Y; \kappa) \geq \tau_\alpha^N \right) - \Phi\left( \frac{m\text{NAMMD}(\mathbb{P}, \mathbb{Q}; \kappa) - \tau_\alpha^N}{\sqrt{m}\sigma_{\mathbb{P},\mathbb{Q}}} \right) \to 0 \ .$$

It is evident that maximizing the test power is equivalent to optimizing the following term

$$\frac{\text{NAMMD}(\mathbb{P}, \mathbb{Q}; \kappa)}{\sigma_{\mathbb{P},\mathbb{Q}}} = \frac{\text{MMD}(\mathbb{P}, \mathbb{Q}; \kappa)}{\sqrt{4E[H_{1,2}H_{1,3}] - 4(E[H_{1,2}])^2}} \ .$$

Recall that

$$\widehat{\text{NAMMD}}(X, Y; \kappa) = \sum_{i \neq j} H_{i,j} / \sum_{i \neq j} (4K - \kappa(\boldsymbol{x}_i, \boldsymbol{x}_j) - \kappa(\boldsymbol{y}_i, \boldsymbol{y}_j)) \ ,$$

with $H_{i,j} = \kappa(\boldsymbol{x}_i, \boldsymbol{x}_j) + \kappa(\boldsymbol{y}_i, \boldsymbol{y}_j) - \kappa(\boldsymbol{x}_i, \boldsymbol{y}_j) - \kappa(\boldsymbol{y}_i, \boldsymbol{x}_j)$ and

$$\sigma_{X,Y} = \frac{\sqrt{((4m - 8)\zeta_1 + 2\zeta_2)/(m - 1)}}{(m^2 - m)^{-1} \sum_{i \neq j} 4K - \kappa(\boldsymbol{x}_i, \boldsymbol{x}_j) - \kappa(\boldsymbol{y}_i, \boldsymbol{y}_j)} \ ,$$

where $\zeta_1$ and $\zeta_2$ are standard variance components of the MMD [45, 46]. The details of the $\zeta_1$ and $\zeta_2$ are provided in Appendix B.3.

We have the empirical $t$-statistic for test power estimation as follows

$$\frac{\widehat{\text{NAMMD}}(X, Y; \kappa)}{\sigma_{X,Y}} = \frac{\widehat{\text{MMD}}(X, Y; \kappa)}{\sqrt{((4m - 8)\zeta_1 + 2\zeta_2)/(m - 1)}} \ , \tag{5}$$

It is evident that the $t$-statistic for test power estimation of NAMMD is equal to the $t$-statistic for test power estimation of MMD [47]. We take gradient method [74] for the optimization of Eqn. 5. Algorithm 1 presents the detailed description on optimization.

### B.6 METHODOLOGY OF NAMMD-BASED TWO-SAMPLE TEST

Although the NAMMD is specially designed for DCT, it is still a statistic to measure the distributional discrepancy between two distributions. Thus, it is interesting to see how it performs in two-sample testing (TST) scenarios. In TST, we aim to assess the equivalence between distributions $\mathbb{P}$ and $\mathbb{Q}$ with null and alternative hypotheses as follows

$$\boldsymbol{H}'_0 : \mathbb{P} = \mathbb{Q} \quad \text{and} \quad \boldsymbol{H}'_1 : \mathbb{P} \neq \mathbb{Q} \ .$$

Following MMD-based TST [47], we implement our NAMMD-based TST via a permutation test, which estimates the null distribution by repeatedly re-computing the estimator with samples randomly reassigned to $X$ or $Y$. Specifically, denote by $B$ the iteration number of permutation test. Let $\boldsymbol{\Pi}_{2m}$ be the set of all possible permutations of $\{1, \ldots, 2m\}$ over the pooled sample $Z = \{\boldsymbol{x}_1, \ldots, \boldsymbol{x}_m, \boldsymbol{y}_1, \ldots, \boldsymbol{y}_m\} = \{\boldsymbol{z}_1, \ldots, \boldsymbol{z}_m, \boldsymbol{z}_{m+1}, \ldots, \boldsymbol{z}_{2m}\}$. In $b$-th iteration ($b \in [B]$), we generate a permutation $\boldsymbol{\pi} = (\pi_1, \ldots, \pi_{2m}) \in \boldsymbol{\Pi}_{2m}$ and then calculate the empirical estimator of NAMMD statistic as follows

$$T_b = \widehat{\text{NAMMD}}(X_{\boldsymbol{\pi}}, Y_{\boldsymbol{\pi}}, \kappa) \ ,$$

where $X_{\boldsymbol{\pi}} = \{\boldsymbol{z}_{\pi_1}, \boldsymbol{z}_{\pi_2}, ..., \boldsymbol{z}_{\pi_m}\}$ and $Y_{\boldsymbol{\pi}} = \{\boldsymbol{z}_{\pi_{m+1}}, \boldsymbol{z}_{\pi_{m+2}}, ..., \boldsymbol{z}_{\pi_{2m}}\}$.

During such process, we obtain $B$ statistics $T_1, T_2, ..., T_B$ and introduce the testing threshold for the null hypothesis $\boldsymbol{H}_0 : \text{NAMMD}(\mathbb{P}, \mathbb{Q}, \kappa) = 0$ as follows

$$\hat{\tau}'_\alpha = \arg\min_\tau \left\{ \sum_{b=1}^{B} \frac{\mathbb{I}[T_b \leq \tau]}{B} \geq 1 - \alpha \right\} \ .$$

Finally, we have the following test with the testing threshold $\tau_\alpha$ as follows

$$h'(X, Y, \kappa) = \mathbb{I}[\widehat{\text{NAMMD}}(X, Y, \kappa) > \hat{\tau}'_\alpha] \ .$$

## C DETAILED PROOFS OF THEORETICAL RESULTS

To begin, we define the concept of the $U$-statistic, which is a key statistical tool.

**Definition 10.** [45] Let $h(\boldsymbol{x}_1, \boldsymbol{x}_2, \ldots, \boldsymbol{x}_r)$ be a symmetric function of $r$ arguments. Suppose we have a random sample $\boldsymbol{x}_1, \boldsymbol{x}_2, \ldots, \boldsymbol{x}_m$ from some distribution. The U-statistic is given by:

$$U_m = \binom{m}{r}^{-1} \sum_{1 \leq i_1 < i_2 < \cdots < i_r \leq m} h(\boldsymbol{x}_{i_1}, \boldsymbol{x}_{i_2}, ..., \boldsymbol{x}_{i_r}) \ .$$

Here, $\binom{m}{r}$ is the number of ways to choose $r$ distinct indices from $m$, i.e., the binomial coefficient, and the summation is taken over all possible $r$-tuples from the sample.

We further present the large deviation for U-statistic as follows.

**Theorem 11.** [75] If the function $h$ is bounded, $a \leq h(\boldsymbol{x}_{i_1}, \boldsymbol{x}_{i_2}, ..., \boldsymbol{x}_{i_r}) \leq b$, we have

$$\Pr(|U_m - \theta| \geq t) \leq 2 \exp\left(-2\lfloor m/r \rfloor t^2/(b-a)^2\right) \ ,$$

where $\theta = E[h(\boldsymbol{x}_{i_1}, \boldsymbol{x}_{i_2}, ..., \boldsymbol{x}_{i_r})]$.

### C.1 DETAILED PROOFS OF LEMMA 2

We begin with the empirical estimator of MMD as

$$\widehat{\text{MMD}}^2(X, Y; \kappa) = 1/(m(m-1)) \sum_{i \neq j} \kappa(\boldsymbol{x}_i, \boldsymbol{x}_j) + \kappa(\boldsymbol{y}_i, \boldsymbol{y}_j) - \kappa(\boldsymbol{x}_i, \boldsymbol{y}_j) - \kappa(\boldsymbol{y}_i, \boldsymbol{x}_j) \ .$$

Given this, we introduce a useful theorem as follows.

**Lemma 12.** If $\mathbb{P} \neq \mathbb{Q}$, a standard central limit theorem holds [45, Section 5.5.1],

$$\sqrt{m} \left( \widehat{\text{MMD}}^2(X, Y; \kappa) - \text{MMD}^2(\mathbb{P}, \mathbb{Q}; \kappa) \right) \xrightarrow{d} \mathcal{N}\left(0, \sigma_M^2\right) \ ,$$

$$\sigma_M^2 := 4E[H_{1,2}H_{1,3}] - 4(E[H_{1,2}])^2 \ ,$$

where $H_{i,j} = \kappa(\boldsymbol{x}_i, \boldsymbol{x}_j) + \kappa(\boldsymbol{y}_i, \boldsymbol{y}_j) - \kappa(\boldsymbol{x}_i, \boldsymbol{y}_j) - \kappa(\boldsymbol{y}_i, \boldsymbol{x}_j)$ and the expectation are taken with respect to $\boldsymbol{x}_1, \boldsymbol{x}_2, \boldsymbol{x}_3 \overset{i.i.d.}{\sim} \mathbb{P}$ and $\boldsymbol{y}_1, \boldsymbol{y}_2, \boldsymbol{y}_3 \overset{i.i.d.}{\sim} \mathbb{Q}$.

We now present the proofs of Lemma 2 as follows.

*Proof.* Recall the empirical estimator of our NAMMD distance

$$
\begin{aligned}
m\widehat{\text{NAMMD}}(X, Y; \kappa) &= \frac{\sum_{i \neq j} \kappa(\boldsymbol{x}_i, \boldsymbol{x}_j) + \kappa(\boldsymbol{y}_i, \boldsymbol{y}_j) - \kappa(\boldsymbol{x}_i, \boldsymbol{y}_j) - \kappa(\boldsymbol{y}_i, \boldsymbol{x}_j)}{\sum_{i \neq j} 4K - \kappa(\boldsymbol{x}_i, \boldsymbol{x}_j) - \kappa(\boldsymbol{y}_i, \boldsymbol{y}_j)} \\
&= \frac{m\widehat{\text{MMD}}^2(X, Y; \kappa)}{1/(m^2 - m) \sum_{i \neq j} 4K - \kappa(\boldsymbol{x}_i, \boldsymbol{x}_j) - \kappa(\boldsymbol{y}_i, \boldsymbol{y}_j)} .
\end{aligned}
$$

As a U-statistic, by the large deviation bound (Theorem 11), it is easy to see that,

$$
1/(m(m-1)) \sum_{i \neq j} 4K - \kappa(\boldsymbol{x}_i, \boldsymbol{x}_j) - \kappa(\boldsymbol{y}_i, \boldsymbol{y}_j) \xrightarrow{p} 4K - \|\boldsymbol{\mu}_{\mathbb{P}}\|^2_{\mathcal{H}_\kappa} - \|\boldsymbol{\mu}_{\mathbb{Q}}\|^2_{\mathcal{H}_\kappa} ,
$$

where $\xrightarrow{p}$ denotes convergence in probability.

If $\text{NAMMD}(\mathbb{P}, \mathbb{Q}; \kappa) = \epsilon > 0$, we have $\text{MMD}(\mathbb{P}, \mathbb{Q}; \kappa) > 0$. Furthermore, from Lemma 12, we have, for $\mathbb{P} \neq \mathbb{Q}$,

$$
\sqrt{m} \left( \text{MMD}^2(X, Y; \kappa) - \text{MMD}^2(\mathbb{P}, \mathbb{Q}; \kappa) \right) \xrightarrow{d} \mathcal{N}(0, \sigma_M^2) .
$$

Then, by applying Slutsky's theorem [76], we obtain

$$
\frac{\sqrt{m}\text{MMD}^2(X, Y; \kappa)}{1/(m(m-1)) \sum_{i \neq j} 4K - \kappa(\boldsymbol{x}_i, \boldsymbol{x}_j) - \kappa(\boldsymbol{y}_i, \boldsymbol{y}_j)} - \frac{\sqrt{m}\text{MMD}^2(\mathbb{P}, \mathbb{Q}; \kappa)}{4K - \|\boldsymbol{\mu}_{\mathbb{P}}\|^2_{\mathcal{H}_\kappa} - \|\boldsymbol{\mu}_{\mathbb{Q}}\|^2_{\mathcal{H}_\kappa}}
$$

$$
\xrightarrow{d} \mathcal{N}\left(0, \frac{\sigma_M^2}{\left(4K - \|\boldsymbol{\mu}_{\mathbb{P}}\|^2_{\mathcal{H}_\kappa} - \|\boldsymbol{\mu}_{\mathbb{Q}}\|^2_{\mathcal{H}_\kappa}\right)^2}\right) .
$$

Recalling the definition of NAMMD, we have

$$
\sqrt{m}\widehat{\text{NAMMD}}(X, Y; \kappa) - \sqrt{m}\text{NAMMD}^2(\mathbb{P}, \mathbb{Q}; \kappa) \xrightarrow{d} \mathcal{N}\left(0, \frac{\sigma_M^2}{(4K - \|\boldsymbol{\mu}_{\mathbb{P}}\|^2_{\mathcal{H}_\kappa} - \|\boldsymbol{\mu}_{\mathbb{Q}}\|^2_{\mathcal{H}_\kappa})^2}\right) ,
$$

which can be expressed as

$$
\sqrt{m} \left( \widehat{\text{NAMMD}}(X, Y; \kappa) - \epsilon \right) \xrightarrow{d} \mathcal{N}\left(0, \frac{4E[H_{1,2}H_{1,3}] - 4(E[H_{1,2}])^2}{(4K - \|\boldsymbol{\mu}_{\mathbb{P}}\|^2_{\mathcal{H}_\kappa} - \|\boldsymbol{\mu}_{\mathbb{Q}}\|^2_{\mathcal{H}_\kappa})^2}\right) .
$$

This completes the proof. □

## C.2 DETAILED PROOFS OF LEMMA 4

We present the proofs of Lemma 4 as follows.

*Proof.* For simplicity, we let

$$
\hat{A} = \sqrt{((4m-8)\zeta_1 + 2\zeta_2)/(m-1)} \quad \text{and} \quad A = \sqrt{4E[H_{1,2}H_{1,3}] - 4(E[H_{1,2}])^2} ,
$$

and

$$
\hat{B} = (m^2 - m)^{-1} \sum_{i \neq j} 4K - \kappa(\mathbf{x}_i, \mathbf{x}_j) - \kappa(\mathbf{y}_i, \mathbf{y}_j) \quad \text{and} \quad B = 4K - \|\boldsymbol{\mu}_{\mathbb{P}}\|^2_{\mathcal{H}_\kappa} - \|\boldsymbol{\mu}_{\mathbb{Q}}\|^2_{\mathcal{H}_\kappa} .
$$

Build on these results, we can bound the bias as follows:

$$\left| E[\sigma^2_{X,Y}] - \sigma^2_{\mathbb{P},\mathbb{Q}} \right| = \left| E\left[\frac{\hat{A}^2}{\hat{B}^2}\right] - \frac{A^2}{B^2} \right| = \left| E\left[\frac{\hat{A}^2}{\hat{B}^2}\right] - E\left[\frac{\hat{A}^2}{B^2}\right] + E\left[\frac{\hat{A}^2}{B^2}\right] - \frac{A^2}{B^2} \right|$$

$$= \left| E\left[\frac{\hat{A}^2}{\hat{B}^2}\right] - E\left[\frac{\hat{A}^2}{B^2}\right] \right|$$

$$\leq E\left[ \left| \frac{\hat{A}^2}{\hat{B}^2} - \frac{\hat{A}^2}{B^2} \right| \right]$$

$$= E\left[ \left| \frac{\hat{A}^2(B - \hat{B})(B + \hat{B})}{\hat{B}^2 B^2} \right| \right]$$

$$\leq C * E\left[ \left| B - \hat{B} \right| \right]$$

where $C > 0$ is a constant that ensures $\frac{\hat{A}^2(B+\hat{B})}{\hat{B}^2 B^2} \leq C$, and it exists since the kernel is bounded. The second equation is based on the unbiased variance estimator of the $U$-statistic, i.e. $\hat{A}$. Based on the large deviation bound for $B$, we have

$$\Pr\left( \left| B - \hat{B} \right| \geq t \right) \leq 2\exp\left(-mt^2/4K^2\right)$$

and

$$C * E\left[ \left| B - \hat{B} \right| \right] = C * \int_0^\infty \Pr\left( \left| B - \hat{B} \right| \geq t \right) dt$$

$$\leq C * \int_0^\infty 2\exp\left(-mt^2/4K^2\right) dt$$

$$= C * \int_0^\infty 2\exp\left(-u\right)\frac{K}{\sqrt{m}\sqrt{u}} du$$

$$= C * \frac{2K\sqrt{\pi}}{\sqrt{m}} = O\left(\frac{1}{\sqrt{m}}\right).$$

This completes the proof. $\qquad\square$

## C.3 Detailed Proofs of Lemma 5

*Proof.* Recall our NAMMD distance as follows:

$$\text{NAMMD}(\mathbb{P},\mathbb{Q};\kappa) = \frac{\|\boldsymbol{\mu}_\mathbb{P} - \boldsymbol{\mu}_\mathbb{Q}\|^2_{\mathcal{H}_\kappa}}{4K - \|\boldsymbol{\mu}_\mathbb{P}\|^2_{\mathcal{H}_\kappa} - \|\boldsymbol{\mu}_\mathbb{Q}\|^2_{\mathcal{H}_\kappa}} = \frac{\text{MMD}^2(\mathbb{P},\mathbb{Q};\kappa)}{4K - \|\boldsymbol{\mu}_\mathbb{P}\|^2_{\mathcal{H}_\kappa} - \|\boldsymbol{\mu}_\mathbb{Q}\|^2_{\mathcal{H}_\kappa}}.$$

Given two i.i.d. samples $X = \{\boldsymbol{x}_1, \boldsymbol{x}_2, ..., \boldsymbol{x}_m\} \sim \mathbb{P}^m$ and $Y = \{\boldsymbol{y}_1, \boldsymbol{y}_2, ..., \boldsymbol{y}_m\} \sim \mathbb{Q}^m$, we have the empirical estimator as follows

$$\widehat{\text{NAMMD}}(X,Y;\kappa) = \frac{\sum_{i \neq j} \kappa(\boldsymbol{x}_i, \boldsymbol{x}_j) + \kappa(\boldsymbol{y}_i, \boldsymbol{y}_j) - \kappa(\boldsymbol{x}_i, \boldsymbol{y}_j) - \kappa(\boldsymbol{y}_i, \boldsymbol{x}_j)}{\sum_{i \neq j} 4K - \kappa(\boldsymbol{x}_i, \boldsymbol{x}_j) - \kappa(\boldsymbol{y}_i, \boldsymbol{y}_j)}$$

$$= \frac{\widehat{\text{MMD}}^2(X,Y;\kappa)}{1/(m^2 - m)\sum_{i \neq j} 4K - \kappa(\boldsymbol{x}_i, \boldsymbol{x}_j) - \kappa(\boldsymbol{y}_i, \boldsymbol{y}_j)}.$$

We denote by

$$A = \left| \widehat{\text{NAMMD}}(X,Y;\kappa) - \text{NAMMD}(\mathbb{P},\mathbb{Q};\kappa) \right|$$

$$= \left| \frac{\widehat{\text{MMD}}^2(X,Y;\kappa) - \text{MMD}^2(\mathbb{P},\mathbb{Q};\kappa) + \text{MMD}^2(\mathbb{P},\mathbb{Q};\kappa)}{1/(m^2 - m)\sum_{i \neq j} 4K - \kappa(\boldsymbol{x}_i, \boldsymbol{x}_j) - \kappa(\boldsymbol{y}_i, \boldsymbol{y}_j)} - \frac{\text{MMD}^2(\mathbb{P},\mathbb{Q};\kappa)}{4K - \|\boldsymbol{\mu}_\mathbb{P}\|^2_{\mathcal{H}_\kappa} - \|\boldsymbol{\mu}_\mathbb{Q}\|^2_{\mathcal{H}_\kappa}} \right|.$$

Given this, we let

$$B = \left| \frac{\widehat{\mathrm{MMD}}^2(X,Y;\kappa) - \mathrm{MMD}^2(\mathbb{P},\mathbb{Q};\kappa)}{1/(m^2-m)\sum_{i\neq j} 4K - \kappa(\boldsymbol{x}_i,\boldsymbol{x}_j) - \kappa(\boldsymbol{y}_i,\boldsymbol{y}_j)} \right| ,$$

and

$$C = \left| \frac{\mathrm{MMD}^2(\mathbb{P},\mathbb{Q};\kappa)}{1/(m^2-m)\sum_{i\neq j} 4K - \kappa(\boldsymbol{x}_i,\boldsymbol{x}_j) - \kappa(\boldsymbol{y}_i,\boldsymbol{y}_j)} - \frac{\mathrm{MMD}^2(\mathbb{P},\mathbb{Q};\kappa)}{4K - \|\boldsymbol{\mu}_\mathbb{P}\|_{\mathcal{H}_\kappa}^2 - \|\boldsymbol{\mu}_\mathbb{Q}\|_{\mathcal{H}_\kappa}^2} \right| .$$

It is easy to see that $A \le B + C$ and we have

$$\Pr(A \ge t) \le \Pr(B + C \ge t) \le \Pr(B \ge b) + \Pr(C \ge c) ,$$

for $b + c = t$ with $t > 0$ and $b, c \ge 0$.

Based on the large deviation bound for U-statistic (Theorem 11), we have

$$\Pr(B \ge b) \le \Pr\left( \left| \widehat{\mathrm{MMD}}^2(X,Y;\kappa) - \mathrm{MMD}^2(\mathbb{P},\mathbb{Q};\kappa) \right| / 2K \ge b \right) \le 2\exp\left(-mb^2/4\right) ,$$

In a similar manner, we have

$$\Pr(C \ge c)$$

$$\le \Pr\left( \frac{\mathrm{MMD}^2(\mathbb{P},\mathbb{Q};\kappa)|\sum_{i\neq j}(\kappa(\boldsymbol{x}_i,\boldsymbol{x}_j) + \kappa(\boldsymbol{y}_i,\boldsymbol{y}_j))/(m^2-m)) - \|\boldsymbol{\mu}_\mathbb{P}\|_{\mathcal{H}_\kappa}^2 - \|\boldsymbol{\mu}_\mathbb{Q}\|_{\mathcal{H}_\kappa}^2|}{(1/(m^2-m)\sum_{i\neq j} 4K - \kappa(\boldsymbol{x}_i,\boldsymbol{x}_j) - \kappa(\boldsymbol{y}_i,\boldsymbol{y}_j))\cdot(4K - \|\boldsymbol{\mu}_\mathbb{P}\|_{\mathcal{H}_\kappa}^2 - \|\boldsymbol{\mu}_\mathbb{Q}\|_{\mathcal{H}_\kappa}^2)} \ge c \right)$$

$$\le \Pr\left( \left| \sum_{i\neq j} \frac{\kappa(\boldsymbol{x}_i,\boldsymbol{x}_j)}{m(m-1)} + \frac{\kappa(\boldsymbol{y}_i,\boldsymbol{y}_j)}{m(m-1)} - \|\boldsymbol{\mu}_\mathbb{P}\|_{\mathcal{H}_\kappa}^2 - \|\boldsymbol{\mu}_\mathbb{Q}\|_{\mathcal{H}_\kappa}^2 \right| \frac{\mathrm{MMD}^2(\mathbb{P},\mathbb{Q};\kappa)}{4K^2} \ge c \right)$$

$$\le \Pr\left( \left| \sum_{i\neq j} \frac{\kappa(\boldsymbol{x}_i,\boldsymbol{x}_j)}{m(m-1)} + \frac{\kappa(\boldsymbol{y}_i,\boldsymbol{y}_j)}{m(m-1)} - \|\boldsymbol{\mu}_\mathbb{P}\|_{\mathcal{H}_\kappa}^2 - \|\boldsymbol{\mu}_\mathbb{Q}\|_{\mathcal{H}_\kappa}^2 \right| / 2K \ge c \right)$$

$$\le 2\exp\left(-mc^2\right)$$

For simplicity, let $b = 2t/3$ and $c = t/3$, we have

$$\begin{aligned} \Pr(A \ge t) &\le& \Pr(B \ge 2t/3) + \Pr(C \ge t/3) \\ &=& 4\exp\left(-mt^2/9\right) . \end{aligned}$$

This completes the proof. $\qquad\square$

### C.4 DETAILED PROOFS OF THEOREM 6

We present the proofs of Theorem 6 as follows.

*Proof.* Under null hypothesis $\boldsymbol{H}_0 : \mathrm{NAMMD}(\mathbb{P},\mathbb{Q};\kappa) \le \epsilon$ with $\epsilon \in (0,1)$, the type-I error is

$$\Pr(\mathrm{NAMMD}(X,Y;\kappa) > \tau_\alpha),$$

where $\hat{\tau}_\alpha = \epsilon + \sigma_{X,Y}\mathcal{N}_{1-\alpha}/\sqrt{m}$ (as defined in Eqn. (3)) is the $(1-\alpha)$-quantile of the asymptotic Gaussian distribution in Theorem 2 with $\mathrm{NAMMD}(\mathbb{P},\mathbb{Q};\kappa) = \epsilon$.

Recall that $\sigma_{X,Y}$ is the estimator of $\sigma_{\mathbb{P},\mathbb{Q}}^2$, where

$$\sigma_{\mathbb{P},\mathbb{Q}} = \frac{\sqrt{4E[H_{1,2}H_{1,3}] - 4(E[H_{1,2}])^2}}{4K - \|\boldsymbol{\mu}_\mathbb{P}\|_{\mathcal{H}_\kappa}^2 - \|\boldsymbol{\mu}_\mathbb{Q}\|_{\mathcal{H}_\kappa}^2} ,$$

and

$$\sigma_{X,Y} = \frac{\sqrt{((4m-8)\zeta_1 + 2\zeta_2)/(m-1)}}{(m^2-m)^{-1}\sum_{i\neq j} 4K - \kappa(\boldsymbol{x}_i,\boldsymbol{x}_j) - \kappa(\boldsymbol{y}_i,\boldsymbol{y}_j)},$$

where $\zeta_1$ and $\zeta_2$ are standard variance components of the MMD [45, 46] (Appendix B.3).

We begin by showing that $\sigma_{X,Y}$ converges to $\sigma_{\mathbb{P},\mathbb{Q}}$. As detailed in Appendix B.3, the terms in the numerator involving $\zeta_1$ and $\zeta_2$ converge in probability. We now present the convergence of the denominator

$$(m^2 - m)^{-1} \sum_{i \neq j} 4K - \kappa(\boldsymbol{x}_i, \boldsymbol{x}_j) - \kappa(\boldsymbol{y}_i, \boldsymbol{y}_j) \,,$$

which can be regarded as a $U$-statistic, and it follows that

$$(m^2 - m)^{-1} \sum_{i \neq j} 4K - \kappa(\boldsymbol{x}_i, \boldsymbol{x}_j) - \kappa(\boldsymbol{y}_i, \boldsymbol{y}_j) \xrightarrow{p} 4K - \|\boldsymbol{\mu}_{\mathbb{P}}\|_{\mathcal{H}_\kappa}^2 - \|\boldsymbol{\mu}_{\mathbb{Q}}\|_{\mathcal{H}_\kappa}^2 \,,$$

by the large deviation bound (Theorem 11) for $U$-statistic.

Hence, by the continuous mapping theorem, we have that

$$\sigma_{X,Y} \xrightarrow{p} \sigma_{\mathbb{P},\mathbb{Q}} \,.$$

Next, we prove asymptotic type-I error control based on the convergence of the variance. The null hypothesis $\boldsymbol{H}_0 : \text{NAMMD}(\mathbb{P}, \mathbb{Q}; \kappa) \leq \epsilon$ with $\epsilon \in (0, 1)$ is composite, covering three cases: 1) $\text{NAMMD}(\mathbb{P}, \mathbb{Q}; \kappa) = \epsilon$; 2) $\text{NAMMD}(\mathbb{P}, \mathbb{Q}; \kappa) = \epsilon' \in (0, \epsilon)$; 3) $\text{NAMMD}(\mathbb{P}, \mathbb{Q}; \kappa) = 0$. We now prove that under three cases the type-I error $\Pr(\text{NAMMD}(X, Y; \kappa) > \hat{\tau}_\alpha)$ are all bounded by $\alpha$.

- Case 1: $\text{NAMMD}(\mathbb{P}, \mathbb{Q}; \kappa) = \epsilon$. Since $\hat{\tau}_\alpha = \epsilon + \sigma_{X,Y} \mathcal{N}_{1-\alpha}/\sqrt{m}$ corresponds to the $(1 - \alpha)$-quantile of the asymptotic Gaussian distribution with $\text{NAMMD}(\mathbb{P}, \mathbb{Q}; \kappa) = \epsilon$ from Lemma 2, the following equality holds asymptotically

$$\Pr(\text{NAMMD}(X, Y; \kappa) > \hat{\tau}_\alpha) = \alpha.$$

- Case 2: $\text{NAMMD}(\mathbb{P}, \mathbb{Q}; \kappa) = \epsilon' \in (0, \epsilon)$. The $(1 - \alpha)$-quantile of the asymptotic Gaussian distribution with $\text{NAMMD}(\mathbb{P}, \mathbb{Q}; \kappa) = \epsilon'$ is $\hat{\tau}'_\alpha = \epsilon' + \sigma_{X,Y} \mathcal{N}_{1-\alpha}/\sqrt{m}$ from Lemma 2. Then, the following equality holds asymptotically

$$\Pr(\text{NAMMD}(X, Y; \kappa) > \hat{\tau}'_\alpha) = \alpha,$$

Since $\epsilon' < \epsilon$, we have $\hat{\tau}'_\alpha < \hat{\tau}_\alpha$ and

$$\Pr(\text{NAMMD}(X, Y; \kappa) > \hat{\tau}_\alpha) < \Pr(\text{NAMMD}(X, Y; \kappa) > \hat{\tau}'_\alpha) = \alpha$$

Hence, type-I error is bounded by $\alpha$.

- Case 3: $\text{NAMMD}(\mathbb{P}, \mathbb{Q}; \kappa) = 0$. According to the Lemma 5, we have that

$$\Pr(\text{NAMMD}(X, Y; \kappa) > \hat{\tau}_\alpha) < \Pr(\text{NAMMD}(X, Y; \kappa) > \epsilon) \leq 2 \exp(-m\epsilon^2/9) \,.$$

This probability decays exponentially with the sample size $m$, implying that

$$\Pr(\text{NAMMD}(X, Y; \kappa) > \hat{\tau}_\alpha) \leq \alpha \,,$$

holds in the asymptotic manner.

This completes the proof. $\qquad\square$

## C.5 DETAILED PROOFS OF THEOREM 7

*Proof.* Under the alternative hypothesis $\boldsymbol{H}_1 : \text{NAMMD}(\mathbb{P}, \mathbb{Q}; \kappa) > \epsilon$ with $\epsilon \in (0, 1)$, , we need to correctly reject the null hypothesis $\boldsymbol{H}_0 : \text{NAMMD}(\mathbb{P}, \mathbb{Q}; \kappa) \leq \epsilon$. According to Eqn. 3, we set $\hat{\tau}_\alpha$ as the $(1 - \alpha)$-quantile of the asymptotic null distribution of $\text{NAMMD}(\mathbb{P}, \mathbb{Q}; \kappa) = \epsilon$ from Lemma 2 as,

$$\hat{\tau}_\alpha = \epsilon + \frac{\sigma_{X,Y} \mathcal{N}_{1-\alpha}}{\sqrt{m}} \,,$$

where the empirical estimator of variance is given by

$$\sigma_{X,Y} = \frac{\sqrt{((4m - 8)\zeta_1 + 2\zeta_2)/(m - 1)}}{(m^2 - m)^{-1} \sum_{i \neq j} 4K - \kappa(\boldsymbol{x}_i, \boldsymbol{x}_j) - \kappa(\boldsymbol{y}_i, \boldsymbol{y}_j)} \,,$$

where $\zeta_1$ and $\zeta_2$ are standard variance components of the MMD [45, 46]. We present the details of the estimator in Appendix B.3.

It is easy to see that, for $\kappa(\cdot,\cdot) \leq K$,

$$(m^2 - m)^{-1} \sum_{i \neq j} 4K - \kappa(\boldsymbol{x}_i, \boldsymbol{x}_j) - \kappa(\boldsymbol{y}_i, \boldsymbol{y}_j) \geq 2K \quad \text{and} \quad \zeta_1 \leq 4K^2 \quad \text{and} \quad \zeta_2 \leq 4K^2 \ ,$$

Hence, as we can see,

$$\begin{aligned}
\sigma_{X,Y} &\leq \frac{\sqrt{(4m-6)/(m-1)4K^2}}{2K} \\
&\leq 4K/2K \\
&\leq 2 \ ,
\end{aligned}$$

and we have

$$\hat{\tau}_\alpha \leq \epsilon + \frac{2\mathcal{N}_{1-\alpha}}{\sqrt{m}} \ .$$

In a similar manner, to ensure the rejection, we have

$$\widehat{\text{NAMMD}}(X, Y; \kappa) > \epsilon + \frac{2\mathcal{N}_{1-\alpha}}{\sqrt{m}}.$$

To derive the bound, the following holds with at least probability $1 - \upsilon$,

$$\widehat{\text{NAMMD}}(X, Y; \kappa) \geq \text{NAMMD}(\mathbb{P}, \mathbb{Q}; \kappa) - \sqrt{\frac{9 \log 2/\upsilon}{m}} \ ,$$

then, we have

$$\text{NAMMD}(\mathbb{P}, \mathbb{Q}; \kappa) - \sqrt{\frac{9 \log 2/\upsilon}{m}} > \epsilon + \frac{2\mathcal{N}_{1-\alpha}}{\sqrt{m}} \ ,$$

which leads to

$$m \geq \frac{\left(2 * \mathcal{N}_{1-\alpha} + \sqrt{9 \log 2/\upsilon}\right)^2}{(\text{NAMMD}(\mathbb{P}, \mathbb{Q}; \kappa) - \epsilon)^2} \ .$$

This completes the proof. $\qquad \square$

## C.6 Detailed Proofs and Explanations of Theorem 9

### C.6.1 Detailed Proofs of Theorem 9

Given Definition 8, we assume $\mathbb{P}_1$ and $\mathbb{Q}_1$ are known, and $X$ and $Y$ are two i.i.d. samples drawn from $\mathbb{P}_2$ and $\mathbb{Q}_2$. The goals of distribution closeness testing are to correctly reject null hypotheses with calculated statistics $\widehat{\text{NAMMD}}(X, Y; \kappa)$ and $\widehat{\text{MMD}}(X, Y; \kappa)$.

For simplicity, we let

$$\begin{aligned}
\text{NORM}(\mathbb{P}_1, \mathbb{Q}_1; \kappa) &= 4K - \|\boldsymbol{\mu}_{\mathbb{P}_1}\|_{\mathcal{H}_\kappa}^2 - \|\boldsymbol{\mu}_{\mathbb{Q}_1}\|_{\mathcal{H}_\kappa}^2 \ , \\
\text{NORM}(\mathbb{P}_2, \mathbb{Q}_2; \kappa) &= 4K - \|\boldsymbol{\mu}_{\mathbb{P}_2}\|_{\mathcal{H}_\kappa}^2 - \|\boldsymbol{\mu}_{\mathbb{Q}_2}\|_{\mathcal{H}_\kappa}^2 \ ,
\end{aligned}$$

and rewrite the empirical estimator with $X$ and $Y$ as follows

$$\widehat{\text{NORM}}(X, Y; \kappa) = 1/(m^2 - m) \sum_{i \neq j} 4K - \kappa(\boldsymbol{x}_i, \boldsymbol{x}_j) - \kappa(\boldsymbol{y}_i, \boldsymbol{y}_j) \ .$$

*Proof.* [5]

---

[5] In this proof, $\tau_\alpha^M$ and $\tau_\alpha^N$ are the asymptotic $(1 - \alpha)$-quantile of distributions of the MMD and NAMMD estimator, under the null hypothesis $\boldsymbol{H}_0^M : \text{MMD}(\mathbb{P}_2, \mathbb{Q}_2; \kappa) \leq \epsilon^M$ and $\boldsymbol{H}_0^N : \text{NAMMD}(\mathbb{P}_2, \mathbb{Q}_2; \kappa) \leq \epsilon^N$ for distribution closeness testing. Here, $\epsilon^M = \text{MMD}(\mathbb{P}_1, \mathbb{Q}_1; \kappa)$ and $\epsilon^N = \text{NAMMD}(\mathbb{P}_1, \mathbb{Q}_1; \kappa)$. The respective null distributions for MMD and NAMMD are presented in Lemmas 12 and 2. In practical, since these null distributions are normal, we directly estimate the testing thresholds $\tau_\alpha^M$ and $\tau_\alpha^N$ by computing the variances of the corresponding normal distributions [28, 27, 25, 32].

Let $\tau_\alpha^M$ and $\tau_\alpha^N$ be the true $(1-\alpha)$-quantiles of asymptotic null distributions of $\sqrt{m}\widehat{\text{MMD}}$ and $\sqrt{m}\widehat{\text{NAMMD}}$, respectively. Specifically, from Lemma 12, we have

$$\tau_\alpha^M = \sqrt{m}\text{MMD}(\mathbb{P}_1, \mathbb{Q}_1; \kappa) + \sigma_M \mathcal{N}_{1-\alpha} ,$$

where

$$\sigma_M^2 := 4E[H_{1,2}H_{1,3}] - 4(E[H_{1,2}])^2 , \qquad (6)$$

and $H_{i,j} = \kappa(\boldsymbol{x}_i, \boldsymbol{x}_j) + \kappa(\boldsymbol{y}_i, \boldsymbol{y}_j) - \kappa(\boldsymbol{x}_i, \boldsymbol{y}_j) - \kappa(\boldsymbol{y}_i, \boldsymbol{x}_j)$, and the expectation are taken with respect to $\boldsymbol{x}_1, \boldsymbol{x}_2, \boldsymbol{x}_3 \overset{\text{i.i.d.}}{\sim} \mathbb{P}_2$ and $\boldsymbol{y}_1, \boldsymbol{y}_2, \boldsymbol{y}_3 \overset{\text{i.i.d.}}{\sim} \mathbb{Q}_2$.

In a similar manner, from Lemma 2, we have

$$\begin{aligned}
\tau_\alpha^N &= \sqrt{m}\text{NAMMD}(\mathbb{P}_1, \mathbb{Q}_1; \kappa) + \sigma_{\mathbb{P}_2, \mathbb{Q}_2}\mathcal{N}_{1-\alpha} \\
&= \frac{\sqrt{m}\text{MMD}(\mathbb{P}_1, \mathbb{Q}_1; \kappa)}{4K - \|\boldsymbol{\mu}_{\mathbb{P}_1}\|_{\mathcal{H}_\kappa}^2 - \|\boldsymbol{\mu}_{\mathbb{Q}_1}\|_{\mathcal{H}_\kappa}^2} + \frac{\sigma_M \mathcal{N}_{1-\alpha}}{4K - \|\boldsymbol{\mu}_{\mathbb{P}_2}\|_{\mathcal{H}_\kappa}^2 - \|\boldsymbol{\mu}_{\mathbb{Q}_2}\|_{\mathcal{H}_\kappa}^2} \\
&= \frac{\sqrt{m}\text{MMD}(\mathbb{P}_1, \mathbb{Q}_1; \kappa)}{\text{NORM}(\mathbb{P}_1, \mathbb{Q}_1; \kappa)} + \frac{\sigma_M \mathcal{N}_{1-\alpha}}{\text{NORM}(\mathbb{P}_2, \mathbb{Q}_2; \kappa)} ,
\end{aligned}$$

It is easy to see that $\sqrt{m}\widehat{\text{MMD}}(X, Y; \kappa) > \tau_\alpha^M$ is equivalent to

$$\sqrt{m}\widehat{\text{MMD}}(X, Y; \kappa) - \sqrt{m}\text{MMD}(\mathbb{P}_1, \mathbb{Q}_1; \kappa) > \sigma_M \mathcal{N}_{1-\alpha} , \qquad (7)$$

and in a similar manner, $\sqrt{m}\widehat{\text{NAMMD}}(X, Y; \kappa) > \tau_\alpha^N$ is equivalent to

$$\frac{\text{NORM}(\mathbb{P}_2, \mathbb{Q}_2; \kappa)}{\widehat{\text{NORM}}(X, Y; \kappa)}\sqrt{m}\widehat{\text{MMD}}(X, Y; \kappa) - \frac{\text{NORM}(\mathbb{P}_2, \mathbb{Q}_2; \kappa)}{\text{NORM}(\mathbb{P}_1, \mathbb{Q}_1; \kappa)}\sqrt{m}\text{MMD}(\mathbb{P}_1, \mathbb{Q}_1; \kappa) > \sigma_M \mathcal{N}_{1-\alpha} ,$$

$$(8)$$

Hence, to ensure

$$\sqrt{m}\widehat{\text{MMD}}(X, Y; \kappa) > \tau_\alpha^M \quad \Rightarrow \quad \sqrt{m}\widehat{\text{NAMMD}}(X, Y; \kappa) > \tau_\alpha^N , \qquad (9)$$

we must verify that, according to Eqn. 7 and 8,

$$\left(\frac{\text{NORM}(\mathbb{P}_2, \mathbb{Q}_2; \kappa)}{\widehat{\text{NORM}}(X, Y; \kappa)} - 1\right)\sqrt{m}\widehat{\text{MMD}}(X, Y; \kappa) \geq \left(\frac{\text{NORM}(\mathbb{P}_2, \mathbb{Q}_2; \kappa)}{\text{NORM}(\mathbb{P}_1, \mathbb{Q}_1; \kappa)} - 1\right)\sqrt{m}\text{MMD}(\mathbb{P}_1, \mathbb{Q}_1; \kappa) .$$

$$(10)$$

Based on Eqn. 7, the inequality in Eqn. 10 can be adjusted to

$$\begin{aligned}
&\frac{\text{NORM}(\mathbb{P}_2, \mathbb{Q}_2; \kappa) - \widehat{\text{NORM}}(X, Y; \kappa)}{\widehat{\text{NORM}}(X, Y; \kappa)} \\
&\geq \frac{\text{NORM}(\mathbb{P}_2, \mathbb{Q}_2; \kappa) - \text{NORM}(\mathbb{P}_1, \mathbb{Q}_1; \kappa)}{\text{NORM}(\mathbb{P}_1, \mathbb{Q}_1; \kappa)}\frac{\sqrt{m}\text{MMD}(\mathbb{P}_1, \mathbb{Q}_1; \kappa)}{\sqrt{m}\text{MMD}(\mathbb{P}_1, \mathbb{Q}_1; \kappa) + \sigma_M \mathcal{N}_{1-\alpha}} \\
&\geq \sqrt{m}\text{NAMMD}(\mathbb{P}_1, \mathbb{Q}_1; \kappa)\frac{\text{NORM}(\mathbb{P}_2, \mathbb{Q}_2; \kappa) - \text{NORM}(\mathbb{P}_1, \mathbb{Q}_1; \kappa)}{\sqrt{m}\text{MMD}(\mathbb{P}_1, \mathbb{Q}_1; \kappa) + \sigma_M \mathcal{N}_{1-\alpha}} .
\end{aligned}$$

Given this, we have

$$\begin{aligned}
&\text{NORM}(\mathbb{P}_2, \mathbb{Q}_2; \kappa) \\
&\geq \left(1 + \sqrt{m}\text{NAMMD}(\mathbb{P}_1, \mathbb{Q}_1; \kappa)\frac{\text{NORM}(\mathbb{P}_2, \mathbb{Q}_2; \kappa) - \text{NORM}(\mathbb{P}_1, \mathbb{Q}_1; \kappa)}{\sqrt{m}\text{MMD}(\mathbb{P}_1, \mathbb{Q}_1; \kappa) + \sigma_M \mathcal{N}_{1-\alpha}}\right)\widehat{\text{NORM}}(X, Y; \kappa) \\
&\geq (1 - \Delta)\widehat{\text{NORM}}(X, Y; \kappa) ,
\end{aligned}$$

where we let, for simplicity

$$\Delta = \sqrt{m}\text{NAMMD}(\mathbb{P}_1, \mathbb{Q}_1; \kappa)\frac{\|\boldsymbol{\mu}_{\mathbb{P}_2}\|_{\mathcal{H}_\kappa}^2 + \|\boldsymbol{\mu}_{\mathbb{Q}_2}\|_{\mathcal{H}_\kappa}^2 - \|\boldsymbol{\mu}_{\mathbb{P}_1}\|_{\mathcal{H}_\kappa}^2 - \|\boldsymbol{\mu}_{\mathbb{Q}_1}\|_{\mathcal{H}_\kappa}^2}{\sqrt{m}\text{MMD}(\mathbb{P}_1, \mathbb{Q}_1; \kappa) + \sigma_M \mathcal{N}_{1-\alpha}} .$$

Here, by assuming $\|\boldsymbol{\mu}_{\mathbb{P}_1}\|_{\mathcal{H}_\kappa}^2 + \|\boldsymbol{\mu}_{\mathbb{Q}_1}\|_{\mathcal{H}_\kappa}^2 < \|\boldsymbol{\mu}_{\mathbb{P}_2}\|_{\mathcal{H}_\kappa}^2 + \|\boldsymbol{\mu}_{\mathbb{Q}_2}\|_{\mathcal{H}_\kappa}^2$, we have $\Delta \in (0, 1/2)$.

As we can see, $\text{NORM}(\mathbb{P}_2, \mathbb{Q}_2; \kappa) \geq (1 - \Delta)\widehat{\text{NORM}}(X, Y; \kappa)$ is equivalent to

$$(1 - \Delta)\widehat{\text{NORM}}(X, Y; \kappa) - (1 - \Delta)\text{NORM}(\mathbb{P}_2, \mathbb{Q}_2; \kappa) \leq \Delta \cdot \text{NORM}(\mathbb{P}_2, \mathbb{Q}_2; \kappa) \,,$$

which is

$$\widehat{\text{NORM}}(X, Y; \kappa) - \text{NORM}(\mathbb{P}_2, \mathbb{Q}_2; \kappa) \leq \frac{\Delta}{1 - \Delta}\text{NORM}(\mathbb{P}_2, \mathbb{Q}_2; \kappa) \,.$$

Using the large deviation bound as follows

$$P\left(\widehat{\text{NORM}}(X, Y; \kappa) - \text{NORM}(\mathbb{P}_2, \mathbb{Q}_2; \kappa) \geq t\right) \leq \exp(-mt^2/4K^2) \,,$$

with $t > 0$, the Eqn. 9 holds with probability at least

$$1 - \exp\left(-m\left(\frac{\Delta}{1 - \Delta}\text{NORM}(\mathbb{P}_2, \mathbb{Q}_2; \kappa)\right)^2 / 4K^2\right) \,.$$

This completes the proof of first part.

From Lemma 12, we have the test power of MMD test as follows

$$p_M = \Pr\left(\sqrt{m}\widehat{\text{MMD}}^2(X, Y; \kappa) \geq \tau_\alpha^M\right) \,,$$

with

$$\Pr\left(\sqrt{m}\widehat{\text{MMD}}^2(X, Y; \kappa) \geq \tau_\alpha^M\right) - \Phi\left(\frac{\sqrt{m}\text{MMD}^2(\mathbb{P}_2, \mathbb{Q}_2; \kappa) - \tau_\alpha^M}{\sigma_M}\right) \to 0 \,,$$

which is equivalent to

$$\Phi\left(\frac{\sqrt{m}(\text{MMD}^2(\mathbb{P}_2, \mathbb{Q}_2; \kappa) - \text{MMD}^2(\mathbb{P}_1, \mathbb{Q}_1; \kappa)) - \sigma_M\mathcal{N}_{1-\alpha}}{\sigma_M}\right) \,.$$

The test power of NAMMD test is given by, according to Lemma 2,

$$p_N = \Pr\left(\sqrt{m}\widehat{\text{NAMMD}}(X, Y; \kappa) \geq \tau_\alpha^N\right) \,,$$

with

$$\Pr\left(\sqrt{m}\widehat{\text{NAMMD}}(X, Y; \kappa) \geq \tau_\alpha^N\right) - \Phi\left(\frac{\sqrt{m}\text{NAMMD}(\mathbb{P}_2, \mathbb{Q}_2; \kappa) - \tau_\alpha^N}{\sigma_{\mathbb{P}_2, \mathbb{Q}_2}}\right) \to 0 \,,$$

which is equivalent to

$$\Phi\left(\frac{\sqrt{m}\left(\text{MMD}^2(\mathbb{P}_2, \mathbb{Q}_2; \kappa) - \frac{\text{NORM}(\mathbb{P}_2, \mathbb{Q}_2; \kappa)}{\text{NORM}(\mathbb{P}_1, \mathbb{Q}_1; \kappa)}\text{MMD}^2(\mathbb{P}_1, \mathbb{Q}_1; \kappa)\right) - \sigma_M\mathcal{N}_{1-\alpha}}{\sigma_M}\right) \,.$$

For simplicity, we let

$$A = \frac{\sqrt{m}(\text{MMD}^2(\mathbb{P}_2, \mathbb{Q}_2; \kappa) - \text{MMD}^2(\mathbb{P}_1, \mathbb{Q}_1; \kappa)) - \sigma_M\mathcal{N}_{1-\alpha}}{\sigma_M} \,,$$

and

$$B = \sqrt{m}\left(1 - \frac{\text{NORM}(\mathbb{P}_2, \mathbb{Q}_2; \kappa)}{\text{NORM}(\mathbb{P}_1, \mathbb{Q}_1; \kappa)}\right)\frac{\text{MMD}^2(\mathbb{P}_1, \mathbb{Q}_1; \kappa)}{\sigma_M} \,.$$

Similarly, by assuming $\|\boldsymbol{\mu}_{\mathbb{P}_1}\|_{\mathcal{H}_\kappa}^2 + \|\boldsymbol{\mu}_{\mathbb{Q}_1}\|_{\mathcal{H}_\kappa}^2 < \|\boldsymbol{\mu}_{\mathbb{P}_2}\|_{\mathcal{H}_\kappa}^2 + \|\boldsymbol{\mu}_{\mathbb{Q}_2}\|_{\mathcal{H}_\kappa}^2$, we have $B > 0$ with $\text{NORM}(\mathbb{P}_1, \mathbb{Q}_1; \kappa) > \text{NORM}(\mathbb{P}_2, \mathbb{Q}_2; \kappa)$.

As we can see,

$$\varsigma = p_N - p_M = \frac{1}{\sqrt{2\pi}} \int_A^{A+B} e^{-t^2/2} dt \ .$$

*which indicates that the NAMMD-based DCT achieves higher test power than the MMD-based DCT by a margin of $\varsigma$.*

Next, we examine the case where $\varsigma \geq 1/65$. Considering the following inequality holds

$$0 \leq \frac{\sqrt{m}\text{MMD}^2(\mathbb{P}_2, \mathbb{Q}_2; \kappa) - \tau_\alpha^M}{\sigma_M} \leq 0.7 \ ,$$

which is equivalent to

$$0 \leq A \leq 0.7 \ .$$

It follows that

$$m_A^- \leq m \leq m_A^+ \ ,$$

where

$$m_A^- = \left( \frac{\mathcal{N}_{1-\alpha}\sigma_M}{\text{MMD}^2(\mathbb{P}_2, \mathbb{Q}_2; \kappa) - \text{MMD}^2(\mathbb{P}_1, \mathbb{Q}_1; \kappa)} \right)^2 \ ,$$

$$m_A^+ = \left( \frac{(\mathcal{N}_{1-\alpha} + 0.7)\sigma_M}{\text{MMD}^2(\mathbb{P}_2, \mathbb{Q}_2; \kappa) - \text{MMD}^2(\mathbb{P}_1, \mathbb{Q}_1; \kappa)} \right)^2 \ .$$

In a similar manner, let $B \geq 0.05$, we have

$$m \geq m_B \ ,$$

where

$$m_B = \left( 20 \left( 1 - \frac{\text{NORM}(\mathbb{P}_2, \mathbb{Q}_2; \kappa)}{\text{NORM}(\mathbb{P}_1, \mathbb{Q}_1; \kappa)} \right) \frac{\text{MMD}^2(\mathbb{P}_1, \mathbb{Q}_1; \kappa)}{\sigma_M} \right)^{-2} \ .$$

By introducing

$$C_1 \leq m \leq C_2 \ ,$$

with

$$C_1 = \max \left\{ m_A^-, m_B \right\} \qquad \text{and} \qquad C_2 = m_A^+ \ ,$$

it follows that $B \geq 0.05$ and $-0.75 \leq A \leq 0.70$, and the lower bound of the power improvement is given by

$$\varsigma = p_N - p_M \geq \frac{1}{\sqrt{2\pi}} \int_{0.7}^{0.75} e^{-t^2/2} dt \geq 1/65 \ .$$

This completes the proof. $\qquad\qquad\qquad\qquad\qquad\qquad\qquad\qquad\qquad\qquad\qquad\quad\square$

### C.6.2 DETAILED EXPLANATION ON THE CONDITION AND CONSTANTS IN THEOREM 9

In Theorem, the condition

$$\|\boldsymbol{\mu}_{\mathbb{P}_1}\|^2_{\mathcal{H}_\kappa} + \|\boldsymbol{\mu}_{\mathbb{Q}_1}\|^2_{\mathcal{H}_\kappa} < \|\boldsymbol{\mu}_{\mathbb{P}_2}\|^2_{\mathcal{H}_\kappa} + \|\boldsymbol{\mu}_{\mathbb{Q}_2}\|^2_{\mathcal{H}_\kappa}$$

is closely related to the variance of the distributions, as discussed in Section 1. Specifically, the kernel-based variance is defined as

$$\text{Var}(\mathbb{P}; \kappa) = E_{\boldsymbol{x} \sim \mathbb{P}}[\kappa(\boldsymbol{x}, \boldsymbol{x})] - \|\boldsymbol{\mu}_{\mathbb{P}}\|^2_{\mathcal{H}_\kappa} = K - \|\boldsymbol{\mu}_{\mathbb{P}}\|^2_{\mathcal{H}_\kappa} \ ,$$

where $\kappa(\boldsymbol{x}, \boldsymbol{x}') = \Psi(\boldsymbol{x} - \boldsymbol{x}') \leq K$ with $K > 0$ for a positive-definite $\Psi(\cdot)$ and $\Psi(\boldsymbol{0}) = K$.

Given the variance term, the condition can be equivalently expressed as

$$\text{Var}(\mathbb{P}_1; \kappa) + \text{Var}(\mathbb{Q}_1; \kappa) > \text{Var}(\mathbb{P}_2; \kappa) + \text{Var}(\mathbb{Q}_2; \kappa) \ .$$

In the NAMMD distance, we incorporate the norms of distributions (i.e., variance information of distributions), and we analyze its advantages through Theorem 9 using the following example.

**Example 1.** From Appendix C.6.1, we have that

$$C_1 = \max\left\{m_A^-, m_B\right\} ,$$

$$C_2 = \left(\frac{(\mathcal{N}_{1-\alpha} + 0.7)\sigma_M}{\text{MMD}^2(\mathbb{P}_2, \mathbb{Q}_2; \kappa) - \text{MMD}^2(\mathbb{P}_1, \mathbb{Q}_1; \kappa)}\right)^2 ,$$

with

$$m_A^- = \left(\frac{\mathcal{N}_{1-\alpha}\sigma_M}{\text{MMD}^2(\mathbb{P}_2, \mathbb{Q}_2; \kappa) - \text{MMD}^2(\mathbb{P}_1, \mathbb{Q}_1; \kappa)}\right)^2 ,$$

$$m_B = \left(20\left(1 - \frac{\text{NORM}(\mathbb{P}_2, \mathbb{Q}_2; \kappa)}{\text{NORM}(\mathbb{P}_1, \mathbb{Q}_1; \kappa)}\right)\frac{\text{MMD}^2(\mathbb{P}_1, \mathbb{Q}_1; \kappa)}{\sigma_M}\right)^{-2} .$$

Consider the reference distribution pair $\mathbb{P}_1 = \mathcal{N}(0, 1.1)$ and $\mathbb{Q}_1 = \mathcal{N}(0, 1.6)$, and the distribution pair $\mathbb{P}_2 = \mathcal{N}(0, 0.5)$ and $\mathbb{Q}_2 = \mathcal{N}(0, 1.0)$ for testing. A Gaussian kernel $\kappa$ with bandwidth 1.0 is employed. Under this setup, it follows that

$$\|\boldsymbol{\mu}_{\mathbb{P}_1}\|^2_{\mathcal{H}_\kappa} = E_{\boldsymbol{x}, \boldsymbol{x}' \sim \mathbb{P}_1}[\exp(-\|\boldsymbol{x} - \boldsymbol{x}'\|^2_2)] = \int_{-\infty}^{\infty} \frac{\exp\left(-z^2\right)\exp\left(-z^2/(2 \times 0.02)\right)}{(2\pi \times 0.02)^{0.5}}dz = 0.4303 ,$$

where we denote $\boldsymbol{z} = \boldsymbol{x} - \boldsymbol{x}'$ in the evaluation of the integral; similarly, we obtain that

$$\|\boldsymbol{\mu}_{\mathbb{Q}_1}\|^2_{\mathcal{H}_\kappa} = E_{\boldsymbol{y}, \boldsymbol{y}' \sim \mathbb{Q}_1}[\exp(-\|\boldsymbol{y} - \boldsymbol{y}'\|^2_2)] = \int_{-\infty}^{\infty} \frac{\exp\left(-z^2\right)\exp\left(-z^2/(2 \times 2)\right)}{(2\pi \times 2)^{0.5}}dz = 0.3676 ,$$

by denoting $\boldsymbol{z} = \boldsymbol{y} - \boldsymbol{y}'$ in the evaluation of the integral; similarly, we obtain that

$$\langle\boldsymbol{\mu}_{\mathbb{P}_1}, \boldsymbol{\mu}_{\mathbb{Q}_1}\rangle_{\mathcal{H}_\kappa} = E_{\boldsymbol{x} \sim \mathbb{P}_1, \boldsymbol{y} \sim \mathbb{Q}_1}[\exp(-\|\boldsymbol{x} - \boldsymbol{y}\|^2_2)] = \int_{-\infty}^{\infty} \frac{\exp\left(-z^2\right)\exp\left(-z^2/(2 \times 2)\right)}{(2\pi \times 2)^{0.5}}dz = 0.3953,$$

by denoting $\boldsymbol{z} = \boldsymbol{x} - \boldsymbol{y}'$ in the evaluation of the integral. Based on these norms, we can calculate that $\text{MMD}^2(\mathbb{P}_1, \mathbb{Q}_1; \kappa) = 0.0073$.

In a similar manner, we have that

$$\|\boldsymbol{\mu}_{\mathbb{P}_2}\|^2_{\mathcal{H}_\kappa} = 0.5773, \quad \|\boldsymbol{\mu}_{\mathbb{Q}_2}\|^2_{\mathcal{H}_\kappa} = 0.4472, \quad \langle\boldsymbol{\mu}_{\mathbb{P}_2}, \boldsymbol{\mu}_{\mathbb{Q}_2}\rangle_{\mathcal{H}_\kappa} = 0.5, \quad \text{MMD}^2(\mathbb{P}_2, \mathbb{Q}_2; \kappa) = 0.0245 .$$

For the variance term $\sigma_M$ defined over $\mathbb{P}_2$ and $\mathbb{Q}_2$, which is difficult to compute analytically, we approximate its value using empirical estimates obtained from 1,000 runs with 10,000 samples each. Specifically,

$$\sigma_M^2 = 0.0274 .$$

Finally, we compute $m_A^- = 250.5810$, $m_B = 256.6816$ and

$$C_1 = 256.6816 \qquad \text{and} \qquad C_2 = 509.2431 .$$

# D  DETAILS OF OUR EXPERIMENTS

## D.1  DETAILS OF THE EXPERIMENTS IN FIGURE 1

### D.1.1  KEY EXPERIMENTAL PARAMETERS OF THE EXPERIMENTS IN FIGURE 1

The $p$-values in Figure 1 are obtained using a standard permutation test. Specifically, for illustration, we use two underlying 2-dimensional Gaussian distributions with a MMD value $\text{MMD}(\mathbb{P}, \mathbb{Q}) = 0.15$ and we use 4 samples from each distribution for every experiment. A Gaussian kernel with bandwidth 1 is used throughout. In the permutation procedure, for each test, we approximate the null distribution using 1000 permutations. To reduce randomness induced by the small sample size, we repeat the entire procedure 100 times, and Figure 1 reports the average $p$-value and its standard deviation over these repetitions.

### D.1.2  RELATIONSHIP BETWEEN THE $p$-VALUE OF THE MMD ESTIMATOR AND THE RKHS NORMS OF THE DISTRIBUTIONS

As shown in Figure 1c, the empirical results indicate that, for distribution pairs $(\mathbb{P}, \mathbb{Q})$ with the same MMD value (e.g., $\mathrm{MMD}(\mathbb{P}, \mathbb{Q}; \kappa) = 0.15$), those with larger RKHS norms in $\mathcal{H}_\kappa$ tend to yield smaller $p$-values in two-sample testing (TST). A smaller $p$-value indicates that $\mathbb{P}$ and $\mathbb{Q}$ are less likely satisfy the null hypothesis (i.e., $\mathbb{P} = \mathbb{Q}$); and hence, the two distributions $\mathbb{P}$ and $\mathbb{Q}$ are inferred to be more distinguishable and therefore less close to each other with larger RKHS norms (i.e., $\|\mathbb{P}\|_{\mathcal{H}_\kappa}^2$ and $\|\mathbb{Q}\|_{\mathcal{H}_\kappa}^2$).

The observed decrease in the $p$-value of MMD as the RKHS norms of the distributions increase can be attributed to a corresponding reduction in the standard deviation of the MMD estimator, as illustrated in Figure 6. This reduction arises from the increased concentration of the distributions in the RKHS: as the norms of the distributions grow, their RKHS variances, i.e, $\mathrm{Var}(\mathbb{P}; \kappa) = K - \|\boldsymbol{\mu}_\mathbb{P}\|_{\mathcal{H}_\kappa}^2$ and $\mathrm{Var}(\mathbb{Q}; \kappa) = K - \|\boldsymbol{\mu}_\mathbb{Q}\|_{\mathcal{H}_\kappa}^2$ decrease accordingly. Specifically, a smaller standard deviation implies a lower likelihood that the MMD estimator under the null hypothesis (i.e., $\mathbb{P} = \mathbb{Q}$, with MMD equal to

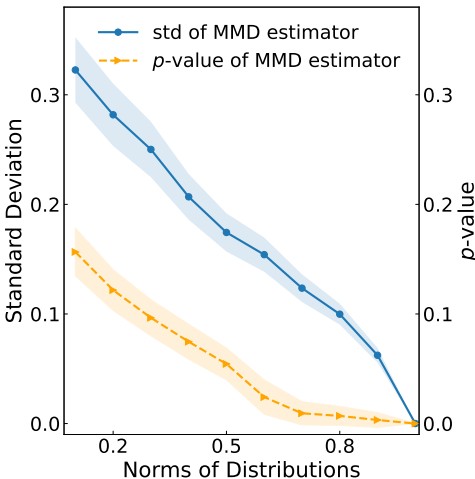

**Figure 6:** The relationship between the $p$-value and the standard deviation of the MMD estimator in two-sample testing, in connection with the norms of the underlying distributions in RKHS.

zero) falls within the region typically associated with the alternative hypothesis (i.e., MMD equal to 0.15). Consequently, the $p$-value, defined as this probability, decreases as the RKHS norms of the distributions increase.

## D.2 DETAILS OF EXPERIMENTS WITH DISTRIBUTIONS OVER IDENTICAL DOMAIN

**Data Construction.** Let $\mathbb{P}_n = \{p_1, p_2, ..., p_n\}$ and $\mathbb{Q}_n = \{q_1, q_2, ..., q_n\}$ be two discrete distributions over the same domain $Z = \{\boldsymbol{z}_1, \boldsymbol{z}_2, ..., \boldsymbol{z}_n\} \subseteq \mathbb{R}^d$ such that $\sum_{i=1}^n p_i = 1$ and $\sum_{i=1}^n q_i = 1$. We define the total variation [48] of $\mathbb{P}_n$ and $\mathbb{Q}_n$ as

$$\mathrm{TV}(\mathbb{P}_n, \mathbb{Q}_n) = \sup_{S \subseteq Z} (\mathbb{P}_n(S) - \mathbb{Q}_n(S)) = \frac{1}{2} \sum_{i=1}^n |p_i - q_i| = \frac{1}{2} \|\mathbb{P}_n - \mathbb{Q}_n\|_1 \in [0, 1] \,.$$

As we can see, the corresponding NAMMD distance can be calculated as

$$
\begin{aligned}
\mathrm{NAMMD}(\mathbb{P}_n, \mathbb{Q}_n; \kappa) &= \frac{\|\boldsymbol{\mu}_{\mathbb{P}_n} - \boldsymbol{\mu}_{\mathbb{Q}_n}\|_{\mathcal{H}_\kappa}^2}{4K - \|\boldsymbol{\mu}_{\mathbb{P}_n}\|_{\mathcal{H}_\kappa}^2 - \|\boldsymbol{\mu}_{\mathbb{Q}_n}\|_{\mathcal{H}_\kappa}^2} \\
&= \frac{\sum_{i,j} p_i p_j \kappa(\boldsymbol{z}_i, \boldsymbol{z}_j) + q_i q_j \kappa(\boldsymbol{z}_i, \boldsymbol{z}_j) - 2 p_i q_j \kappa(\boldsymbol{z}_i, \boldsymbol{z}_j)}{4K - \sum_{i,j} (p_i p_j \kappa(\boldsymbol{z}_i, \boldsymbol{z}_j) + q_i q_j \kappa(\boldsymbol{z}_i, \boldsymbol{z}_j))} \,.
\end{aligned}
$$

Here, we take the uniform distribution $\mathbb{P}_n = \{1/n, 1/n, ..., 1/n\}$ over sample $Z$, where $p_i = 1/n$ for $i \in \{1, 2, ..., n\}$. We construct discrete distribution $\mathbb{Q}_n$, which is $\epsilon' \in [0, 1]$ total variation away from the uniform distribution $\mathbb{P}_n$, as follows: We initiate the $\mathbb{Q}_n = \mathbb{P}_n$ and randomly split the sample $Z$ into two parts. In the first part, we increase the sample probability of each element by $\epsilon'/n$; and in the second part, we decrease the sample probability of each element by $\epsilon'/n$.

**Testing Threshold for Canonne's test.** Under null hypothesis $H_0' : \mathrm{TV}(\mathbb{P}_n, \mathbb{Q}_n) = \epsilon'$, we set testing threshold $\tau_\alpha'$ as the $(1 - \alpha)$-quantile of the estimated null distribution of Canonne's statistic by resampling method, which repeatedly re-computing the empirical estimator of distance with the samples randomly drawn from $\mathbb{P}_n$ and $\mathbb{Q}_n$.

Specifically, denote by $B$ the iteration number of resampling method. In $b$-th iteration ($b \in [B]$), we randomly draw two samples $X$ and $X'$ from $\mathbb{P}_n$, and two samples $Y$ and $Y'$ from $\mathbb{Q}_n$. The sample sizes are set to be the same as the size of testing samples. Denote by $X_i$ and $X_i'$ the occurrences of $\boldsymbol{z}_i$ in samples $X$ and $X'$ respectively, and let $Y_i$ and $Y_i'$ be the occurrences of $\boldsymbol{z}_i$ in samples $Y$ and $Y'$

respectively. We then calculate the test statistic based on total variation given in Canonne's test as

$$T_b' = \sum_{i=1}^{n} \frac{(X_i - Y_i)^2 - X_i - Y_i}{\widehat{f}_i},$$

with the term

$$\widehat{f}_i := \max\left\{|X_i' - Y_i'|, X_i' + Y_i', 1\right\}.$$

During such process, we obtain $B$ statistics $T_1', T_2', ..., T_B'$ and set testing threshold as

$$\tau_\alpha' = \arg\min_\tau \left\{\sum_{b=1}^{B} \frac{\mathbb{I}[T_b' \le \tau]}{B} \ge 1 - \alpha\right\}.$$

## D.3  DETAILS OF EXPERIMENTS WITH DISTRIBUTIONS OVER DIFFERENT DOMAINS

---

**Algorithm 2** Construction of distribution

---

**Input**: Two samples $Z$ and $Z'$, a kernel $\kappa$, step size $\eta$
**Output**: Two samples $Z$ and $Z'$
1: **for** NAMMD$(\mathbb{P}, \mathbb{Q}; \kappa) \ne \epsilon$ **do**
2:    Calculate the objective value $\mathcal{L}(Z, Z' \mid \kappa)$ according to Eqn. 11
3:    Calculate gradient $\nabla \mathcal{L}(Z, Z' \mid \kappa)$
4:    Gradient descend with step size $\eta$ by the Adam method
5: **end for**

---

Let $\mathbb{P}$ and $\mathbb{Q}$ be discrete uniform distributions over $Z = \{z_i\}_{i=1}^{m}$ and $Z' = \{z_i'\}_{i=1}^{m}$, respectively. As we can see, our NAMMD distance can be calculated as

$$
\begin{aligned}
\text{NAMMD}(\mathbb{P}, \mathbb{Q}; \kappa) &= \frac{\|\boldsymbol{\mu}_\mathbb{P} - \boldsymbol{\mu}_\mathbb{Q}\|_{\mathcal{H}_\kappa}^2}{4K - \|\boldsymbol{\mu}_\mathbb{P}\|_{\mathcal{H}_\kappa}^2 - \|\boldsymbol{\mu}_\mathbb{Q}\|_{\mathcal{H}_\kappa}^2} \\
&= \frac{1/m^2 \sum_{i,j} \kappa(z_i, z_j) + \kappa(z_i', z_j') - 2\kappa(z_i, z_j')}{4K - 1/m^2 \sum_{i,j} \left(\kappa(z_i, z_j) + \kappa(z_i', z_j')\right)}.
\end{aligned}
$$

Notably, NAMMD$(\mathbb{P}, \mathbb{Q}; \kappa) = 0$ can be effortlessly achieved by setting $Z = Z'$.

Here, we learn samples $Z$ and $Z'$ given NAMMD$(\mathbb{P}, \mathbb{Q}; \kappa) = \epsilon$ as follows

$$\mathcal{L}(Z, Z' \mid \kappa) = (\text{NAMMD}(\mathbb{P}, \mathbb{Q}; \kappa) - \epsilon)^2 \tag{11}$$

We take gradient method [74] for the optimization of Eqn. 11. Algorithm 2 presents the detailed description on optimization. The corresponding calculation of MMD$(\mathbb{P}, \mathbb{Q}; \kappa)$ is given as follows

$$
\begin{aligned}
\text{MMD}(\mathbb{P}, \mathbb{Q}; \kappa) &= \|\boldsymbol{\mu}_\mathbb{P} - \boldsymbol{\mu}_\mathbb{Q}\|_{\mathcal{H}_\kappa}^2 \\
&= 1/m^2 \sum_{i,j} \kappa(z_i, z_j) + \kappa(z_i', z_j') - 2\kappa(z_i, z_j').
\end{aligned}
$$

## D.4  DETAILS OF STATE-OF-THE-ART TWO-SAMPLE TESTING METHODS

The details of six state-of-the-art two-sample testing methods used in the experiments (which are summarized in Figure 2) for test power comparison.

- MMDFuse: A fusion of MMD with multiple Gaussian kernels via a soft maximum [32];
- MMD-D: MMD with a learnable Deep kernel [26];
- MMDAgg: MMD with aggregation of multiple Gaussian kernels and multiple testing [31];
- AutoTST: Train a binary classifier of AutoML with a statistic about class probabilities [49];
- ME$_{\text{MaBiD}}$: Embeddings over multiple test locations and multiple Mahalanobis kernels [29];
- ACTT: MMDAgg with an accelerated optimization via compression [50].

## D.5 DETAILS OF OUR NAMMDFUSE

Following the fusing statistics approach [32], we introduce the NAMMDFuse statistic through exponentiation of NAMMD with samples $X$ and $Y$ as follows

$$\widehat{\text{FUSE}}(X, Y) = \frac{1}{\lambda} \log \left( E_{\kappa \sim \pi(\langle X, Y \rangle)} \left[ \exp \left( \lambda \frac{\widehat{\text{NAMMD}}(X, Y; \kappa)}{\sqrt{\widehat{N}(X, Y)}} \right) \right] \right)$$

where $\lambda > 0$ and $\widehat{N}(X, Y) = \frac{1}{m(m-1)} \sum_{i \neq j}^{m} \kappa(\mathbf{x}_i, \mathbf{x}_j)^2 + \kappa(\mathbf{y}_i, \mathbf{y}_j)^2$ is permutation invariant. $\pi(\langle X, Y \rangle)$ is the prior distribution on the kernel space $\mathcal{K}$. In experiments, we set the prior distribution $\pi(\langle X, Y \rangle)$ and the kernel space $\mathcal{K}$ to be the same for MMDFuse.

## D.6 DETAILS OF DIFFERENT KERNELS

The details of the various kernels used in the experiments (which are summarized in Table 13) for test power comparison in two-sample testing, employing the same kernel for NAMMD and MMD.

- Gaussian: $\text{G}(\boldsymbol{x}, \boldsymbol{y}) = \exp(-\|\boldsymbol{x} - \boldsymbol{y}\|^2 / 2\gamma^2)$ for $\gamma > 0$ [77];
- Laplace: $\text{L}(\boldsymbol{x}, \boldsymbol{y}) = \exp(-\|\boldsymbol{x} - \boldsymbol{y}\|_1 / \gamma)$ for $\gamma > 0$ [32];
- Deep: $\text{D}(\boldsymbol{x}, \boldsymbol{y}) = [(1 - \lambda)\text{G}(\phi_\omega(\boldsymbol{x}), \phi_\omega(\boldsymbol{y})) + \lambda]\text{G}(\boldsymbol{x}, \boldsymbol{y})$ for $\lambda > 0$ and network $\phi_\omega$ [26];
- Mahalanobis: $\text{M}(\boldsymbol{x}, \boldsymbol{y}) = \exp\left(-(\boldsymbol{x} - \boldsymbol{y})^T M(\boldsymbol{x} - \boldsymbol{y}) / 2\gamma^2\right)$ for $\gamma > 0$ and $M \succ 0$ [29].

## D.7 DETAILS OF CONFIDENCE AND ACCURACY MARGINS

We can test the confidence margin between source dataset $S$ and target dataset $T$ for a model $f$. Let $f(x)$ represent the probability assigned by the model $f$ to the true label. We define the confidence margin as

$$|E_{\boldsymbol{x} \in S}[1 - f(\boldsymbol{x})] - E_{\boldsymbol{x} \in T}[1 - f(\boldsymbol{x})]| . \tag{12}$$

A smaller margin indicates similar model performance in the source and target dataset.

In a similar manner, we can also define the accuracy margin as follows

$$|E_{\boldsymbol{x} \in S}[f(\boldsymbol{x}; y_{\boldsymbol{x}})] - E_{\boldsymbol{x} \in T}[f(\boldsymbol{x}; y_{\boldsymbol{x}})]| , \tag{13}$$

where $f(\boldsymbol{x}; y_{\boldsymbol{x}}) = 1$ if the model $f$ correctly predicts the true label $y_{\boldsymbol{x}}$, and $f(\boldsymbol{x}; y_{\boldsymbol{x}}) = 0$ otherwise.

We present the confidence and accuracy margins between the original ImageNet and its variants in Table 6, with the values computed using the pre-trained ResNet50 model.

**Table 6:** Confidence and accuracy margins between the original ImageNet and its variants.

|  | ImageNetsk | ImageNetr | ImageNetv2 | ImageNeta |
|---|---|---|---|---|
| Accuracy Margin | 0.529 | 0.564 | 0.751 | 0.827 |
| Confidence Margin | 0.504 | 0.549 | 0.684 | 0.764 |

# E ADDITIONAL EXPERIMENTAL RESULTS

## E.1 COMPARISON WITH TOTAL VARIATION: SENSITIVITY TO SAMPLE STRUCTURE.

We demonstrate that our NAMMD better captures the differences between distributions by exploiting intrinsic structures. For each dataset, we sample ten elements and randomly selecting one element to serve as the base $\boldsymbol{z}_0$. The remaining elements are sorted as $\boldsymbol{z}_1, \boldsymbol{z}_2, ..., \boldsymbol{z}_9$ with $\|\boldsymbol{z}_0 - \boldsymbol{z}_1\|^2 \geq \|\boldsymbol{z}_0 - \boldsymbol{z}_2\|^2 \geq \cdots \geq \|\boldsymbol{z}_0 - \boldsymbol{z}_9\|^2$. For each element $\boldsymbol{z}_i$, we construct the Dirac distribution $\delta_{\boldsymbol{z}_i}$ with support only at element $\boldsymbol{z}_i$, and we calculate the distance $\text{NAMMD}(\delta_{\boldsymbol{z}_0}, \delta_{\boldsymbol{z}_i}, \kappa)$. We repeat this 10 times, using a Gaussian kernel with $\gamma = 1$ for blob, higgs, and hdgm, and $\gamma = 10$ for mnist.

From Figure 7, it is evident that our $\text{NAMMD}(\delta_{\boldsymbol{z}_0}, \delta_{\boldsymbol{z}_i}, \kappa)$ distance increases as $\|\boldsymbol{z}_0 - \boldsymbol{z}_i\|^2$ decrease for all datasets. This is different from previous total variation $\text{TV}(\delta_{\boldsymbol{z}_0}, \delta_{\boldsymbol{z}_i}) = 1$ for $i \in \{1, 2, ..., 9\}$,

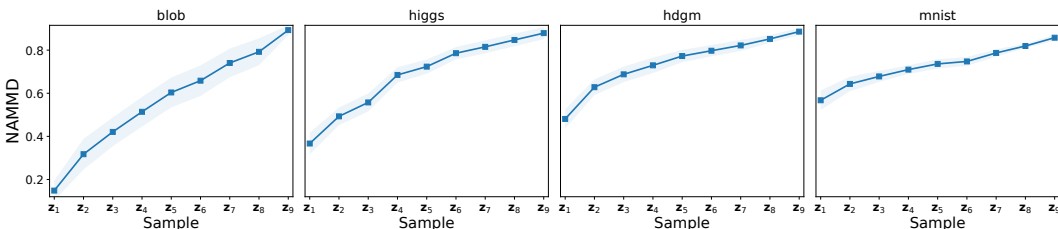

**Figure 7:** The NAMMD distance between $\delta(\boldsymbol{z}_0)$ and $\delta(\boldsymbol{z}_i)$ with $i \in \{1, 2, \ldots, 9\}$.

which merely measures the difference between probability mass functions of two distributions. In comparison, our NAMMD distance can effectively capture intrinsic structures and complex patterns in real-word datasets by leveraging kernel trick.

## E.2  COMPARISONS ON NAMMD AND WASSERSTEIN ON DISCRETE DISTRIBUTIONS

**Table 7:** Comparisons of test power (mean±std) between NAMMD and Wasserstein under different $\epsilon'$ values.

| Dataset | $\epsilon' = 0.1$ | | $\epsilon' = 0.3$ | | $\epsilon' = 0.5$ | | $\epsilon' = 0.7$ | |
|---|---|---|---|---|---|---|---|---|
| | NAMMD | Wasser. | NAMMD | Wasser. | NAMMD | Wasser. | NAMMD | Wasser. |
| blob | **.968**±.022 | .295±.047 | **.912**±.053 | .343±.037 | **.960**±.020 | .265±.023 | **.961**±.029 | .549±.051 |
| cifar10 | **.919**±.017 | .765±.020 | **.923**±.021 | .721±.022 | **.997**±.002 | .733±.024 | **.999**±.001 | .718±.026 |
| hdgm | **.942**±.023 | .866±.023 | **.946**±.017 | .823±.028 | **.965**±.014 | .821±.027 | **.989**±.004 | .788±.024 |
| higgs | **.908**±.050 | .612±.059 | **.947**±.027 | .427±.045 | **.962**±.023 | .548±.046 | **.995**±.005 | .548±.053 |
| mnist | **.931**±.024 | .867±.020 | **.965**±.007 | .818±.023 | **.997**±.001 | .823±.027 | **1.00**±.000 | .822±.034 |
| Average | **.934**±.027 | .681±.034 | **.939**±.025 | .626±.031 | **.976**±.012 | .638±.030 | **.989**±.008 | .685±.038 |

We compare the test power of DCT using our NAMMD against the Wasserstein distance on *discrete distributions supported on the same finite set*, following exactly the experimental setup of Table 1. As shown in Table 7, NAMMD-based DCT consistently achieves higher test power than the Wasserstein-based approach across different datasets and $\epsilon'$ levels.

## E.3  COMPARISONS ON RESPECTIVELY SELECTED KERNELS FOR MMD AND NAMMD.

**Table 8:** Comparisons of test power (mean±std) on distribution closeness testing with respect to different NAMMD values, and the bold denotes the highest mean between tests with our NAMMD and original MMD. Notably, different selected kernel are applied for NAMMD and MMD respectively in this table.

| Dataset | $\epsilon = 0.1$ | | $\epsilon = 0.3$ | | $\epsilon = 0.5$ | | $\epsilon = 0.7$ | |
|---|---|---|---|---|---|---|---|---|
| | MMD | NAMMD | MMD | NAMMD | MMD | NAMMD | MMD | NAMMD |
| blob | .939±.009 | **.983**±.004 | .968±.007 | **.991**±.002 | .952±.010 | **.999**±.001 | .934±.010 | **1.00**±.000 |
| higgs | .914±.051 | **.972**±.009 | .934±.056 | **.976**±.007 | .967±.021 | **.994**±.002 | .949±.036 | **.1.00**±.000 |
| hdgm | .925±.071 | **.976**±.005 | .915±.069 | **.978**±.004 | .913±.058 | **.984**±.004 | .938±.052 | **1.00**±.000 |
| mnist | .951±.006 | **.962**±.005 | .955±.032 | **.961**±.021 | .935±.049 | **.967**±.036 | .977±.011 | **.992**±.002 |
| cifar10 | .976±.012 | **.987**±.006 | .971±.007 | **.988**±.003 | .991±.004 | **1.00**±.000 | **1.00**±.000 | **1.00**±.000 |
| Average | .941±.030 | **.976**±.006 | .949±.034 | **.979**±.007 | .952±.028 | **.989**±.009 | .960±.022 | **.998**±.000 |

Similar to Table 2 (where the experiments are performed using the same kernel for both MMD and NAMMD), we conduct experiments with different selected kernels for NAMMD and MMD. For MMD, the kernel selection remains the same as in the experiments in Table 2, and we denote the kernel for MMD as $\kappa^{\mathrm{M}}$. However, for NAMMD, we select the kernel $\kappa^{\mathrm{N}}$ similar to the experiments in Table 2, but with an additional regularization term related to the norms of the original distributions in

the dataset (i.e., $4K - \|\boldsymbol{\mu}_{\mathbb{P}}\|^2_{\mathcal{H}_\kappa} - \|\boldsymbol{\mu}_{\mathbb{Q}}\|^2_{\mathcal{H}_\kappa}$) during the optimization. Notably, these kernel selection methods are heuristic for distribution closeness testing, as obtaining a test power estimator for distribution closeness testing with multiple distribution pairs and selecting an optimal global kernel for distribution closeness testing based on the estimator remain open questions and poses a significant challenge. We use $\kappa^{\mathrm{N}}$ for the construction distribution pairs $(\mathbb{P}_1, \mathbb{Q}_1)$ and $(\mathbb{P}_2, \mathbb{Q}_2)$. Following Definition 8, we perform NAMMD distribution closeness testing with $\kappa^{\mathrm{N}}$ and MMD distribution closeness testing with $\kappa^{\mathrm{M}}$ respectively. Table 8 summarizes the average test powers and standard deviations of NAMMD distribution closeness testing and MMD distribution closeness testing. It is evident that our NAMMD test achieves better performance than the MMD test, and this improvement when using different selected kernels for NAMMD and MMD can be explained by the analysis for distribution closeness testing based on Theorem 9.

### E.4 TYPE-I ERROR EXPERIMENTS

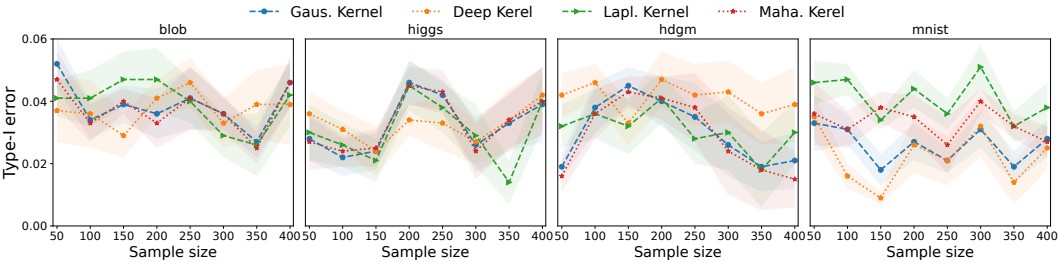

**Figure 8:** type-I error is controlled around $\alpha = 0.05$ w.r.t different kernels for our NAMMD test with $\epsilon = 0$.

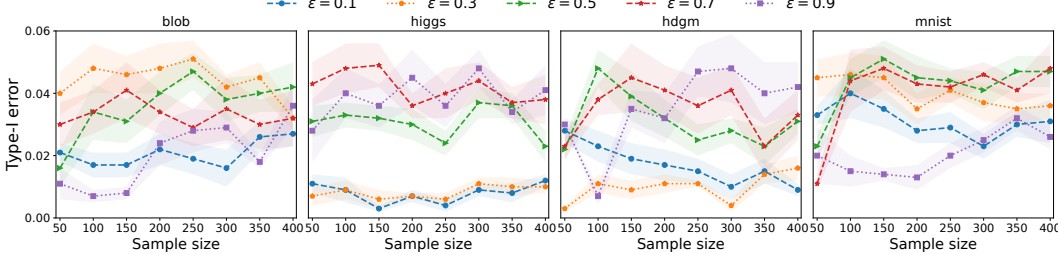

**Figure 9:** The type-I error is controlled around $\alpha = 0.05$ w.r.t different $\epsilon \in (0, 1)$ for our NAMMD test.

From Figure 8, it is evident that the type-I error of our NAMMD test is controlled around $\alpha = 0.05$ with respect to different kernels and datasets in two-sample testing (i.e. distribution closeness testing with $\epsilon = 0$) by using permutation tests. In a similar manner, Figure 9 shows that the type-I error of our NAMMD test is controlled around $\alpha = 0.05$ with respect to different $\epsilon \in (0, 1)$ and datasets in distribution closeness testing, where we derive the testing threshold based on asymptotic distribution. These results are nicely in accordance with Theorem 6.

**Table 9:** Type-I error (mean±std) across different sample sizes on ImageNet-SK.

|  | 20 | 30 | 40 | 50 | 60 | 70 | 80 | 90 |
|---|---|---|---|---|---|---|---|---|
| MMD | .034±.022 | .039±.014 | .047±.052 | .053±.034 | .041±.024 | .053±.035 | .046±.036 | .033±.008 |
| NAMMD | .045±.016 | .031±.014 | .037±.012 | .035±.025 | .024±.004 | .045±.016 | .047±.006 | .036±.018 |

**Table 10:** Type-I error (mean±std) across different sample sizes on ImageNet-V2.

|  | 20 | 30 | 40 | 50 | 60 | 70 | 80 | 90 |
|---|---|---|---|---|---|---|---|---|
| MMD | .051±.031 | .054±.022 | .048±.033 | .022±.014 | .018±.014 | .033±.014 | .033±.013 | .024±.005 |
| NAMMD | .041±.013 | .034±.022 | .049±.023 | .051±.014 | .020±.005 | .053±.024 | .036±.014 | .046±.005 |

**Table 11:** Type-I error (mean±std) across different sample sizes on ImageNet-A.

|  | 20 | 30 | 40 | 50 | 60 | 70 | 80 | 90 |
|---|---|---|---|---|---|---|---|---|
| MMD | .053±.013 | .037±.023 | .054±.024 | .029±.003 | .059±.016 | .049±.016 | .047±.008 | .043±.008 |
| NAMMD | .052±.012 | .040±.013 | .036±.004 | .048±.003 | .040±.005 | .043±.005 | .048±.007 | .043±.007 |

**Table 12:** Type-I error (mean±std) across different sample sizes on ImageNet-R.

|  | 20 | 30 | 40 | 50 | 60 | 70 | 80 | 90 |
|---|---|---|---|---|---|---|---|---|
| MMD | .031±.005 | .031±.005 | .040±.005 | .049±.004 | .050±.008 | .066±.005 | .044±.005 | .051±.007 |
| NAMMD | .035±.007 | .038±.004 | .043±.005 | .051±.005 | .051±.008 | .070±.005 | .049±.005 | .058±.009 |

Following Definition 8, we take ImageNet as $\mathbb{P}_1$ and $\mathbb{P}_2$, and sequentially assign each of its variants (ImageNet-SK, ImageNet-V2, ImageNet-A, and ImageNet-R) as both $\mathbb{Q}_1$ and $\mathbb{Q}_2$. In this setup, the null hypothesis holds because the distributional distance between $(\mathbb{P}_2, \mathbb{Q}_2)$ is not substantially larger than that between $(\mathbb{P}_1, \mathbb{Q}_1)$, and therefore the alternative hypothesis (i.e., $d(\mathbb{P}_2, \mathbb{Q}_2) > \epsilon$ with $\epsilon = d(\mathbb{P}_1, \mathbb{Q}_1)$) does not arise. From Table 9 to Table 12, both NAMMD- and MMD-based DCT maintain type-I error close to the nominal level $\alpha = 0.05$, even with limited sample sizes (e.g., 20).

### E.5    COMPARISONS ON VARIOUS KERNELS.

For further comparison, we evaluate our NAMMD test (with $\epsilon = 0$) against the MMD test in terms of test power with the same kernel. We perform this experiments across four frequently used kernels (Appendix D.6): 1). Gaussian kernel [77]; 2). Laplace kernel [32]; 3). Deep kernel [26]; 4). Mahalanobis kernel [29]. Following [29, 26], we learn kernels on a subset of each available dataset for 2000 epochs, and then test on 100 random same size subsets from remaining dataset. The ratio is set to 1 : 1 for training and test sample sizes. We repeat such process 10 times for each dataset. For our NAMMD test, the null hypothesis is NAMMD($\mathbb{P}, \mathbb{Q}, \kappa) = 0$, and we apply permutation test.

**Table 13:** Comparisons of test power (mean±std) on two-sample testing with the same kernel, and the bold denotes the highest mean between our NAMMD test and the original MMD test.

| Dataset | Gaus. Kernel | | Maha. Kernel | | Deep Kernel | | Lapl. Kernel | |
|---|---|---|---|---|---|---|---|---|
| | MMD | NAMMD | MMD | NAMMD | MMD | NAMMD | MMD | NAMMD |
| blob | .600±.090 | **.616±.090** | **1.00±.000** | **1.00±.000** | .859±.084 | **.863±.083** | .359±.088 | **.364±.088** |
| higgs | .563±.073 | **.566±.075** | .904±.087 | **.905±.086** | .796±.091 | **.797±.091** | .556±.062 | **.581±.062** |
| hdgm | .707±.042 | **.713±.041** | .801±.097 | **.805±.095** | .332±.087 | **.334±.086** | .090±.012 | **.100±.013** |
| mnist | .405±.019 | **.411±.020** | .970±.013 | **.975±.012** | .462±.100 | **.467±.098** | .873±.016 | **.881±.010** |
| cifar10 | .219±.017 | **.222±.020** | .984±.007 | **.987±.006** | .997±.003 | **1.00±.000** | .998±.002 | **1.00±.000** |
| Average | .499±.048 | **.506±.049** | .932±.041 | **.934±.040** | .689±.073 | **.692±.072** | .575±.036 | **.585±.035** |

Table 13 summarizes the average of test powers and standard deviations of our NAMMD test and the MMD test with the same kernel. NAMMD test achieves better performance than original MMD test as for Gaussian, Laplace, Mahalanobis and Deep kernels. It is because scaling maximum mean discrepancy with the norms of mean embeddings improves the effectiveness of NAMMD test in two-sample testing.

### E.6    RUNTIME COMPARISON.

Table 14 present the time costs of the proposed permutation-based method, NAMMDFuse (which aggregates multiple NAMMD statistics with different kernels, as shown in Appendix D.5). NAMMD-Fuse exhibits similar time costs to MMDFuse and is significantly faster than most baseline methods.

**Table 14:** Comparisons of runtime (seconds) on two-sample testing with permutation test, corresponding to the experiments shown in Figure 2.

| Samp. Size | ACTT | AutoTST | MEmabid | MMD-D | MMDAgg | MMDFuse | NAMMDFuse |
|---|---|---|---|---|---|---|---|
| 50 | 35.918 | 681.669 | 45.945 | 303.413 | 13.621 | 9.340 | 9.602 |
| 100 | 42.035 | 707.498 | 53.125 | 308.542 | 14.742 | 11.686 | 12.107 |
| 150 | 44.429 | 707.368 | 82.473 | 446.897 | 16.037 | 13.298 | 13.744 |
| 200 | 44.981 | 734.686 | 83.341 | 448.066 | 17.031 | 14.015 | 14.388 |
| 250 | 45.129 | 731.910 | 83.877 | 451.478 | 20.730 | 16.573 | 16.921 |
| 300 | 46.402 | 750.984 | 158.909 | 747.656 | 21.316 | 19.404 | 19.989 |
| 350 | 46.077 | 809.401 | 159.829 | 743.727 | 22.301 | 23.441 | 23.439 |
| 400 | 46.994 | 847.017 | 232.811 | 1025.473 | 23.655 | 27.632 | 27.845 |

Recall that the NAMMD is defined as:

$$
\begin{aligned}
\text{NAMMD}(\mathbb{P}, \mathbb{Q}; \kappa) &= \frac{\|\boldsymbol{\mu}_\mathbb{P} - \boldsymbol{\mu}_\mathbb{Q}\|_{\mathcal{H}_\kappa}^2}{4K - \|\boldsymbol{\mu}_\mathbb{P}\|_{\mathcal{H}_\kappa}^2 - \|\boldsymbol{\mu}_\mathbb{Q}\|_{\mathcal{H}_\kappa}^2} \\
&= \frac{\|\boldsymbol{\mu}_\mathbb{P}\|_{\mathcal{H}_\kappa}^2 + \|\boldsymbol{\mu}_\mathbb{Q}\|_{\mathcal{H}_\kappa}^2 - 2\langle \boldsymbol{\mu}_\mathbb{P}, \boldsymbol{\mu}_\mathbb{Q} \rangle_{\mathcal{H}_\kappa}}{4K - \|\boldsymbol{\mu}_\mathbb{P}\|_{\mathcal{H}_\kappa}^2 - \|\boldsymbol{\mu}_\mathbb{Q}\|_{\mathcal{H}_\kappa}^2} \\
&= \frac{E_{\boldsymbol{x}, \boldsymbol{x}' \sim \mathbb{P}^2}[\kappa(\boldsymbol{x}, \boldsymbol{x}')] + E_{\boldsymbol{y}, \boldsymbol{y}' \sim \mathbb{Q}^2}[\kappa(\boldsymbol{y}, \boldsymbol{y}')] - 2E_{\boldsymbol{x} \sim \mathbb{P}, \boldsymbol{y} \sim \mathbb{Q}}[\kappa(\boldsymbol{x}, \boldsymbol{y}')]}{4K - E_{\boldsymbol{x}, \boldsymbol{x}' \sim \mathbb{P}^2}[\kappa(\boldsymbol{x}, \boldsymbol{x}')] - E_{\boldsymbol{y}, \boldsymbol{y}' \sim \mathbb{Q}^2}[\kappa(\boldsymbol{y}, \boldsymbol{y}')]} .
\end{aligned}
$$

where the kernel $\kappa(\boldsymbol{x}, \boldsymbol{x}') = \Psi(\boldsymbol{x} - \boldsymbol{x}')$ is positive-definite with $\Psi(\boldsymbol{0}) = K$ and $\Psi(\boldsymbol{x} - \boldsymbol{x}') \leq K$ for all $\boldsymbol{x}, \boldsymbol{x}'$, and $K > 0$. Notably, the scaling term of NAMMD, $4K - \|\boldsymbol{\mu}_\mathbb{P}\|_{\mathcal{H}_\kappa}^2 - \|\boldsymbol{\mu}_\mathbb{Q}\|_{\mathcal{H}_\kappa}^2$, which can be efficiently computed using intermediate quantities from MMD, thus incurring negligible additional cost. Overall, the computational overhead introduced by NAMMD is minimal. In the formulation of NAMMD, all computations in the RKHS can be expressed as inner products, often computed via pairwise distances. This avoids the need to explicitly compute RKHS embeddings and helps reduce computational complexity. During the permutation test, we precompute the pairwise inner products and reuse them by rearranging the indices to obtain permutation results, eliminating the need to recompute them for each permutation. This strategy can be implemented efficiently.

## F    LIMITATION STATEMENT

Our analysis in this paper focuses on kernels of the form $\kappa(\boldsymbol{x}, \boldsymbol{x}') = \Psi(\boldsymbol{x} - \boldsymbol{x}') \leq K$ with a positive-definite $\Psi(\cdot)$ and $\Psi(\boldsymbol{0}) = K$, including Laplace [32], Mahalanobis [29] and Deep kernels [26] (frequently used in kernel-based hypothesis testing). For these kernels, *a larger norm of mean embedding $\|\boldsymbol{\mu}_\mathbb{P}\|_{\mathcal{H}_\kappa}^2$ indicates a smaller variance $Var(\mathbb{P}; \kappa) = K - \|\boldsymbol{\mu}_\mathbb{P}\|_{\mathcal{H}_\kappa}^2$, which corresponds to a more tightly concentrated distribution in the RKHS.* Leveraging this property, we gain the insight that two distributions can be separated more effectively at the same MMD distance with larger norms as discussed in Appendix D.1.2. Hence, we scale MMD using $4K - \|\boldsymbol{\mu}_\mathbb{P}\|_{\mathcal{H}_\kappa}^2 - \|\boldsymbol{\mu}_\mathbb{Q}\|_{\mathcal{H}_\kappa}^2$, making the new NAMMD increase with the norms $\|\boldsymbol{\mu}_\mathbb{P}\|_{\mathcal{H}_\kappa}^2$ and $\|\boldsymbol{\mu}_\mathbb{Q}\|_{\mathcal{H}_\kappa}^2$. Figure 1c and 1d demonstrate that our NAMMD exhibits a stronger correlation with the $p$-value in testing, while MMD is held constant. We also prove that scaling improves NAMMD's effectiveness as a closeness measure in Theorem 9.

However, all these improvements rely on the property that *"A larger norm of mean embedding $\|\boldsymbol{\mu}_\mathbb{P}\|_{\mathcal{H}_\kappa}^2$ indicates a smaller variance $Var(\mathbb{P}; \kappa) = K - \|\boldsymbol{\mu}_\mathbb{P}\|_{\mathcal{H}_\kappa}^2$, which corresponds to a more tightly concentrated distribution $\mathbb{P}$".* The proposed method may not work well for kernels where the embedding norm of distribution may increases as the data variance increases. For these kernels, the "less informative" of MMD still arises when assessing the closeness levels for multiple distribution pairs with the same kernel, i.e., MMD value can be the same for many pairs of distributions that have different norms in the same RKHS. We will demonstrate this by further considering two other types of kernels as follows.

**Unbounded kernels for bounded data**: For polynomial kernels of the form

$$\kappa(\mathbf{x}, \mathbf{x}') = (\mathbf{x}^T \mathbf{x}' + c)^d ,$$

We define $\mathbb{P}_1 = \{\frac{1}{4}, \frac{3}{4}\}$ and $\mathbb{Q}_1 = \{\frac{1}{2}, \frac{1}{2}\}$ be discrete distributions over vector domains $\{(\sqrt{c}, ..., 0), (-\sqrt{c}, ..., 0)\}$, respectively. Furthermore, we define $\mathbb{P}_2 = \{\frac{3}{4}, \frac{1}{4}\}$ and $\mathbb{Q}_2 = \{1, 0\}$ be

discrete distributions over domains $\{(\sqrt{c}, ..., 0), (-\sqrt{c}, ..., 0)\}$. It is evident that

$$\text{MMD}(\mathbb{P}_1, \mathbb{Q}_1; \kappa) = \text{MMD}(\mathbb{P}_2, \mathbb{Q}_2; \kappa) = \frac{1}{8}(2c)^d ,$$

with different norms for distributions pairs $\|\boldsymbol{\mu}_{\mathbb{P}_1}\|_{\mathcal{H}_\kappa}^2 + \|\boldsymbol{\mu}_{\mathbb{Q}_1}\|_{\mathcal{H}_\kappa}^2 = \frac{9}{8}(2c)^d$, and $\|\boldsymbol{\mu}_{\mathbb{P}_2}\|_{\mathcal{H}_\kappa}^2 + \|\boldsymbol{\mu}_{\mathbb{Q}_2}\|_{\mathcal{H}_\kappa}^2 = \frac{13}{8}(2c)^d$. Specifically, we have $\|\boldsymbol{\mu}_{\mathbb{P}_1}\|_{\mathcal{H}_\kappa}^2 = \frac{5}{8}(2c)^d$, $\|\boldsymbol{\mu}_{\mathbb{Q}_1}\|_{\mathcal{H}_\kappa}^2 = \frac{1}{2}(2c)^d$, $\|\boldsymbol{\mu}_{\mathbb{P}_2}\|_{\mathcal{H}_\kappa}^2 = \frac{5}{8}(2c)^d$ and $\|\boldsymbol{\mu}_{\mathbb{Q}_2}\|_{\mathcal{H}_\kappa}^2 = (2c)^d$.

In a similar manner, for matrix products kernels of the form

$$\kappa(\mathbf{x}, \mathbf{x}') = (\mathbf{x}^T M \mathbf{x}' + c)^d ,$$

and denote by $M_{11}$ the element in the first row and first column of the matrix $M$. We define $\mathbb{P}_1 = \{\frac{1}{4}, \frac{3}{4}\}$ and $\mathbb{Q}_1 = \{\frac{1}{2}, \frac{1}{2}\}$ over vector domains $\{(\sqrt{c/M_{11}}, ..., 0), (-\sqrt{c/M_{11}}, ..., 0)\}$, respectively. Furthermore, we define $\mathbb{P}_2 = \{\frac{3}{4}, \frac{1}{4}\}$ and $\mathbb{Q}_2 = \{1, 0\}$ over domains $\{(\sqrt{c/M_{11}}, ..., 0), (-\sqrt{c/M_{11}}, ..., 0)\}$. We obtain the same results as for polynomial kernels.

**Kernels with a positive limit at infinity**: Using the kernel as $\kappa(\mathbf{x}, \mathbf{x}') = \exp(-\frac{\|\mathbf{x}-\mathbf{x}'\|^2}{2\gamma})$ when $\|\mathbf{x} - \mathbf{x}'\|_\infty < K$, and otherwise $\kappa(\mathbf{x}, \mathbf{x}')$ with positive constants $K$ and $c$. We define $\mathbb{P}_1 = \{\frac{1}{4}, \frac{3}{4}\}$ and $\mathbb{Q}_1 = \{\frac{3}{4}, \frac{1}{4}\}$ over vector domains $\{(K, ..., 0), (4K, ..., 0)\}$, respectively. Furthermore, we define $\mathbb{P}_2 = \{\frac{1}{2}, \frac{1}{2}\}$ and $\mathbb{Q}_2 = \{1, 0\}$ over domains $\{(K, ..., 0), (4K, ..., 0)\}$. It is evident that

$$\text{MMD}(\mathbb{P}_1, \mathbb{Q}_1; \kappa) = \text{MMD}(\mathbb{P}_2, \mathbb{Q}_2; \kappa) = \frac{1}{2}(1 - c) ,$$

with different norms for pairs $\|\boldsymbol{\mu}_{\mathbb{P}_1}\|_{\mathcal{H}_\kappa} + \|\boldsymbol{\mu}_{\mathbb{Q}_1}\|_{\mathcal{H}_\kappa}^2 = \frac{5+3c}{4}$, and $\|\boldsymbol{\mu}_{\mathbb{P}_2}\|_{\mathcal{H}_\kappa}^2 + \|\boldsymbol{\mu}_{\mathbb{Q}_2}\|_{\mathcal{H}_\kappa}^2 = \frac{3+c}{2}$. Specifically, we have $\|\boldsymbol{\mu}_{\mathbb{P}_1}\|_{\mathcal{H}_\kappa}^2 = \frac{5+3c}{8}$, $\|\boldsymbol{\mu}_{\mathbb{Q}_1}\|_{\mathcal{H}_\kappa}^2 = \frac{5+3c}{8}$, $\|\boldsymbol{\mu}_{\mathbb{P}_2}\|_{\mathcal{H}_\kappa}^2 = \frac{1+c}{2}$ and $\|\boldsymbol{\mu}_{\mathbb{Q}_2}\|_{\mathcal{H}_\kappa}^2 = 1$.

For these kernels, the relationship between the norm of mean embedding and the variance of distribution is not monotonic, where a smaller norm of mean embedding may indicate a smaller variance or a larger variance, depending on the properties of the data distributions. Hence, when using these kernels for distribution closeness testing, mitigating the issue (i.e., MMD being the same for multiple pairs of distributions with different norms in the same RKHS) by incorporating norms of distributions becomes more challenging, potentially leading to a more complex distance design.

## G  STATEMENT ON THE USE OF LARGE LANGUAGE MODELS (LLMS)

LLMs were used solely as a general-purpose assistant to polish the writing and improve clarity. They did not contribute to research ideation, methodology, analysis, or results.

