# OpenReview forum: "A Kernel Distribution Closeness Testing"
_ICLR.cc/2026/Conference — Submitted to ICLR 2026_

### Official Review · Reviewer_m3FE · 2025-10-26

**Soundness:** 3
**Presentation:** 3
**Contribution:** 2
**Rating:** 4
**Confidence:** 3

**Summary:**

The paper proposes NAMMD (Norm-Adaptive Maximum Mean Discrepancy), a normalization of the standard MMD that divides the discrepancy measure by the sum of the RKHS norms of the kernel mean embeddings. This normalization is intended to adjust for scale differences between distributions and to improve robustness when comparing distributions with different variances or magnitudes. The authors demonstrate that under certain norm conditions (Theorem 9), NAMMD can outperform MMD, and they apply the resulting statistic to distribution closeness testing (DCT), comparing it with Canonne’s total-variation-based DCT. Experimental results show promising improvements in detection power across several benchmarks.

**Strengths:**

- The paper introduces a conceptually simple yet elegant modification of MMD that aims to adapt to distributional scales, a long-standing issue in kernel-based two-sample testing.

- The theoretical analysis (especially Theorem 9) provides partial intuition for when NAMMD may outperform MMD.

- The empirical results indicate improved detection power in some settings, particularly for scale-shifted distributions.

- The work situates itself in the broader line of kernel-based hypothesis testing and could inspire further studies on adaptive or normalized discrepancies.

**Weaknesses:**

- The motivation for the specific normalization by the sum of RKHS norms remains unclear. It is not obvious why this normalization is preferable to other kernel-based measures such as Kernel Canonical Correlation Analysis (KCCA; Akaho, 2001) or Hilbert–Schmidt Independence Criterion (HSIC; Gretton et al., 2005), both of which normalize by feature-space variance or covariance to account for scale differences. Clarifying the conceptual distinction between NAMMD and these established approaches would help position the contribution more precisely.

- The paper does not clearly situate NAMMD relative to prior normalized or relative MMD variants, including Normalized MMD (Muandet et al., 2012) and Relative MMD (Bounliphone et al., 2015). It remains somewhat ambiguous whether NAMMD introduces a fundamentally new normalization principle or whether it can be viewed as a reformulation of these earlier approaches.

- The paper should explain how p-values are computed in the NAMMD-based test (e.g., via asymptotic approximation, permutation, or bootstrap). Since the normalization modifies the scaling of the MMD statistic, its asymptotic variance and null distribution could differ from those of standard MMD. A comparison would clarify how the normalization affects calibration and test power.

- Theorem 9 considers only a specific norm condition favoring NAMMD. A more systematic analysis of how kernel bandwidths or scale ratios between P and Q influence this condition would clarify whether it is merely sufficient or also necessary.

- In Section 5.1, only average test power is reported. The empirical type-I error rates of both Canonne’s total-variation-based DCT (Canonne et al., 2023) and the NAMMD-based DCT are not shown, making it unclear whether both tests operate at the same nominal significance level. Reporting empirical type-I error under the null hypothesis would make the comparison more rigorous and interpretable.

[References]
- Akaho, S. (2001). A kernel method for canonical correlation analysis.
- Gretton, A., Bousquet, O., Smola, A., & Schölkopf, B. (2005). Measuring statistical dependence with Hilbert–Schmidt norms.
- Muandet, K., Fukumizu, K., Sriperumbudur, B. K., & Schölkopf, B. (2012). Learning from distributions via support measure machines.
- Bounliphone, W., Bellet, A., & Tommasi, M. (2015). A test of relative similarity for model selection in generative models.
- Canonne, C. L., Kamath, G., & Steinke, T. (2023). Distribution closeness testing via total variation.

**Questions:**

- Conceptual distinction — Could the authors clarify how NAMMD differs conceptually from correlation-based kernel measures such as KCCA (Akaho, 2001) or HSIC (Gretton et al., 2005)? In what sense does the normalization by the sum of RKHS norms capture a different form of scale or variance adjustment?

- Relation to prior normalized MMDs — How does NAMMD relate to previously proposed normalized or relative MMD variants, such as those by Muandet et al. (2012) or Bounliphone et al. (2015)? Is the proposed normalization theoretically novel, or can it be interpreted as a reparameterization of these approaches?

- Statistical inference — How are p-values estimated in the NAMMD-based test? Since the normalization changes the scaling of MMD, does its null distribution differ in variance or asymptotic form? A short explanation or comparison would improve reproducibility and theoretical clarity.

- Norm condition in Theorem 9 — Theorem 9 focuses on a specific norm condition that favors NAMMD. Could the authors discuss whether this condition might fail, and what happens in such regimes? For example, would NAMMD reduce to standard MMD, or behave pathologically?

- Empirical validity — In Section 5.1, the comparison with Canonne’s total-variation-based DCT lacks type-I error reporting. Could the authors include empirical type-I error rates under the null to ensure that both methods are calibrated at the same nominal significance level (e.g., α = 0.05)?

---

> ### Author Response · Authors · 2025-11-22
>
> We are grateful for your careful and detailed reviews and for the time you devoted to our manuscript. We address your questions as follows.
>
> >[W1] The motivation for the specific normalization by the sum of RKHS norms remains unclear. It is not obvious why this normalization is preferable to other kernel-based measures such as Kernel Canonical Correlation Analysis (KCCA; Akaho, 2001) or Hilbert–Schmidt Independence Criterion (HSIC; Gretton et al., 2005), both of which normalize by feature-space variance or covariance to account for scale differences. Clarifying the conceptual distinction between NAMMD and these established approaches would help position the contribution more precisely.
> >[Q1] Conceptual distinction — Could the authors clarify how NAMMD differs conceptually from correlation-based kernel measures such as KCCA (Akaho, 2001) or HSIC (Gretton et al., 2005)? In what sense does the normalization by the sum of RKHS norms capture a different form of scale or variance adjustment?
>
> [W1Q1A] Thank you for your question. **We propose NAMMD for an important problem that has not been extensively addressed in prior work: distribution closeness testing (DCT)**. KCCA and HSIC are correlation based kernel measures that are designed for different tasks, namely correlation analysis and independence testing between random variables. Their normalizations account for scale using feature space variance or covariance, which is appropriate when the goal is to measure dependence or correlation between random variables. In contrast, DCT is a different task that aims to assess the closeness/difference between two distributions in RKHS. For this purpose, the relevant scale is the magnitude of the RKHS mean embeddings. **Different distribution pairs may share the same MMD value but differ greatly in their RKHS norms**, which corresponds to different levels of closeness. This observation indicates that MMD is less informative when comparing the closenss levels between different distribution pairs (for example, two pairs $(P_1, Q_1)$ and $(P_2, Q_2)$ may have the same MMD value, but $P_1$ and $Q_1$ may be easier to distinguish, that is, less close to each other, than $P_2$ and $Q_2$ because their RKHS norms differ). Hence, **NAMMD normalizes using the sum of RKHS norms, which matches the objective of DCT** rather than the correlation oriented goals of KCCA or HSIC.
>
> >[W2] The paper does not clearly situate NAMMD relative to prior normalized or relative MMD variants, including Normalized MMD (Muandet et al., 2012) and Relative MMD (Bounliphone et al., 2015). It remains somewhat ambiguous whether NAMMD introduces a fundamentally new normalization principle or whether it can be viewed as a reformulation of these earlier approaches.
> >[Q2] Relation to prior normalized MMDs — How does NAMMD relate to previously proposed normalized or relative MMD variants, such as those by Muandet et al. (2012) or Bounliphone et al. (2015)? Is the proposed normalization theoretically novel, or can it be interpreted as a reparameterization of these approaches?
>
> [W2Q2A] Thank you for the question. Similar to our response in [W1Q1A], both Normalized MMD (Muandet et al., 2012) and Relative MMD (Bounliphone et al., 2015) were proposed for **different tasks**, and their formulations do not include the type of normalization we introduce. Muandet et al. (2012) consider learning from distributions by using the raw kernel mean embeddings as features, and their statistic does not involve a variance based or norm based normalization. Bounliphone et al. (2015) propose a relative similarity test that evaluates whether a distribution $P$ or $Q$ is closer to a reference distribution $U$. Their statistic is based on a difference of two MMD values, such as $MMD(U, Q)$ minus $MMD(U, P)$, and does not involve any variance adjustment or normalization term.

---

> > ### Author Response · Authors · 2025-11-22
> >
> > >[W3] The paper should explain how p-values are computed in the NAMMD-based test (e.g., via asymptotic approximation, permutation, or bootstrap). Since the normalization modifies the scaling of the MMD statistic, its asymptotic variance and null distribution could differ from those of standard MMD. A comparison would clarify how the normalization affects calibration and test power.
> > >[Q3] Statistical inference — How are p-values estimated in the NAMMD-based test? Since the normalization changes the scaling of MMD, does its null distribution differ in variance or asymptotic form? A short explanation or comparison would improve reproducibility and theoretical clarity.
> >
> > [W3Q3A] Thank you for the question. We will add the details of the $p$ value computation. In this paper, we conduct statistical inference by computing the test statistic and comparing it with the $(1-\alpha)$-quantile of its null distribution. This is the **testing-threshold view** of hypothesis testing. It is equivalent to the **$p$-value view**, in which one first computes a $p$-value and then compares it with $\alpha$. The two views are fully consistent.
> >
> > For the NAMMD-based DCT, we estimate the $p$-value using the **asymptotic null distribution** derived in Lemma 2. Under the null hypothesis $H_0:\mathrm{NAMMD}(P,Q)\le\epsilon$, we focus on the boundary case $\mathrm{NAMMD}(P,Q)=\epsilon$, since this case determines the tightest null scenario for controlling the Type I error. In this setting, the NAMMD statistic has an estimated asymptotically normal form
> > $$
> > \sqrt{m}(\widehat{\mathrm{NAMMD}} - \epsilon)
> > \xrightarrow{d}
> > \mathcal N(0,\sigma_{X,Y}^2),
> > $$
> > with $\sigma_{X,Y}=\frac{\sqrt{((4m-8)\zeta_1+2\zeta_2)/(m-1)}}{(m^2-m)^{-1}\sum_{i\neq j }4K-\kappa(x_i,x_j)-\kappa(y_i,y_j)}$ (see Page 4, Eqn. (2) for more details).
> >
> > Based on this asymptotic distribution, the $p$-value is computed as
> > $$
> > p = 1 - \Phi\left(\frac{\sqrt{m}(\widehat{\mathrm{NAMMD}}-\epsilon)}
> > {\sigma_{X,Y}}\right).
> > $$
> >
> > The MMD-based DCT follows the same principle with the estimated asymptotically normal form from Lemma 12(Appendix D.1). Under the null hypothesis $H_0: MMD(P,Q)\leq\epsilon$, we have
> >
> > $$
> > \sqrt{m}(\widehat{\mathrm{MMD}} - \epsilon)
> > \xrightarrow{d}
> > \mathcal N(0,\widehat{\sigma}_{M}^2),
> > $$
> >
> > with $\widehat{\sigma}_{M}=\sqrt{((4m-8)\zeta_1+2\zeta_2)/(m-1)}$.
> >
> > Based on this asymptotic distribution, the $p$-value is computed as
> > $$
> > p = 1 - \Phi\left(\frac{\sqrt{m}(\widehat{\mathrm{MMD}}-\epsilon)}
> > {\widehat{\sigma}_{M}}\right).
> > $$

---

> > > ### Author Response · Authors · 2025-11-22
> > >
> > > >[W4] Theorem 9 considers only a specific norm condition favoring NAMMD. A more systematic analysis of how kernel bandwidths or scale ratios between P and Q influence this condition would clarify whether it is merely sufficient or also necessary.
> > > >[Q4] Norm condition in Theorem 9 — Theorem 9 focuses on a specific norm condition that favors NAMMD. Could the authors discuss whether this condition might fail, and what happens in such regimes? For example, would NAMMD reduce to standard MMD, or behave pathologically?
> > >
> > > [W4Q4A] Thanks for the insightful comments. In Theorem 9, we compare MMD and NAMMD **using the same fixed kernel**, and the condition $\\|\mu_{P1}\\|^2 + \\|\mu_{Q1}\\|^2 \leq \\|\mu_{P2}\\|^2 + \\|\mu_{Q2}\\|^2$ is a **sufficient condition** stated at the level of kernel mean embedding norms. **The effects of both the kernel bandwidth and the scale ratio between $P$ and $Q$ are already reflected through these norms (see Appendix D.6.2 for an example).** Since bandwidth choices and distributional scales enter the analysis via their impact on the kernel mean embeddings, the condition already subsumes their influence. A separate treatment of bandwidth or scale parameters would require a more detailed analysis that depends on both the specific distributions and the chosen kernel, including factors such as distributional shape, kernel type, and higher-order structure, which is beyond the scope of this work. We will clarify this point in the revised manuscript.
> > >
> > > In the context of comparing MMD and NAMMD in DCT, Theorem 9 focuses on the setting
> > > $$
> > > \mathrm{MMD}(P_1, Q_1) < \mathrm{MMD}(P_2, Q_2)
> > > \quad\text{and}\quad
> > > \mathrm{NAMMD}(P_1, Q_1) < \mathrm{NAMMD}(P_2, Q_2).
> > > $$
> > > Under this setting, when both NAMMD and MMD produce valid comparisons between $(P_1, Q_1)$ and $(P_2, Q_2)$, the condition $\\|\mu_{P1}\\|^2 + \\|\mu_{Q1}\\|^2 \leq \\|\mu_{P2}\\|^2 + \\|\mu_{Q2}\\|^2$ is a sufficient condition under which NAMMD achieves higher test power. This condition not always hold, but this does not affect the main motivation for proposing NAMMD. The motivation of our work is to mitigate a limitation of MMD in DCT: different distribution pairs may share the same MMD value while having different RKHS norms, making MMD less informative for assessing closeness levels. **Theorem 9 is not intended to motivate NAMMD, but rather to compare the test power of NAMMD and MMD in a setting where this limitation does not arise. Its purpose is to show that even when MMD performs well, NAMMD can still offer advantages under an interpretable condition**. We will revise the manuscript to make this connection more explicit.
> > >
> > > Furthermore, for clarity, we will discuss its practicality **under this specific setting**. Since $\mathrm{MMD}(P,Q) = \\|\mu_P\\|^2 - 2\langle \mu_P, \mu_Q\rangle + \\|\mu_Q\\|^2$, a larger sum $\\|\mu_P\\|^2 + \\|\mu_Q\\|^2$ typically contributes to a larger MMD value when the cross-term is fixed or comparable. Thus, the condition $\\|\mu_{P_1}\\|^2 + \\|\mu_{Q_1}\\|^2 \leq \\|\mu_{P_2}\\|^2 + \\|\mu_{Q_2}\\|^2$ is strongly aligned with (though not equivalent to) the practical requirement that $\mathrm{MMD}(P_1,Q_1)$ is smaller than $\mathrm{MMD}(P_2,Q_2)$. Empirically, this condition holds frequently in our experiments.
> > >
> > > The failure of this condition does not cause NAMMD to behave pathologically. Theoretically, we show that the NAMMD-based DCT is valid and controls the Type-I error (Theorem 6). In addition, Lemma 5 and Theorem 7 establish the consistency of the NAMMD test, ensuring that its power converges to 1 as the sample size increases. **These guarantees hold regardless of whether the condition $\\|\mu_{P1}\\|^2 + \\|\mu_{Q1}\\|^2 \leq \\|\mu_{P2}\\|^2 + \\|\mu_{Q2}\\|^2$ is satisfied**.

---

> > > > ### Author Response · Authors · 2025-11-22
> > > >
> > > > >[W5] In Section 5.1, only average test power is reported. The empirical type-I error rates of both Canonne’s total-variation-based DCT (Canonne et al., 2023) and the NAMMD-based DCT are not shown, making it unclear whether both tests operate at the same nominal significance level. Reporting empirical type-I error under the null hypothesis would make the comparison more rigorous and interpretable.
> > > > >[Q5] Empirical validity — In Section 5.1, the comparison with Canonne’s total-variation-based DCT lacks type-I error reporting. Could the authors include empirical type-I error rates under the null to ensure that both methods are calibrated at the same nominal significance level (e.g., α = 0.05)?
> > > >
> > > > [W5Q5A] Thank you for the question. All experiments in Section 5.1 are conducted with a significance level $\alpha = 0.05$. Since the comparison focuses on DCT performance, the relevant quantity is the **empirical type-I error of the DCT**, i.e., the probability of incorrectly rejecting $H_0: d(P,Q)\le \epsilon$ when it is true. We report DCT type-I error results of different $\epsilon$ in Figures 9 (Appendix E.7). For completeness, we additionally provide the empirical type-I error rates for the exact setting of Table 1, ensuring that both Canonne’s DCT and our NAMMD-based DCT are calibrated at the same level:
> > > >
> > > > |  | ϵ' = 0.1 | ϵ' = 0.1 | ϵ' = 0.3 |  ϵ' = 0.3   | ϵ' = 0.5 | ϵ' = 0.5  | ϵ' = 0.7 | ϵ' = 0.7  |
> > > > |---------|----------|--------|----------|--------|----------|--------|----------|--------|
> > > > |   | Canonne's | NAMMD | Canonne's | NAMMD | Canonne's | NAMMD | Canonne's | NAMMD |
> > > > | blob    | .052±.008 | .034±.008 | .040±.004 | .057±.013 | .047±.008 | .058±.006 | .044±.008 | .051±.009 |
> > > > | higgs   | .060±.010 | .060±.013 | .056±.008 | .048±.011 | .042±.009 | .061±.010 | .037±.007 | .065±.010 |
> > > > | hdgm    | .056±.009 | .037±.004 | .040±.009 | .035±.008 | .048±.012 | .045±.008 | .044±.008 | .044±.008 |
> > > > | mnist   | .053±.007 | .053±.012 | .067±.013 | .065±.010 | .045±.007 | .057±.004 | .054±.011 | .058±.007 |
> > > > | cifar10 | .039±.010 | .046±.006 | .037±.005 | .058±.010 | .044±.007 | .045±.007 | .063±.008 | .044±.007 |
> > > > | Average | .052±.007 | .046±.010 | .048±.012 | .053±.010 | .045±.002 | .053±.007 | .048±.009 | .052±.008 |

---

> > > > > ### Comment · Reviewer_m3FE · 2025-11-25
> > > > >
> > > > > Thank you to the authors for their detailed rebuttal and for carefully addressing my comments. Below I briefly summarize which concerns have been resolved and which remain only partially addressed.
> > > > >
> > > > > - Motivation for NAMMD (W1Q1):
> > > > >    The rebuttal clarified the intended role of NAMMD as a statistic for DCT and the use of RKHS norms to represent how easily two distributions can be distinguished. This improves my understanding, and I consider this concern to be partially addressed. However, the rationale for choosing this particular normalization still feels somewhat heuristic. I would encourage the authors to explain this motivation more clearly and convincingly in the main text.
> > > > >
> > > > > - Relation to normalized/relative MMD and HSIC/KCCA (W1Q1, W2Q2):
> > > > >    The explanation that previous “normalized MMD”, Relative MMD, and HSIC/KCCA address different problems and do not employ the same RKHS-norm scaling largely resolves my concerns about novelty and positioning. For the camera-ready version, it would be helpful to present these differences more concretely, including formulas, and to use terminology that avoids confusion with earlier uses of “normalized MMD”.
> > > > >
> > > > > - Asymptotic normality and finite-sample calibration (W3Q3):
> > > > >    The intended use of asymptotic normality at the boundary of the composite null is now clear and is methodologically standard, so my procedural concern is resolved. At the same time, regarding finite-sample calibration, I would still like to see a more systematic investigation, for example by examining type-I error behavior across different sample sizes and kernel choices, or by comparing asymptotic thresholds with permutation or bootstrap-based thresholds.
> > > > >
> > > > > - Condition and scope of Theorem 9 (Q3, W4Q4):
> > > > >    The rebuttal clarified that the norm condition in Theorem 9 is a technical assumption for a specific comparison scenario and not the main motivation for NAMMD as a whole. I now have a better understanding of this result. I would appreciate it if the paper explicitly stated that the scope of this theorem is limited.
> > > > >
> > > > > - Type-I error and comparison to Canonne et al. (W5Q5):
> > > > >    The plan to add type-I error results for the same setting as in Table 1 directly addresses my earlier concern. Assuming the new table demonstrates reasonable calibration, I consider this point resolved.
> > > > >
> > > > >
> > > > > The rebuttal has improved my understanding and has mitigated several of my concerns, but the strength of the motivation for the normalization and the finite-sample calibration still feel only partially resolved. I therefore keep my overall score unchanged.

---

> > > > > > ### Author Response · Authors · 2025-11-26
> > > > > >
> > > > > > Thank you for your additional comments and for taking the time to engage with our rebuttal. We are glad that the earlier clarifications were helpful, and we will integrate them into the main text to improve clarity and reduce potential confusion. We address your remaining questions below.
> > > > > >
> > > > > > >[Q1] However, the rationale for choosing this particular normalization still feels somewhat heuristic.
> > > > > >
> > > > > > [A1] Thank you for the question. We understand the concern regarding whether the proposed normalization appears heuristic. We will clarify this motivation more explicitly in the revised manuscript by relating our design to prior work. Several prior normalization strategies for kernel tests have been developed in the **two-sample testing (TST)** setting, but their objectives differ from ours. For example, Kernel FDA [1] focuses on testing equality of the mean and covariance of two distributions; Spectral-MMD [2] introduces normalization to achieve minimax separation rates under $\Delta$-separated alternatives; and analytic or feature-based approaches [3–4] are constructed to improve sensitivity to local discrepancies. These methods are highly effective for detecting differences under $P=Q$, the classical TST null. However, DCT places more emphasis on **tolerating small deviations** rather than amplification of all differences, so the motivations behind these normalizations do not directly align with the closeness-testing objective.
> > > > > >
> > > > > > Our normalization is **instead motivated by a concrete limitation of MMD in DCT**: different distribution pairs can have **the same MMD value** while their RKHS norms are very different, which actually have different levels of closeness. As shown in Figure 1c, even when MMD stays the same, the $p$-values become smaller as the RKHS norms grow, meaning that distribution pairs with larger norms are easier to tell apart and therefore *less* close. The goal of NAMMD is to account for this RKHS-norm information and reduce this mismatch, rather than to optimize the statistic under traditional TST criteria.
> > > > > >
> > > > > > [1] Eric, M., Bach, F., & Harchaoui, Z. (2007). Testing for homogeneity with kernel Fisher discriminant analysis. Advances in Neural Information Processing Systems, 20.
> > > > > >
> > > > > > [2] Hagrass, O., Sriperumbudur, B., & Li, B. (2024). Spectral regularized kernel two-sample tests. The Annals of Statistics, 52(3), 1076-1101.
> > > > > >
> > > > > > [3] Chwialkowski, K. P., Ramdas, A., Sejdinovic, D., & Gretton, A. (2015). Fast two-sample testing with analytic representations of probability measures. Advances in Neural Information Processing Systems, 28.
> > > > > >
> > > > > > [4] Jitkrittum, W., Szabó, Z., Chwialkowski, K. P., & Gretton, A. (2016). Interpretable distribution features with maximum testing power. Advances in Neural Information Processing Systems, 29.

---

> > > > > > > ### Author Response · Authors · 2025-11-26
> > > > > > >
> > > > > > > >[Q2] At the same time, regarding finite-sample calibration, I would still like to see a more systematic investigation, for example by examining type-I error behavior across different sample sizes and kernel choices, or by comparing asymptotic thresholds with permutation or bootstrap-based thresholds.
> > > > > > >
> > > > > > > [A2] Thank you for the suggestion. We agree that additional evidence on finite-sample calibration is valuable. Below we report type-I error across different sample sizes under the same setting as Figure 3. Both MMD and NAMMD remain stable across different sample sizes.
> > > > > > >
> > > > > > > ImageNet-SK
> > > > > > > | | **10**        | **20**        | **30**        | **40**        | **50**        | **60**        | **70**        | **80**        | **90**        | **100**       |
> > > > > > > | ---------- | ------------- | ------------- | ------------- | ------------- | ------------- | ------------- | ------------- | ------------- | ------------- | ------------- |
> > > > > > > | **MMD**    | 0.043 ± 0.011 | 0.034 ± 0.022 | 0.039 ± 0.014 | 0.047 ± 0.052 | 0.053 ± 0.034 | 0.041 ± 0.024 | 0.053 ± 0.035 | 0.046 ± 0.036 | 0.033 ± 0.008 | 0.050 ± 0.027 |
> > > > > > > | **NAMMD**  | 0.030 ± 0.012 | 0.045 ± 0.016 | 0.031 ± 0.014 | 0.037 ± 0.012 | 0.035 ± 0.025 | 0.024 ± 0.004 | 0.045 ± 0.016 | 0.047 ± 0.006 | 0.036 ± 0.018 | 0.032 ± 0.017 |
> > > > > > >
> > > > > > > ImageNet-V2
> > > > > > > | | **10**        | **20**        | **30**        | **40**        | **50**        | **60**        | **70**        | **80**        | **90**        | **100**       |
> > > > > > > | ---------- | ------------- | ------------- | ------------- | ------------- | ------------- | ------------- | ------------- | ------------- | ------------- | ------------- |
> > > > > > > | **MMD**    | 0.031 ± 0.009 | 0.051 ± 0.031 | 0.054 ± 0.022 | 0.048 ± 0.033 | 0.022 ± 0.014 | 0.018 ± 0.014 | 0.033 ± 0.014 | 0.033 ± 0.013 | 0.024 ± 0.005 | 0.021 ± 0.003 |
> > > > > > > | **NAMMD**  | 0.029 ± 0.012 | 0.041 ± 0.013 | 0.034 ± 0.022 | 0.049 ± 0.023 | 0.051 ± 0.014 | 0.020 ± 0.005 | 0.053 ± 0.024 | 0.036 ± 0.014 | 0.046 ± 0.005 | 0.051 ± 0.003 |
> > > > > > >
> > > > > > > ImageNet-A
> > > > > > > |  | **10**        | **20**        | **30**        | **40**        | **50**        | **60**        | **70**        | **80**        | **90**        | **100**       |
> > > > > > > | ---- | ----- | ----- | ---- | ------ | ---- | --- | ----- | ------ | ------ | ----- |
> > > > > > > | **MMD**    | 0.033 ± 0.021 | 0.053 ± 0.013 | 0.037 ± 0.023 | 0.054 ± 0.024 | 0.029 ± 0.003 | 0.059 ± 0.016 | 0.049 ± 0.016 | 0.047 ± 0.008 | 0.043 ± 0.008 | 0.045 ± 0.005 |
> > > > > > > | **NAMMD**  | 0.043 ± 0.011 | 0.052 ± 0.012 | 0.040 ± 0.013 | 0.036 ± 0.004 | 0.048 ± 0.003 | 0.040 ± 0.005 | 0.043 ± 0.005 | 0.048 ± 0.007 | 0.043 ± 0.007 | 0.046 ± 0.005 |
> > > > > > >
> > > > > > > ImageNet-R
> > > > > > > | | **10**        | **20**        | **30**        | **40**        | **50**        | **60**        | **70**        | **80**        | **90**        | **100**       |
> > > > > > > | ---- | -------- | ------- | ------ | ----- | ----- | ---- | ---- | ----- | ------ | ---- |
> > > > > > > | **MMD**    | 0.038 ± 0.005 | 0.031 ± 0.005 | 0.031 ± 0.005 | 0.040 ± 0.005 | 0.049 ± 0.004 | 0.050 ± 0.008 | 0.066 ± 0.005 | 0.044 ± 0.005 | 0.051 ± 0.007 | 0.062 ± 0.007 |
> > > > > > > | **NAMMD**  | 0.042 ± 0.007 | 0.035 ± 0.007 | 0.038 ± 0.004 | 0.043 ± 0.005 | 0.051 ± 0.005 | 0.051 ± 0.008 | 0.070 ± 0.005 | 0.049 ± 0.005 | 0.058 ± 0.009 | 0.064 ± 0.006 |
> > > > > > >
> > > > > > > We also include Type-I error results under different kernel choices, shown below. Overall, both MMD and NAMMD remain stable across different kernels.
> > > > > > >
> > > > > > > | Kernel      | NAMMD (mean ± std) | MMD (mean ± std) |
> > > > > > > | --- | ---| ---|
> > > > > > > | Gaussian    |      0.051 ± 0.025 |    0.047 ± 0.009 |
> > > > > > > | Laplace     |      0.040 ± 0.025 |    0.039 ± 0.016 |
> > > > > > > | Mahalanobis |      0.051 ± 0.022 |    0.035 ± 0.016 |
> > > > > > > | Deep kernel |      0.047 ± 0.025 |    0.054 ± 0.022 |
> > > > > > >
> > > > > > >
> > > > > > > Regarding permutation or bootstrap-based thresholds, we will clarify this point more explicitly in the revision. Permutation [1] calibration is widely used in TST, **but it does not directly carry over to DCT**: it relies on exchangeability under the null $H_0:P=Q$, which holds only when $\epsilon=0$. Under the DCT null, the two distributions are allowed to differ, so exchangeability is generally not guaranteed.
> > > > > > >
> > > > > > > Bootstrap-based calibration is also non-trivial in DCT. Most existing bootstrap variants [2-4] for kernel two-sample tests are developed for the strict-equality null and **aim to approximate the null distribution around $P=Q$**. In DCT, however, the relevant calibration needs to reflect the composite, margin-based null $H_0:d(P,Q)\leq\epsilon$.
> > > > > > >
> > > > > > > [1] Hemerik, J., & Goeman, J. (2018). Exact testing with random permutations. Test, 27(4), 811-825.
> > > > > > >
> > > > > > > [2] Janssen, P. (1994). Weighted bootstrapping of U-statistics. Journal of statistical planning and inference, 38(1), 31-41.
> > > > > > >
> > > > > > > [3] Huang, B., Liu, Y., & Peng, L. (2023). Weighted bootstrap for two-sample u-statistics. Journal of Statistical Planning and Inference, 226, 86-99.
> > > > > > >
> > > > > > > [4] Chwialkowski, K., Sejdinovic, D., & Gretton, A. (2014). A wild bootstrap for degenerate kernel tests. Advances in neural information processing systems, 27.

---

### Official Review · Reviewer_6EMd · 2025-10-31

**Soundness:** 2
**Presentation:** 2
**Contribution:** 2
**Rating:** 4
**Confidence:** 3

**Summary:**

The paper studies the problem of distribution closeness testing (DCT), extending the usual two-sample test. The authors propose to use a normalized version of maximum mean discrepancy for this task. Asymptotic theories are developed for their tests.

**Strengths:**

1, The experiments are broad and cover both synthetic and real-world datasets.
2, The paper identifies an important but underexplored problem: distribution closeness testing (DCT)

**Weaknesses:**

1, the paper proposes a normalizing approach, but the paper does not rigorously prove that this scaling yields an optimal variance normalization or minimizes any formal criterion (e.g., unbiasedness or asymptotic efficiency). Thus, NAMMD’s normalization remains heuristic rather than theoretically grounded.

2, I find it somewhat concerning that the theory does not unify DCT and TST despite superficial similarity. The paper reverts to permutation calibration, admitting that the asymptotic distribution does not apply when $\varepsilon = 0$.

3, While the paper provides many figures and comparisons, key experimental parameters (e.g., sample sizes for Fig. 1, variance of estimators) are not reported, limiting reproducibility and interpretability of the less informative claim.

4, Although DCT is motivated as a closeness test, the practical interpretation of $\varepsilon$ (what level of MMD/NAMMD difference implies model transferability or acceptable domain shift) is not concretely defined. In the ImageNet experiments, the choice of $\varepsilon$ and its connection to performance metrics appear ad-hoc.

5, Although DCT is motivated by high-dimensional tasks, the paper does not analyze NAMMD’s behavior under the curse of dimensionality.

6, Most importantly, this papers does not discuss or compare NAMMD to other variance normalized versions of MMD.

**Questions:**

I find figure 1 very misleading and have the following questions:

1, how is the $p$-value computed? Do you use permutations to get the $p$-value.

2, Assuming the p-values are computed using permutations, it depends on the sample size, which is not reported. The number of repetitions is not reported either.

3, Theoretically, when P is not equal to Q, MMD converges to a positive number. While if you permute the samples, it converges to 0. So, p-value is expected to be small.

---

> ### Author Response · Authors · 2025-11-22
>
> We are grateful for your careful and detailed reviews and for the time you devoted to our manuscript. We address your questions as follows.
>
> >[W1] the paper proposes a normalizing approach, but the paper does not rigorously prove that this scaling yields an optimal variance normalization or minimizes any formal criterion (e.g., unbiasedness or asymptotic efficiency). Thus, NAMMD’s normalization remains heuristic rather than theoretically grounded.
>
> [W1A] Thank you for the question. We would like to clarify that NAMMD is proposed specifically for a **new task: distribution closeness testing (DCT)**. Its normalization is not intended to provide an optimal variance correction in the classical two-sample testing setting, nor to minimize criteria such as unbiasedness or asymptotic efficiency. Instead, the normalization is **motivated by a concrete limitation of MMD in DCT**: different distribution pairs can have **the same MMD value** while their RKHS norms are very different, which actually have different levels of closeness. As shown in Figure 1c, even when MMD stays the same, the $p$-values become smaller as the RKHS norms grow, meaning that distribution pairs with larger norms are easier to tell apart and therefore *less* close. The goal of NAMMD is to account for this RKHS-norm information and reduce this mismatch, rather than to optimize the statistic under traditional TST criteria.
>
> We do not claim that NAMMD's normalization is theoretically grounded in terms of two-sample testing. We will emphsize that and make our paper sit within a proper postion.
>
> >[W2]  I find it somewhat concerning that the theory does not unify DCT and TST despite superficial similarity. The paper reverts to permutation calibration, admitting that the asymptotic distribution does not apply when $\epsilon=0$.
>
> [W2A] Thank you for the question. Although DCT and TST share similar test statistics, they are fundamentally **different hypothesis testing problems** because they test on different null and alternative hypotheses. In TST, the goal is to test the null hypothesis $H_0: P = Q$ against the alternative $H_1: P \neq Q$. In contrast, DCT tests $H_0: d(P, Q) \leq \epsilon$ versus $H_1: d(P, Q) > \epsilon$ for some $\epsilon > 0$. Because the null hypotheses differ, the asymptotic null distributions of the test statistics are also different. **Permutation test is routinely used in TST but can not apply to DCT:** permutation relies on the exchangeability of $P$ and $Q$ under $H_0: P = Q$, which holds only when $\epsilon = 0$. Although the asymptotic distribution used in DCT does not apply when $\epsilon = 0$, the procedure remains valid and controls Type-I error. This is explicitly established in Theorem 6, whose proof includes the boundary case $\epsilon = 0$. Moreover, Lemma 5 and Theorem 7 imply the consistency of the NAMMD-based test (for both TST and DCT), ensuring that the test power converges to $1$ as the sample size grows.

---

> > ### Author Response · Authors · 2025-11-22
> >
> > >[W3] While the paper provides many figures and comparisons, key experimental parameters (e.g., sample sizes for Fig. 1, variance of estimators) are not reported, limiting reproducibility and interpretability of the less informative claim.
> >
> > >[Q1] I find figure 1 very misleading and have the following questions: 1).how is the $p$-value computed? Do you use permutations to get the $p$-value. 2).Assuming the p-values are computed using permutations, it depends on the sample size, which is not reported. The number of repetitions is not reported either. 3).Theoretically, when $P$ is not equal to $Q$, MMD converges to a positive number. While if you permute the samples, it converges to $0$. So, $p$-value is expected to be small.
> >
> > [W3Q1A] Thank you for the detailed questions. We are happy to clarify the experimental settings for Figure 1.
> > **Computation of the p-values.** The $p$-values in Figure 1 are obtained using a standard permutation test. Specifically, for illustration, we use two underlying 2-dimensional Gaussian distributions with a MMD value $MMD(P,Q)=0.15$ and we use 4 samples from each distribution for every experiment. A Gaussian kernel with bandwidth $1$ is used throughout.
> > **Permutation procedure.** For each test, we approximate the null distribution using **1000 permutations**. To reduce randomness induced by the small sample size, we repeat the entire procedure **100 times**, and Figure 1 reports the **average $p$-value and its standard deviation** over these repetitions.
> > **Why $p$-values do not converge to zero.** Although the MMD statistic converges to a positive population value when $P\neq Q$, with 4 samples, the empirical MMD estimator exhibits high variance. As a result, the observed statistic often overlaps substantially with the permutation distribution, and the resulting permutation $p$-value does not approach zero even under $P\neq Q$. This behavior is consistent with finite-sample properties of MMD under small $n$. The variance of the MMD estimator used in Figure 1 is reported in Figure 6 of Appendix B.
> >
> > **Variance of NAMMD.** For completeness, we also report the estimated variance of the NAMMD estimator under the same setting:
> > | Norm | 0.1      | 0.2      | 0.3      | 0.4      | 0.5      | 0.6      | 0.7      | 0.8      | 0.9      | 1.0      |
> > |------|----------|----------|----------|----------|----------|----------|----------|----------|----------|----------|
> > | Estimated Variance | 0.057584±0.001530 | 0.072078±0.000201 | 0.069961±0.001807 | 0.072001±0.001876 | 0.061721±0.002529 | 0.055402±0.002425 | 0.047977±0.002533 | 0.039194±0.003513 | 0.020620±0.002390 | 0.000000±0.000000 |
> >
> > When the RKHS norms are equal to 1, i.e., $\\|\mu_P\\|^2_k = \\|\mu_Q\\|^2_k = 1$, the two distributions degenerate into point-mass distributions. In this synthetic setting, the NAMMD estimator becomes a constant by construction. Consequently, its variance is zero, which corresponds to the last column (Norm = 1.0) in the table. We will add these experimental details to the appendix to improve clarity and reproducibility.

---

> > > ### Author Response · Authors · 2025-11-22
> > >
> > > >[W4] I though DCT is motivated as a closeness test, the practical interpretation of $\epsilon$ (what level of MMD/NAMMD difference implies model transferability or acceptable domain shift) is not concretely defined. In the ImageNet experiments, the choice of $\epsilon$ and its connection to performance metrics appear ad-hoc.
> > >
> > > [W4A] Thank you for the question. The parameter $\epsilon$ is a prespecified closeness level chosen before running the test, and its value depends on the application and the practitioner’s notion of “acceptable closeness.” In practice, one can determine $\epsilon$ using a **pair of reference distributions**, as discussed in *Performing DCT in Practice* (Page 5). For example, although ImageNet and Pascal VOC are from different distributions, the model trained on ImageNet can still have good performance on Pascal VOC. Hence, the NAMMD value between ImageNet and Pascal VOC provides a natural closeness level for this task, and we use it as the prespecified $\epsilon$. Once $\epsilon$ is fixed in this way, we can evaluate another dataset $Z$ by comparing its NAMMD distance to ImageNet with the ImageNet–Pascal VOC distance. If the ImageNet–$Z$ distance is below this tolerance level, then $Z$ is considered sufficiently close for the model to transfer well; otherwise, the shift is regarded as sigifinicant large.
> > >
> > > In the ImageNet experiments, we follow exactly this framework. We set $\epsilon$ to be the NAMMD value between the original ImageNet and one of its variants. We then test whether another variant of ImageNet is farther from the original ImageNet than this reference level. **In the testing procedure, the ground truth about whether the test variant is actually closer or farther than the reference variant is not known in advance. We understand that the presentation appear confusing because we only showed test power results under the setting where the test variant is indeed farther.** To address this, we have now added results for the complementary setting where the test variant is closer, which is the type-I error results as follows.
> > >
> > > ImageNet-SK
> > > | | **10**        | **20**        | **30**        | **40**        | **50**        | **60**        | **70**        | **80**        | **90**        | **100**       |
> > > | ---------- | ------------- | ------------- | ------------- | ------------- | ------------- | ------------- | ------------- | ------------- | ------------- | ------------- |
> > > | **MMD**    | 0.043 ± 0.011 | 0.034 ± 0.022 | 0.039 ± 0.014 | 0.047 ± 0.052 | 0.053 ± 0.034 | 0.041 ± 0.024 | 0.053 ± 0.035 | 0.046 ± 0.036 | 0.033 ± 0.008 | 0.050 ± 0.027 |
> > > | **NAMMD**  | 0.030 ± 0.012 | 0.045 ± 0.016 | 0.031 ± 0.014 | 0.037 ± 0.012 | 0.035 ± 0.025 | 0.024 ± 0.004 | 0.045 ± 0.016 | 0.047 ± 0.006 | 0.036 ± 0.018 | 0.032 ± 0.017 |
> > >
> > > ImageNet-V2
> > > | | **10**        | **20**        | **30**        | **40**        | **50**        | **60**        | **70**        | **80**        | **90**        | **100**       |
> > > | ---------- | ------------- | ------------- | ------------- | ------------- | ------------- | ------------- | ------------- | ------------- | ------------- | ------------- |
> > > | **MMD**    | 0.031 ± 0.009 | 0.051 ± 0.031 | 0.054 ± 0.022 | 0.048 ± 0.033 | 0.022 ± 0.014 | 0.018 ± 0.014 | 0.033 ± 0.014 | 0.033 ± 0.013 | 0.024 ± 0.005 | 0.021 ± 0.003 |
> > > | **NAMMD**  | 0.029 ± 0.012 | 0.041 ± 0.013 | 0.034 ± 0.022 | 0.049 ± 0.023 | 0.051 ± 0.014 | 0.020 ± 0.005 | 0.053 ± 0.024 | 0.036 ± 0.014 | 0.046 ± 0.005 | 0.051 ± 0.003 |
> > >
> > > ImageNet-A
> > > |  | **10**        | **20**        | **30**        | **40**        | **50**        | **60**        | **70**        | **80**        | **90**        | **100**       |
> > > | ---------- | ------------- | ------------- | ------------- | ------------- | ------------- | ------------- | ------------- | ------------- | ------------- | ------------- |
> > > | **MMD**    | 0.033 ± 0.021 | 0.053 ± 0.013 | 0.037 ± 0.023 | 0.054 ± 0.024 | 0.029 ± 0.003 | 0.059 ± 0.016 | 0.049 ± 0.016 | 0.047 ± 0.008 | 0.043 ± 0.008 | 0.045 ± 0.005 |
> > > | **NAMMD**  | 0.043 ± 0.011 | 0.052 ± 0.012 | 0.040 ± 0.013 | 0.036 ± 0.004 | 0.048 ± 0.003 | 0.040 ± 0.005 | 0.043 ± 0.005 | 0.048 ± 0.007 | 0.043 ± 0.007 | 0.046 ± 0.005 |
> > >
> > > ImageNet-R
> > > | | **10**        | **20**        | **30**        | **40**        | **50**        | **60**        | **70**        | **80**        | **90**        | **100**       |
> > > | ---------- | ------------- | ------------- | ------------- | ------------- | ------------- | ------------- | ------------- | ------------- | ------------- | ------------- |
> > > | **MMD**    | 0.038 ± 0.005 | 0.031 ± 0.005 | 0.031 ± 0.005 | 0.040 ± 0.005 | 0.049 ± 0.004 | 0.050 ± 0.008 | 0.066 ± 0.005 | 0.044 ± 0.005 | 0.051 ± 0.007 | 0.062 ± 0.007 |
> > > | **NAMMD**  | 0.042 ± 0.007 | 0.035 ± 0.007 | 0.038 ± 0.004 | 0.043 ± 0.005 | 0.051 ± 0.005 | 0.051 ± 0.008 | 0.070 ± 0.005 | 0.049 ± 0.005 | 0.058 ± 0.009 | 0.064 ± 0.006 |

---

> > > > ### Author Response · Authors · 2025-11-22
> > > >
> > > > >[W5] Although DCT is motivated by high-dimensional tasks, the paper does not analyze NAMMD’s behavior under the curse of dimensionality.
> > > >
> > > > [W5A] Thank you for the question. DCT is indeed motivated by high dimensional and complex tasks, and this is also a key reason for considering MMD, which has been widely studied in such settings. Our NAMMD is built on the same kernel based framework as MMD together with a scaling term $4K-\\|\mu_P\\|^2-\\|\mu_Q\\|^2$, where the quantities $\\|\mu_P\\|^2$ and $\\|\mu_Q\\|^2$ are also included in the original MMD. Because NAMMD normalizes MMD using quantities in the same reproducing kernel Hilbert space, **its high dimensional behavior follows that of standard kernel methods**. Selecting kernels that can cope with high dimensional and complex data is crucial, and **we follow the deep kernel approach [1]**, which maps the original high dimensional data into a lower dimensional and more structured representation through a pretrained neural network before applying the kernel. As shown in Figures 3, 4 and 5, NAMMD performs well with deep kernels on CIFAR10 and ImageNet.
> > > >
> > > > [1] Liu, Feng, et al. "Learning deep kernels for non-parametric two-sample tests." International conference on machine learning. PMLR, 2020.
> > > >
> > > > >[W6] Most importantly, this papers does not discuss or compare NAMMD to other variance normalized versions of MMD.
> > > >
> > > > [W6A] Thank you for the question. We discuss related variance-normalized or covariance-aware variants of MMD on Page 4. Several prior works have also exploited second-order information in the RKHS: some analyze covariance operators to derive asymptotic null distributions for two-sample tests [41-44], while others spectrally regularize MMD to incorporate covariance information [45]. In contrast, our approach focuses on the trace of the covariance matrix to mitigate the comparability issues of MMD in distribution closeness testing, while remaining simple and easily estimable from finite sample.
> > > >
> > > > [41] Z. Harchaoui, F. R. Bach, and E. Moulines. Testing for homogeneity with kernel fisher discriminant analysis. In Advances in Neural Information Processing Systems 20, pages 609–616. Curran Associates, Dutchess, NY, 2007.
> > > >
> > > > [42] Z. Harchaoui, F. R. Bach, and E. Moulines. Kernel change-point analysis. In Advances in Neural Information Processing Systems 21, pages 609–616. Curran Associates, Dutchess, NY, 2008.
> > > >
> > > > [43] M. Kirchler, S. Khorasani, M. Kloft, and C. Lippert. Two-sample testing using deep learning. In The 23rd International Conference on Artificial Intelligence and Statistics, pages 1387–1398, Palermo, Italy, 2020.
> > > >
> > > > [44] J. Kübler, W. Jitkrittum, B. Schölkopf, and K. Muandet. Learning kernel tests without data splitting. In Advances in Neural Information Processing Systems 33, pages 6245–6255. Curran Associates, Dutchess, NY, 2020.
> > > >
> > > > [45] O. Hagrass, B. Sriperumbudur, and B. Li. Spectral regularized kernel two-sample tests. Annals of Statistics, 52(3):1076–1101, 2024.

---

### Official Review · Reviewer_Kwpk · 2025-11-01

**Soundness:** 3
**Presentation:** 3
**Contribution:** 2
**Rating:** 4
**Confidence:** 3

**Summary:**

In this paper, authors propose a testing statistic called nom-adaptive MMD for distribution closeness testing. Comparing to the traditional MMD, NAMMD adds a normalisation term,
and fixes the issue that MMD is the same for kernel mean embeddings with different RKHS norms.  The authors several a testing procedure to examine the closeness of the distributions and shows that the proposed NAMMD has a higher testing power on toy and real world problems.

**Strengths:**

The closeness test is indeed different from traditional two-sample tests and represents an underexplored area in hypothesis testing, with potential downstream applications such as classifier transfer.

The issue that kernel mean embeddings in MMD can have different RKHS norms but yield the same MMD value is a legitimate concern, and the solution proposed in Equation (1) is both elegant and easy to implement in practice.

The proposed NAMMD method appears to consistently achieve higher testing power than MMD and total variation–based test statistics, as demonstrated in the experimental results.

The paper is generally well-written.

**Weaknesses:**

The main concern with this paper is that its technical innovation and motivation appear to be unrelated. From Figures 1(a) and 1(b), I can indeed see that MMD is not ideal when a and b have the same MMD value but their kernel mean embeddings have different RKHS norms. However, it is unclear why this becomes an issue specifically in the context of DCT. Without establishing this connection, the motivation remains weak. If DCT is susceptible to this particular issue of MMD, the authors should clarify in the introduction.

The authors also do not clearly explain what distinguishes the DCT test statistic from traditional TST test statistics. From the definition in line 117, it seems that as long as a TST statistic outputs some form of dissimilarity measure, it could be used for DCT as well. This further adds to the confusion about why we should focus on the RKHS norm issue discussed in the previous section, as it is not evident how normalizing the MMD resolves any problem when using a TST statistic (MMD) for DCT.

In the introduction:

> "Besides, extending these methods using continuous total variation involves the estimation of the underlying density functions of the distributions [25, 26]"

However, this is not accurate. There have been many efforts to adapt total variation to continuous data. For example, [1] propose a general nonparametric estimator for integral probability metrics (of which TV, MMD, and Wasserstein distances are special cases). Similarly, [2] introduce a general framework for estimating f-divergences (of which TV is one example), which has been widely adopted in generative model training.

[1] https://arxiv.org/abs/0901.2698
[2] https://arxiv.org/abs/1606.00709

The empirical comparison also does not include other widely used discrepancy measures beyond MMD and TV in the discrete example. For instance, f-divergence–based measures have been used in TST and could serve as a natrual candidate test statistics for DCT.

**Questions:**

What is the main reason that authors consider MMD over other test statistics? For example, Wasserstein distance and classic divergence, both can be extneded to continous variable settings without estimating densities (see [1] and [2]).

Line 085: Specifically, the MMD value can be the same for many pairs of distributions that have different norms in the RKHS Hκ, which potentially have different closeness levels.

Figure 1 a and b are not helpful demonstrating this as P and Q are equally close in both figures, depending on how close you zoom. Could authors find a better illustration of this problem?

In Figure 1, c and d, why is it not desirable to have MMD stays constant while the p-value of TST changes?

Line 312, how realistic is the condition ||mu_p1||^2 + ||mu_q1||^2 <= ||mu_p2||^2 + ||mu_q2||^2? It looks like that the main results in the proof depends on this particular assumption and there are some explanations in Section 6.2. However, it seems to only suggest the sum of variance of kernel embedding of p1 and q1 should be greater than that of p2 and q2. How strengient is this condition in reality? Could this condition be translated into some more interpretable?

**Details Of Ethics Concerns:**

None.

---

> ### Author Response · Authors · 2025-11-22
>
> We are grateful for your careful and detailed reviews and for the time you devoted to our manuscript. We address your questions as follows.
>
> >[W1] The main concern with this paper is that its technical innovation and motivation appear to be unrelated. From Figures 1(a) and 1(b), I can indeed see that MMD is not ideal when a and b have the same MMD value but their kernel mean embeddings have different RKHS norms. However, it is unclear why this becomes an issue specifically in the context of DCT. Without establishing this connection, the motivation remains weak. If DCT is susceptible to this particular issue of MMD, the authors should clarify in the introduction.
>
> >[Q2] In Figure 1, c and d, why is it not desirable to have MMD stays constant while the p-value of TST changes?
>
> [W1Q2A] Thank you for the question. In DCT, the goal is to quantify how close two distributions are, so we expect the test statistic to reflect **different levels of distributional closeness** in a clear way. Although MMD has appealing properties when $P = Q$, it may fail to distinguish distribution pairs that have **the same MMD value but different RKHS norms** when $P \neq Q$. This becomes an issue for DCT because such pairs may in fact **represent different levels of closeness, yet MMD assigns them the same discrepancy value**, making it uninformative in this setting.
>
> **Figures 1c–d illustrate this point by investigating the corresponding MMD-based TST $p$-values.** Since the $p$-value measures how unlikely the observed statistic would be under $H_0: P = Q$, a smaller $p$-value indicates that $P$ and $Q$ are **less likely to be close**. In MMD-based TST, the $p$-value is the probability of obtaining a MMD statistic at least as large as the observed one under the null hypothesis $H_0: P = Q$. A smaller $p$-value means that the observed MMD would be unlikely if $P$ and $Q$ were actually identical, providing stronger evidence against $H_0$. Therefore, a smaller $p$-value indicates that $P$ and $Q$ are more significantly different—that is, less close—even when their MMD values are the same.
>
> >[W2] The authors also do not clearly explain what distinguishes the DCT test statistic from traditional TST test statistics. From the definition in line 117, it seems that as long as a TST statistic outputs some form of dissimilarity measure, it could be used for DCT as well. This further adds to the confusion about why we should focus on the RKHS norm issue discussed in the previous section, as it is not evident how normalizing the MMD resolves any problem when using a TST statistic (MMD) for DCT.
>
> [W2A] Thank you for the question. Classical TST statistics are constructed for testing *exact equality* of distributions, i.e., the null hypothesis $P = Q. Although these statistics output a notion of discrepancy, **DCT imposes an additional requirement**: distributions that are easier to distinguish should yield larger statistic values. Under this criterion, standard MMD can be uninformative for DCT (as discussed in [W1A]), because **the same MMD value can correspond to very different levels of distinguishability**. In particular, when the RKHS norms $\\|\mu_P\\|^2_k$
> and $\\|\mu_Q\\|^2_k$ grow, the permutation $p$-value decreases—indicating the distributions are more distinguishable—yet the MMD value itself does not increase.
>
> To mitigate this limitation, we rescale the MMD statistic with RKHS norms
> $$
> 4K - \\|\mu_P\\|^2_{\kappa} - \\|\mu_Q\\|^2_{\kappa},
> $$
> so that NAMMD increases with the RKHS norms $\\|\mu_P\\|^2_{\kappa}$ and $\\|\mu_Q\\|^2_{\kappa}$. This normalization ensures that stronger distributional differences correspond to larger statistic values. As shown in Figure 1d, a larger NAMMD indeed leads to a smaller (p)-value, indicating that the two distributions are more distinguishable.

---

> > ### Author Response · Authors · 2025-11-22
> >
> > >[W3] In the introduction:"Besides, extending these methods using continuous total variation involves the estimation of the underlying density functions of the distributions [25, 26]" However, this is not accurate. There have been many efforts to adapt total variation to continuous data. For example, [1] propose a general nonparametric estimator for integral probability metrics (of which TV, MMD, and Wasserstein distances are special cases). Similarly, [2] introduce a general framework for estimating f-divergences (of which TV is one example), which has been widely adopted in generative model training.
> >
> > [W3A] Thanks for pointing out these interesting references. We will revise the introduction to more accurately reflect existing nonparametric approaches for estimating TV and related divergences in continuous settings. The works [1] and [2] indeed provide general frameworks for estimating integral probability metrics and $f$-divergences, respectively, and these frameworks yield lower bounds for the total variation distance between continuous distributions. However, these results do not contradict the fact that **no estimator can achieve strong consistency for TV in full generality** [3], except under additional structural assumptions on the distribution family [4]. The main drawback of inconsistency in DCT is that the statistic fails to converge to the true closeness measure as the sample size increases. As a result, it cannot guarantee a correct ordering of distribution pairs by their closeness, which is essential for DCT. We will incorporate this discussion into the revised manuscript and add a brief comparison to alternative divergence measures.
> >
> > [1] Sriperumbudur, Bharath K., et al. "On integral probability metrics,\phi-divergences and binary classification." arXiv preprint arXiv:0901.2698 (2009).
> >
> > [2] Nowozin, Sebastian, Botond Cseke, and Ryota Tomioka. "f-gan: Training generative neural samplers using variational divergence minimization." Advances in neural information processing systems 29 (2016).
> >
> > [3] Devroye, Luc, and László Győrfi. "No empirical probability measure can converge in the total variation sense for all distributions." The Annals of Statistics (1990): 1496-1499.
> >
> > [4] Barron, Andrew R., Lhszl Gyorfi, and Edward C. van der Meulen. "Distribution estimation consistent in total variation and in two types of information divergence." IEEE transactions on Information Theory 38.5 (2002): 1437-1454.

---

> > > ### Author Response · Authors · 2025-11-22
> > >
> > > >[W4] The empirical comparison also does not include other widely used discrepancy measures beyond MMD and TV in the discrete example. For instance, f-divergence–based measures have been used in TST and could serve as a natrual candidate test statistics for DCT.
> > >
> > > [W4A] Thank you for raising this point. Several discrepancy measures such as TV and $f$-divergences have indeed been studied extensively in the TST. These methods, however, are typically designed for the setting where the null hypothesis is $P = Q$. In contrast, DCT is based on the margin-based null hypothesis $H_0:d(P,Q)\leq\epsilon$. This structural difference makes it **non-trivial to directly apply TST methods to DCT**, and we update the discussion to better reflect this distinction.
> > >
> > > Regarding TV-based DCT, most existing works focus on **sample-complexity bounds in discrete settings** and do not provide an implementable hypothesis-testing procedure at a significance level $\alpha$. In our discrete experiment, we follow the estimator in [1] and approximate the null distribution **using additional samples drawn from the known reference distribution** at a given $\epsilon$. This approximation is needed because TV does not admit a usable asymptotic null distribution.
> > >
> > > In contrast, our NAMMD-based DCT has a closed-form asymptotic null distribution, which allows us to conduct testing at a chosen significance level $\alpha$ **without the need for extra samples and additional approximation steps**. This yields a practical and statistically principled procedure that TV-based and $f$-divergence-based approaches do not currently provide for DCT for the margin-based null hypothesis $H_0:d(P,Q)\leq\epsilon$.
> > >
> > > For completeness, we additionally evaluate the Wasserstein distance using the **same null-approximation strategy as TV**. This approach suffers from the same limitation as TV, namely that the null distribution must be approximated from extra reference samples. The results aligned with Table 1 are presented below.
> > > | | ϵ' = 0.1 | ϵ' = 0.1 | ϵ' = 0.1 | ϵ' = 0.3 | ϵ' = 0.3 | ϵ' = 0.3 | ϵ' = 0.5 | ϵ' = 0.5 | ϵ' = 0.5 | ϵ' = 0.7 | ϵ' = 0.7 | ϵ' = 0.7 |
> > > |---------|-----------|----------------|----------------------|-----------|----------------|----------------------|-----------|----------------|----------------------|-----------|----------------|----------------------|
> > > |         | Canonone’s | NAMMD | Wasserstein | Canonone’s | NAMMD | Wasserstein | Canonone’s | NAMMD | Wasserstein | Canonone’s | NAMMD | Wasserstein |
> > > | blob    | .856±.023 | **.968±.022** | .295±.047 | .809±.014 | **.912±.053** | .343±.037 | .944±.013 | **.960±.020** | .265±.023 | .998±.002 | **.961±.029** | .549±.051 |
> > > | cifar10 | .686±.030 | **.919±.017** | .765±.020 | .751±.021 | **.923±.021** | .721±.022 | .917±.006 | **.997±.002** | .733±.024 | .981±.004 | **.999±.001** | .718±.026 |
> > > | hdgm    | .861±.011 | **.942±.023** | .866±.023 | .888±.016 | **.946±.017** | .823±.028 | .937±.014 | **.965±.014** | .821±.027 | .987±.004 | **.989±.004** | .788±.024 |
> > > | higgs   | .883±.015 | **.908±.050** | .612±.059 | .825±.010 | **.947±.027** | .427±.045 | .960±.005 | **.962±.023** | .548±.046 | .994±.003 | **.995±.005** | .548±.053 |
> > > | mnist   | .715±.021 | **.931±.024** | .867±.020 | .786±.026 | **.965±.007** | .818±.023 | .896±.013 | **.997±.001** | .823±.027 | .971±.008 | **1.00±.000** | .822±.034 |
> > > | Average | .800±.020 | **.934±.027** | .681±.034 | .812±.017 | **.939±.025** | .626±.031 | .931±.010 | **.976±.012** | .638±.030 | .986±.004 | **.989±.008** | .685±.038 |
> > >
> > >
> > > We will revise our paper to include these discussions and the additional experimental results.
> > >
> > > [1] C. L. Canonne, A. Jain, G. Kamath, and J. Li. The price of tolerance in distribution testing. In Proceedings of the 35th Conference on Learning Theory, pages 573–624, London, UK, 2022.

---

> > > > ### Author Response · Authors · 2025-11-22
> > > >
> > > > >[Q1] What is the main reason that authors consider MMD over other test statistics? For example, Wasserstein distance and classic divergence, both can be extneded to continous variable settings without estimating densities (see [1] and [2]). Line 085: Specifically, the MMD value can be the same for many pairs of distributions that have different norms in the RKHS Hκ, which potentially have different closeness levels. Figure 1 a and b are not helpful demonstrating this as P and Q are equally close in both figures, depending on how close you zoom. Could authors find a better illustration of this problem?
> > > >
> > > > [Q1A] Thanks for your question. Many discrepancy measures, such as Wasserstein distance and classical $f$-divergences, are indeed valuable tools for comparing distributions. Our decision to focus on MMD is mainly driven by the **testing requirements** of the DCT framework. Specifically, DCT needs a statistic for which we can characterize the **null distribution** and thereby conduct a test at a chosen **significance level $\alpha$** with Type I error control.
> > > >
> > > > Among the available choices, MMD offers certain practical advantages for this purpose. It comes with a well-established hypothesis-testing theory, admits unbiased U-statistic estimators, and has been extensively validated in high-dimensional and complex data settings, such as adversarial defense [1], adversarial attack detection [2-3], domain adaptation [4]. These properties make MMD straightforward to integrate into a principled testing procedure, which is essential for DCT.
> > > >
> > > > Regarding Figure 1(a) and 1(b), we appreciate the concern. We will add further discussion in the revision. The two panels use **the same axis scale** (not sure if there is a misunderstanding), and the scatterplots are intended only as a visual illustration rather than a geometric comparison in Euclidean space. **However, what appears “close’’ in Euclidean space is not the closeness notion we use in DCT; instead, DCT focuses on how easily the two distributions can be distinguished, and distributions that are easier to distinguish are regarded as “less close.’’**. In Figure 1b, the black circles and blue triangles form tight clusters with clearly separated locations and very limited overlap, making the two distributions easier to separate. In Figure 1a, the samples are more dispersed and interleaved, creating a larger overlap region and making separation more challenging. Although the point clouds may appear similarly close depending on the zoom level, the distinguishability between the distributions, and therefore their closeness in the sense relevant to DCT, is different. We will clarify this distinction in the revised manuscript.
> > > >
> > > > [1] Zhang, J., Rubinstein, B. I., Zhang, J., & Liu, F. One Stone, Two Birds: Enhancing Adversarial Defense Through the Lens of Distributional Discrepancy. In ICML 2025 Workshop on Reliable and Responsible Foundation Models.
> > > >
> > > > [2] Gao, R., Liu, F., Zhang, J., Han, B., Liu, T., Niu, G., & Sugiyama, M. (2021, July). Maximum mean discrepancy test is aware of adversarial attacks. In International Conference on Machine Learning (pp. 3564-3575). PMLR.
> > > >
> > > > [3] Zhang, S., Liu, F., Yang, J., Yang, Y., Li, C., Han, B., & Tan, M. (2023, July). Detecting adversarial data by probing multiple perturbations using expected perturbation score. In International conference on machine learning (pp. 41429-41451). PMLR.
> > > >
> > > > [4] Chen, C., Fu, Z., Chen, Z., Jin, S., Cheng, Z., Jin, X., & Hua, X. S. (2020, April). Homm: Higher-order moment matching for unsupervised domain adaptation. In Proceedings of the AAAI conference on artificial intelligence (Vol. 34, No. 04, pp. 3422-3429).

---

> > > > > ### Author Response · Authors · 2025-11-22
> > > > >
> > > > > >[Q3] Line 312, how realistic is the condition $\\|\mu_{P1}\\|^2+\\|\mu_{Q1}\\|^2\leq\\|\mu_{P2}\\|^2+\\|\mu_{Q2}\\|^2$? It looks like that the main results in the proof depends on this particular assumption and there are some explanations in Section 6.2. However, it seems to only suggest the sum of variance of kernel embedding of p1 and q1 should be greater than that of p2 and q2. How strengient is this condition in reality? Could this condition be translated into some more interpretable?
> > > > >
> > > > > [Q3A] Thanks for your question. We will add more discussion on this condition as follows. In the context of comparing MMD and NAMMD in DCT, **Theorem 9 focuses on the setting**
> > > > > $$
> > > > > \mathrm{MMD}(P_1, Q_1) < \mathrm{MMD}(P_2, Q_2)
> > > > > \quad\text{and}\quad
> > > > > \mathrm{NAMMD}(P_1, Q_1) < \mathrm{NAMMD}(P_2, Q_2).
> > > > > $$
> > > > > Under this setting, when both NAMMD and MMD produce valid comparisons between $(P_1, Q_1)$ and $(P_2, Q_2)$, **the condition $\\|\mu_{P1}\\|^2 + \\|\mu_{Q1}\\|^2 \leq \\|\mu_{P2}\\|^2 + \\|\mu_{Q2}\\|^2$ arises when NAMMD achieves higher test power**. Since $\mathrm{MMD}(P,Q) = \\|\mu_P\\|^2 - 2\langle \mu_P, \mu_Q\rangle + \\|\mu_Q\\|^2$, a larger sum $\\|\mu_P\\|^2 + \\|\mu_Q\\|^2$ typically contributes to a larger MMD value when the cross-term is fixed or comparable. Thus, the condition $\\|\mu_{P_1}\\|^2 + \\|\mu_{Q_1}\\|^2 \leq \\|\mu_{P_2}\\|^2 + \\|\mu_{Q_2}\\|^2$ is strongly aligned with (though not equivalent to) the practical requirement that $\mathrm{MMD}(P_1,Q_1)$ is smaller than $\mathrm{MMD}(P_2,Q_2)$. Empirically, this condition holds frequently in our experiments.
> > > > >
> > > > > Finally, we would like to clarify the role of this condition in our overall motivation. The main motivation of our work is to mitigate a limitation of MMD in DCT: different distribution pairs may share the same MMD value while having different RKHS norms, making MMD less informative for assessing different closeness levels. **Theorem 9 is not meant to motivate NAMMD, but to compare the test power of NAMMD and MMD in a setting where this limitation does not occur**. Its purpose is to show that even when MMD performs well, NAMMD can still offer advantages under an interpretable condition. We will revise the manuscript to make this connection more explicit.

---

### Meta-Review · Area_Chair_H8HZ · 2026-01-10

**Summary:**

The paper proposes a new method called Norm-Adaptive MMD (NAMMD) to address the problem of Distribution Closeness Testing (DCT), which determines whether the distance between two distributions is at least $\epsilon$-far. The authors argue that the standard Maximum Mean Discrepancy (MMD) is less informative because it does not account for the norm difference of distributions in the Reproducing Kernel Hilbert Space (RKHS). The proposed method NAMMD mitigate this problem by scales the MMD value based on the RKHS norms of the distributions. The authors provide theoretical proofs regarding the asymptotic distribution of NAMMD and claim higher test power with bounded Type-I error across various data types, including synthetic noise and images.

Although the authors raised an interesting question for the MMD based two sample test, the paper seems to require a significant refinement. Especially, the paper has the following concerns:
- Lack of Conceptual Clarity (DCT vs TST):  The distinction between Distribution Closeness Testing (DCT) and Two-Sample Testing (TST) is central to the paper’s motivation, yet the explanation provided is insufficient and remains at a superficial level. The authors suggest that DCT requires a statistic where "more distinguishable distributions should yield more distinguishable statistics," but this remains a subjective and qualitative argument. The paper fails to provide a rigorous, formal objective for why the DCT setting necessitates this specific shift in perspective over existing hypothesis testing frameworks.
- Ad-hoc Normalization Method: The core proposal of normalization appears ad-hoc. While the intention is to observe MMD values relative to RKHS norms, the authors do not provide a strong theoretical justification for why this specific form of scaling is necessary or optimal. There is a lack of comparative analysis against other potential normalization schemes, making the current approach feel like an empirical heuristic rather than a principled theoretical advancement.
- Inconsistencies in Experimental Rigor (Type-I Error): The experimental evaluation raises concerns regarding fairness and reliability. Although the authors specify a significance level, the reported Type-I errors exhibit fluctuations across different settings. Such instability makes it difficult to conclude that the comparison with the standard MMD-based DCT is fair or that the proposed NAMMD-based test is well-calibrated.

While the idea of adaptive MMD scaling is an interesting direction for handling complex data distributions, the current submission suffers from a weak conceptual foundation regarding the DCT task and a lack of rigorous verification in the experimental methodology. The ad-hoc nature of the proposed statistic and the inconsistent Type-I error rates suggest that the work requires more fundamental refinement before it is ready for publication.

**Reviewer Concerns:**

I think the reviewers provided proper reviews and I don't have any concerns on the reviewers' quality.

**Reviewer Scores:**

- Reviewer Kwpk: The reviewers main concern on the necessity of the normalization was addressed by the authors. However, it still lacks rigorous derivation of the method. Hence, the score can be remain.
- Reviewer 6EMd: The concern about the choice of $\epsilon$ was addressed by the authors. However, the choice is still ad-hoc and the rebuttal would not have fully satisfied the reviewer.
- Reviewer m3FE: I think the details of experiments were well addressed by the authors (e.g., calculation of p-value and significance level). On the other hand, the question about the choice of the normalization term was not properly addressed. Thus, I think the score would not have been changed so much even after the full discussion.

---

### Decision · Program_Chairs · 2026-01-26

Reject